# BTZSC: A Benchmark for Zero-Shot Text Classification Across Cross-Encoders, Embedding Models, Rerankers and LLMs

**Ilias Aarab**
European Central Bank*
Frankfurt am Main, Germany
Ilias.Aarab@ecb.europa.eu

## Abstract

Zero-shot text classification (ZSC) offers the promise of eliminating costly task-specific annotation by matching texts directly to human-readable label descriptions. While early approaches have predominantly relied on cross-encoder models fine-tuned for natural language inference (NLI), recent advances in text-embedding models, rerankers, and instruction-tuned large language models (LLMs) have challenged the dominance of NLI-based architectures. Yet, systematically comparing these diverse approaches remains difficult. Existing evaluations, such as MTEB, often incorporate labeled examples through supervised probes or fine-tuning, leaving genuine zero-shot capabilities underexplored. To address this, we introduce **BTZSC**, a comprehensive benchmark of 22 public datasets spanning sentiment, topic, intent, and emotion classification, capturing diverse domains, class cardinalities, and document lengths. Leveraging BTZSC, we conduct a systematic comparison across four major model families, NLI cross-encoders, embedding models, rerankers and instruction-tuned LLMs, encompassing 38 public and custom checkpoints. Our results show that: (i) modern rerankers, exemplified by *Qwen3-Reranker-8B*, set a new state-of-the-art with macro $F_1 = 0.72$; (ii) strong embedding models such as *GTE-large-en-v1.5* substantially close the accuracy gap while offering the best trade-off between accuracy and latency; (iii) instruction-tuned LLMs at 4–12B parameters achieve competitive performance (macro $F_1$ up to 0.67), excelling particularly on topic classification but trailing specialized rerankers; (iv) NLI cross-encoders plateau even as backbone size increases; and (v) scaling primarily benefits rerankers and LLMs over embedding models. BTZSC and accompanying evaluation code are publicly released to support fair and reproducible progress in zero-shot text understanding.[1]

## 1 Introduction

Text classification is a foundational problem in Natural Language Processing (NLP), finding broad applications across diverse domains, including topic categorization of news articles, intent detection in conversational agents, sentiment analysis of product reviews, and emotion recognition in mental health support systems (Sebastiani, 2002; Kowsari et al., 2019). Formally, the task involves assigning one or more predefined labels to textual data based solely on the content of the text (Sebastiani, 2002). However, the supervised approach to text classification necessitates the creation of large-scale, high-quality annotated datasets, a process that is often prohibitively expensive, particularly in specialized domains requiring expert annotators (Settles, 2012).

---

*The views expressed in this paper are of the author only and do not necessarily reflect those of the European Central Bank or the Eurosystem.

[1]Benchmark code and model checkpoints are available at https://github.com/IliasAarab/btzsc, benchmark datasets at https://huggingface.co/datasets/btzsc/btzsc, and the live leaderboard at https://huggingface.co/spaces/btzsc/btzsc-leaderboard.

Zero-shot text classification (ZSC) addresses this challenge by enabling models to predict labels that have not been explicitly observed during training (Yin et al., 2019). The core principle underlying ZSC methods is the exploitation of semantic relationships between input texts and candidate labels. This relationship is typically captured using pretrained language models, which encode semantics based on extensive pretraining on large textual corpora (Yin et al., 2019; Brown et al., 2020). One straightforward approach involves prompting (instruction-tuned) large autoregressive language models (LLMs) directly with textual inputs and candidate label descriptions. While effective, this method entails considerable computational cost and latency, limiting its feasibility in real-time deployment scenarios (Brown et al., 2020).

A widely adopted, more computationally efficient alternative involves fine-tuning pretrained encoder models on Natural Language Inference (NLI) datasets, reframing classification tasks as entailment problems. Specifically, the input text acts as a premise and each candidate label as a hypothesis sentence (Yin et al., 2019; Bowman et al., 2015; Williams et al., 2018). NLI datasets, including SNLI (Bowman et al., 2015) and MultiNLI (Williams et al., 2018), contain sentence pairs annotated with labels indicating entailment, contradiction, or neutrality. By fine-tuning encoders on these corpora, models learn to discern semantic compatibility, thus enabling effective reuse in ZSC scenarios. Despite their success and lower computational demands relative to generative LLMs, improvements in NLI-based cross-encoder methods have plateaued in recent years.

Concurrent to this, significant advances have occurred in the domain of text-embedding models (Reimers & Gurevych, 2019; Gao et al., 2021; Muennighoff et al., 2023). Embedding models learn mappings, $f : \text{text} \rightarrow \mathbb{R}^d$, from textual inputs to dense vector representations, ensuring semantically related texts are closely situated in the embedding space. This characteristic facilitates efficient similarity-based retrieval, and in principle, supports ZSC through nearest-neighbor matching to candidate label embeddings (Reimers & Gurevych, 2019; Gao et al., 2021). The Massive Text Embedding Benchmark (MTEB) systematically evaluates embedding models across various tasks, encompassing 58 datasets categorized into eight families (Muennighoff et al., 2023). However, classification performance within MTEB is primarily assessed through linear probes trained on labeled data atop frozen embeddings, thereby leaving the genuine zero-shot capabilities of embedding models untested (Muennighoff et al., 2023).

Another promising class of models, rerankers, originally cross-encoder or sequence-to-sequence architectures designed to refine the ranking of query-document pairs (e.g., MonoT5 (Nogueira et al., 2020)), can similarly be adapted for ZSC by treating textual inputs as queries and label descriptions as retrievable documents. However, the comparative performance and potential advantages of rerankers in zero-shot classification contexts remain underexplored.

Furthermore, the distinction between encoder-based and generative approaches is becoming increasingly blurred, as modern embedding models frequently leverage distilled or instruction-tuned variants of generative LLMs (e.g., Sentence-T5 (Ni et al., 2021), E5 (Wang et al., 2024)). Given these developments, a unified, controlled evaluation of all major model classes, NLI cross-encoders, embedding models, rerankers, and instruction-tuned LLMs, under zero-shot conditions is still lacking. Existing benchmarks either rely on supervised probes (as in MTEB (Enevoldsen et al., 2025)), focus exclusively on encoder architectures, or do not compare generative and non-generative methods under a consistent protocol.

To address this gap, we present a comprehensive benchmark study spanning 22 datasets across four major classification categories (sentiment, topic, intent, and emotion). This benchmark systematically explores the relative strengths, limitations, and transferability of these approaches, offering a comparative analysis to guide future research directions in zero-shot text classification.

## 2 RELATED WORK

To our knowledge, the proposed benchmark, **BTZSC**, is the first to jointly evaluate NLI cross-encoders, embedding models, rerankers, and instruction-tuned LLMs under a consistent, zero-shot classification protocol. Previous benchmarks for ZSC have typically been limited in scope, often restricted to evaluating a single model family, a narrow task category, or a handful of datasets. For instance, Yin et al. (2019) introduced a foundational NLI-based ZSC benchmark but evaluated exclusively cross-encoder models on only three datasets. Chalkidis et al. (2020) examined zero-shot

learning specifically within multi-label classification but confined their analysis to three hierarchical datasets. Gretz et al. (2023) proposed TTC23, evaluating prompt-based methods solely for topic classification and omitted contemporary embedding and reranking models from their analysis. Lepagnol et al. (2024) further explored the performance of smaller language models (100M-1B parameters) across 15 datasets, yet their work excluded comparisons with embedding and reranker architectures. The Massive Text Embedding Benchmark (MTEB), alongside its multilingual counterpart, has established a mature, broad-ranging evaluation platform covering numerous datasets. However, MTEB assesses classification performance via supervised linear probes trained atop frozen embeddings, thereby leaving unanswered the question of embedding models' genuine zero-shot capability (Muennighoff et al., 2023; Enevoldsen et al., 2025; Chung et al., 2025). Consequently, this fragmented state of evaluation has hindered a clear understanding of cross-family comparative capabilities among these diverse model types.

## 2.1 Zero-Shot Text Classification

Zero-shot text classification fundamentally involves assigning labels unseen during training by assessing semantic compatibility between input texts and candidate labels, typically expressed in natural language. Unlike supervised approaches, ZSC methods avoid task-specific fine-tuning by leveraging pretrained models' semantic representations. A common parallel in vision tasks is zero-shot image recognition with language-aligned models like CLIP (Radford et al., 2021), though textual classification benefits directly from the intrinsic expressivity and flexibility of natural language documents.

**NLI-based cross-encoders** represent one of the earliest and most prominent paradigms for zero-shot text classification. Such methods recast the classification problem into an entailment task, where each candidate label is paired with the input text as a hypothesis-premise pair scored by an NLI model (Yin et al., 2019). This approach has been operationalized effectively by public checkpoints like `facebook/bart-large-mnli` (Lewis et al., 2020), which powers the widely used zero-shot pipeline of Hugging Face Transformers (Wolf et al., 2020). More recent advances, including stronger encoder backbones like DeBERTa-v3 (He et al., 2023) and improved label verbalization techniques, have incrementally enhanced performance. Nonetheless, these improvements have plateaued when compared with rapid advancements from increasingly large generative language models (LLMs).

**Text-embedding models** have subsequently emerged as a highly active research domain, evolving significantly from early sentence embedding techniques such as InferSent (Conneau et al., 2017) and Google's Universal Sentence Encoder (USE) (Cer et al., 2018). Contemporary embedding frameworks, notably E5 (Wang et al., 2024), GTE (Li et al., 2023), BGE (Chen et al., 2024), and Qwen3-Embedding (Zhang et al., 2025), have substantially raised performance standards. These models integrate sophisticated training strategies including billion-scale contrastive pretraining, multilingual supervision, multi-stage data scaling, and instruction fine-tuning. For example, E5 uses an instruction-tuned approach with massive-scale contrastive learning, GTE emphasizes data-scale expansion over parameter scale, and BGE combines dense, sparse, and multi-vector encoding techniques into a multilingual framework capable of handling extensive context lengths. Compared to foundational architectures such as SBERT (Reimers & Gurevych, 2019), these advancements have resulted in improvements on standard benchmarks such as MTEB, demonstrating enhanced performance in semantic representation tasks (Muennighoff et al., 2023). Additionally, embedding models increasingly incorporate distillation from or joint-training with large generative models, effectively blurring distinctions between encoder-based and generative paradigms.

**Reranker models**, originally developed for information retrieval tasks, represent another promising approach for ZSC. Early reranker architectures leveraged cross-encoder models like BERT (Devlin et al., 2019), DPR's combined bi-encoder and cross-encoder architecture (Karpukhin et al., 2020), and late-interaction models such as ColBERT (Khattab & Zaharia, 2020). These methods typically assign relevance scores to a set of candidate documents with respect to a given input query, enabling them to be ranked accordingly. Sequence-to-sequence reranker variants such as MonoT5 have further extended this paradigm by scoring pairs through generative token likelihood estimation, demonstrating effective transferability to new tasks (Nogueira et al., 2020). Recent embedding model families like BGE now provide integrated reranker checkpoints, inheriting their multi-stage training procedures (Chen et al., 2024).

**Instruction-tuned LLMs** have emerged as a powerful paradigm for zero-shot classification, leveraging the general-purpose capabilities acquired through large-scale pretraining and subsequent instruction fine-tuning. Early work by Brown et al. (2020) demonstrated that sufficiently large autoregressive models could perform zero-shot classification via in-context prompting without task-specific training. Subsequent instruction-tuning methods, including FLAN (Wei et al., 2022) and InstructGPT (Ouyang et al., 2022), further enhanced zero-shot generalization by training models to follow natural language instructions across diverse tasks. Recent open-weight models such as LLaMA (Touvron et al., 2023), Mistral (Jiang et al., 2023), Qwen (Bai et al., 2023), and Gemma (Mesnard et al., 2024) have democratized access to instruction-following capabilities, enabling systematic evaluation across parameter scales. For zero-shot classification, these models are typically prompted with the input text and candidate labels, either selecting the label with highest generation probability or parsing a generated response (Sun et al., 2023). While instruction-tuned LLMs offer flexibility and strong performance on diverse tasks, they typically incur higher computational costs compared to lightweight encoder-based alternatives due to their larger parameter counts and autoregressive architecture, motivating research into efficient deployment strategies and smaller-scale variants (Lepagnol et al., 2024).

## 3    BENCHMARK FOR TEXTUAL ZERO-SHOT CLASSIFICATION (BTZSC)

BTZSC presents a comprehensive, task-balanced evaluation suite for zero-shot text classification[2], aiming to serve as a benchmark for diverse model architectures. BTZSC reuses a subset of the dataset pool compiled by Laurer et al. (2023) for transfer learning across a broad range of domains and task types, selecting only datasets we deem sufficiently high-quality and well-suited to zero-shot text classification, and repurposes them as a standardized evaluation suite to compare diverse model families under a unified protocol. The datasets underpin five key criteria to ensure robustness and real-world relevance. First, ensuring task diversity by including at least two datasets for each of sentiment, topic, intent, and emotion classification, mirroring the four most prominent application families. Second, to probe the impact of class granularity, BTZSC covers binary, medium-sized (such as *agnews* with four labels), and high-cardinality settings (for instance, *banking77* with 77 labels). Third, we prioritized domain diversity, drawing from sources spanning news, social media, product reviews, encyclopedic content, and political discourse to assess model robustness under domain shift. Fourth, we incorporated a wide spectrum of document lengths, from micro-texts (under 20 tokens) to longer articles (over 250 tokens). The benchmark is limited to English datasets; multilingual evaluation is left for future work. Full dataset details, including sources, licenses, and preprocessing, are provided in Appendix A.

BTZSC comprises 22 English datasets encompassing the aforementioned task types. As summarized in Table 1, each dataset is characterized by its number of classes, average token length[3], and domain area (such as news, review, or social media). To quantify lexical overlap and domain similarity between datasets, we follow (Thakur et al., 2021) and compute weighted Jaccard similarity by measuring token distribution overlaps for each dataset pair. The resulting $22 \times 22$ similarity matrix, shown in Figure 1, highlights low overlap between different task types, reflecting strong lexical diversity across tasks. At the same time, we observe that datasets derived from similar sources tend to cluster more together, for example, all Wikipedia-based datasets form a distinct group, as do the biasframes-related datasets, demonstrating modest intra-source lexical similarity.

### 3.1    EVALUATION METRICS

To make results comparable across all BTZSC tasks and model families, we adopt a *single, task-agnostic primary metric*: **macro F₁**. Macro averaging gives equal weight to every class irrespective

---

[2]Throughout this paper, we use the term zero-shot to denote that no fine-tuning or labeled examples from BTZSC tasks are used for training or model selection. Models are evaluated purely via document–label semantic matching. A key limitation is that the public datasets used in BTZSC may appear in the pretraining corpora of some models; as these corpora are often only partially documented, we cannot guarantee full corpus novelty. We mitigate this by checking publicly documented pretraining and supervised training data and avoiding models that explicitly list our datasets as supervised training targets, but undocumented overlap may still exist. This limitation is shared with other contemporary benchmarks such as MTEB (Enevoldsen et al., 2025).

[3]computed with the `answerdotai/ModernBERT` tokenizer

| Domain | Dataset | Num Classes | Avg Token Count |
|---|---|---|---|
| **Emotion** | | | |
| dialogue | empathetic | 32 | 132 |
| social-media | dair_ai_emotion | 6 | 20 |
| **Intent** | | | |
| banking | banking77 | 77 | 13 |
| social-media | biasframes_intent | 2 | 27 |
| assistant | massive_intent | 59 | 8 |
| **Sentiment** | | | |
| apps | appreviews | 2 | 49 |
| e-commerce | amazonpolarity | 2 | 103 |
| finance | financialphrasebank | 3 | 29 |
| local-business | yelpreviews | 2 | 164 |
| movies | imdb | 2 | 293 |
| movies | rottentomatoes | 2 | 26 |
| **Topic** | | | |
| education | trueteacher | 2 | 282 |
| news | agnews | 4 | 54 |
| politics | capsotu | 21 | 44 |
| politics | manifesto | 56 | 45 |
| qa-forum | yahootopics | 10 | 137 |
| social-media | biasframes_offensive | 2 | 27 |
| social-media | biasframes_sex | 2 | 28 |
| wikipedia | wikitoxic_insult | 2 | 93 |
| wikipedia | wikitoxic_obscene | 2 | 91 |
| wikipedia | wikitoxic_threat | 2 | 99 |
| wikipedia | wikitoxic_toxicaggregated | 2 | 86 |

Table 1: Summary statistics of BTZSC datasets.

of its frequency, making it appropriate for both binary and multi-class datasets with varying label set cardinalities (Sokolova & Lapalme, 2009). We additionally report (micro) **accuracy**, since it remains the most common headline number in the classification literature and is straightforward to interpret. For a more complete picture of model behavior across classes, we also report macro-averaged **precision** and **recall** for all tasks in Appendix D.

Finally, to probe whether success on natural-language inference transfers to zero-shot classification, we evaluate each model on standard NLI benchmarks (MNLI, ANLI, WANLI, FEVERNLI, LingNLI; details in Section 4) and report the **AUROC**. AUROC is threshold-free and does not require calibrated probabilities; because cosine-similarity scores lie in $[-1, 1]$ rather than representing probabilities, AUROC lets us test whether entailment pairs consistently receive higher similarity than neutral/contradiction pairs.

### 3.2 MODEL TYPES

We categorize the models evaluated in this study according to their underlying architecture and training strategies.

**Transformer Base Models.** As a baseline, we include transformer-based encoder models that have not been further fine-tuned for any specific downstream task. For these models, the final *[CLS]* token representation is extracted and cosine similarity is used to compute the relevance between the input text and each candidate label. The base models considered in this category are the original BERT (*bert-large-uncased* (Devlin et al., 2019)), the increasingly adopted ModernBERT (*ModernBERT-large* (Warner et al., 2024)), and DeBERTa-v3 (*deberta-v3-large* (He et al., 2023)), a popular and robust modification of BERT that has demonstrated strong performance on a variety of NLP benchmarks.

**NLI-based Cross-Encoders.** These models are trained on NLI datasets and perform classification by assessing the degree of entailment between an input text and each candidate label, formulated as a premise-hypothesis pair. *BART-Large-MNLI* is included as the canonical representative, being the first widely used NLI-based cross-encoder for zero-shot classification. We also consider

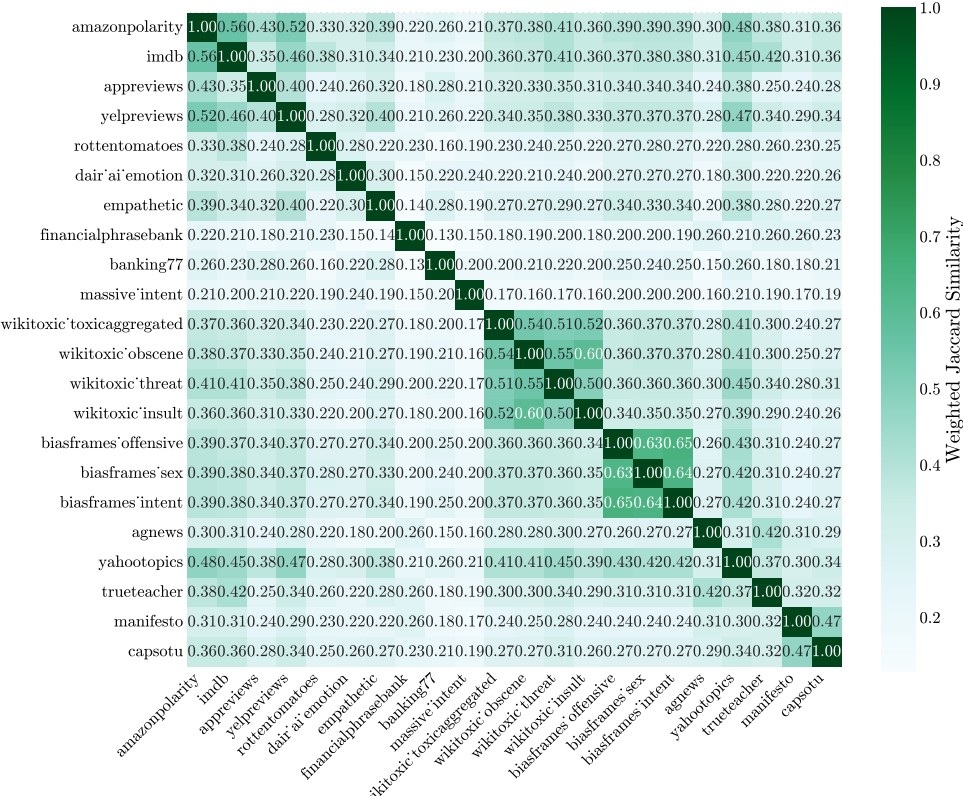

Figure 1: Pairwise weighted Jaccard similarity between datasets.

*NLI-RoBERTa-base* as well as a set of custom-trained cross-encoders using *BERT*, *DeBERTa-v3*, and *ModernBERT* backbones. Both base and large versions are evaluated to analyze the effect of model scale, and two loss variants are tested to assess the impact of training objectives. In total, 11 NLI-based cross-encoders are benchmarked, covering the most widely used configurations in the literature.

**Embedding Models.** This category comprises models optimized to produce fixed-size vector representations of text for a range of downstream tasks, including classification. As a canonical embedding model, *all-MiniLM-L6-v2* (Reimers & Gurevych, 2019) serves as a baseline for this model family. Additionally, we evaluate both base and large variants of BGE, GTE, and E5, all of which use variations of transformer encoders as backbones. To provide contrast, we also include embedding models that leverage large language model architectures, such as Qwen3-Embedding and e5-mistral-7b-instruct; for Qwen3-Embedding, both 0.6B and 8B parameter variants are tested to study the effect of scale. Overall, the embedding model category comprises 11 distinct models.

**Rerankers.** Reranker models are typically employed in information retrieval, where they re-score candidate documents for relevance to a given query. The *ms-marco-MiniLM-L6-v2* model serves as the reranker counterpart to *all-MiniLM-L6-v2* and is used as the baseline for this group. Similarly, *gte-reranker-modernbert-base* and *bge-reranker-base/large* serve as reranking counterparts to their respective embedding models. We further include *Qwen3-Reranker*, a generative reranker that scores document-query relevance by prompting the model to assess relevance. Both the 0.6B and 8B variants of *Qwen3-Reranker* are evaluated to analyze the impact of model size.

**Instruction-tuned LLMs.** This category comprises autoregressive language models that have undergone instruction fine-tuning to follow natural language directives. For zero-shot classification, we frame the task as a multiple-choice problem: the model is prompted with the input text and a list of candidate labels, and classification is performed by computing the conditional probability of each answer token given the prompt, selecting the answer with the highest probability. We evaluate models spanning a range of parameter scales to assess scaling behavior: *Gemma-3-270m-it*

and *Gemma-3-1b-it* (Mesnard et al., 2024) represent the smaller end of the spectrum, *Llama-3.2-3B-Instruct* (Grattafiori et al., 2024), *Phi-4-mini-instruct* (Abdin et al., 2024) and *Qwen3-4B* (Yang et al., 2025) provide mid-scale options, while *Qwen3-8B*, and *Mistral-Nemo-Instruct-2407* (Mistral AI, 2024) cover the larger parameter regime. We cap our evaluation at 12B parameters, as this study focuses on models suitable for low-latency, scalable deployment scenarios where computational efficiency remains a practical constraint.

Table 4 in Appendix B summarizes the models included in the experiments, listing their architecture, training data, and parameter count. In total, the benchmark covers 38 models.

## 4 EXPERIMENTAL SETUP

To facilitate zero-shot classification, each class label is verbalized as a short, semantically clear, and context-rich description. For example, in the Amazon Polarity dataset, the positive class is verbalized as "The overall sentiment within the Amazon product review is {label}," where "label" is substituted with either "positive" or "negative" depending on the ground truth.

For our custom NLI-based cross-encoders, we follow the methodology of Laurer et al. (2023) and train models on a mixture of MNLI (Williams et al., 2018), ANLI (Nie et al., 2020), WANLI (Liu et al., 2022), FEVERNLI (Thorne et al., 2018), and LingNLI (Parrish et al., 2021), datasets, deliberately omitting SNLI due to concerns regarding data quality and label bias. Full details of the training procedure are provided in section C.1 of the Appendix. During inference we collect the entailment logits and attribute the label with the highest logit as the predicted label. For embedding models, we compute the cosine similarity between the text embedding and each label embedding, selecting the label with the highest similarity score as the predicted label.[4] For reranker models, the text to be classified serves as the query, while the verbalized label descriptions are treated as candidate "documents" to be reranked according to their predicted relevance. For generative rerankers such as *Qwen3-Reranker*, the scoring mechanism is detailed in Appendix C.2. For instruction-tuned LLMs, we frame zero-shot classification as a multiple-choice task, prompting the model with the input text and enumerated verbalized label options, and selecting the option with the highest next-token probability (see Appendix C.3 for details).

## 5 RESULTS AND ANALYSIS

In this section, we present and analyze the performance of all evaluated models on the BTZSC benchmark. Table 2 summarizes results across all datasets, grouped by task type, and reports (macro) F1 scores averaged within each task as well as overall, in addition to average (micro) accuracy. Standard deviations are included in parentheses to reflect variability across datasets. Disaggregated results with precision and recall analysis are provided in Appendix D. To verify that our findings are not artifacts of BTZSC's specific dataset composition, we replicate the evaluation on the eight English classification tasks from MTEB v2; rankings are strongly correlated ($\tau = 0.69$, $p < 10^{-8}$) and family-wise conclusions remain consistent; for more granular details see Appendix E.

**Base Transformer Encoders.** Models that are not further fine-tuned or trained on specific semantic matching objectives perform poorly on zero-shot classification tasks. Their inability to align input texts with candidate label descriptions underscores the necessity of explicit training for semantic compatibility.

**NLI-based Cross-Encoders.** Models fine-tuned on NLI data exhibit clear benefits over their off-the-shelf counterparts. Training on a diverse set of NLI datasets, including MNLI, ANLI, WANLI, FEVERNLI, and LINGNLI, yields consistently stronger performance compared to models such as *bart-large-mnli* and *nli-roberta-base*, with multi-dataset models achieving an average improvement of +6 F1 points across all tasks. Scaling model size further enhances performance: large variants outperform their base counterparts by an average of +3.5 F1 points. Figure 2(a) highlights this difference on a more granular level. Task difficulty remains a dominant factor: sentiment classification is relatively easy (median F1 $\approx 0.88$–0.9), topic and intent classification are of intermediate difficulty (F1 $\approx 0.4$-0.55), and emotion detection proves most challenging (F1 $\approx 0.25$-0.35). Larger models

---

[4]For each embedding model family (E5, BGE, GTE, Qwen-Embedding), we follow the official instruction templates and query/document prefixes recommended in the original papers and Hugging Face model cards.

deliver the greatest benefit for more difficult tasks, with performance gains especially pronounced in topic and intent classification. The choice of loss function, whether binary cross-entropy with neutral collapsed or standard three-way cross-entropy, has minimal impact overall. The only systematic deviation appears in topic classification, where the triplet variant shows degradation in performance, though intent classification continues to improve under triplet training. Notably, within this family, *deberta-v3-large-nli-triplet* achieves the highest overall performance, surpassing both the original BERT and ModernBERT variants, corroborating findings from Warner et al. (2024) that *deberta-v3* is still a challenging baseline for various NLP tasks.

**Reranker Models.** Among rerankers, the baseline *ms-marco-MiniLM-L6-v2* does not match the performance of NLI cross-encoders (average F1: 0.42), consistent with the historical view that NLI fine-tuning is advantageous for zero-shot tasks. However, more recent rerankers close the gap substantially. For example, *gte-reranker-modernbert-base* achieves an average F1 of 0.58, just two points below the best NLI cross-encoder (*deberta-v3-large-nli-triplet*), and with lower variance. The strongest reranker, *Qwen3-Reranker-8B*, achieves an average F1 of 0.72 and outperforms all other models, including NLI cross-encoders, by significant margins (+12 F1 and +14 accuracy points). This model is the top overall performer on the benchmark, ranking first in two out of four task categories and second in topic classification. It should be noted, however, that its size (8B parameters) far exceeds that of NLI cross-encoders (typically around 300M parameters). Importantly, even the much smaller *Qwen3-Reranker-0.6B* delivers competitive results, surpassing all NLI cross-encoders in F1 and matching or exceeding their accuracy, underscoring the strength of the reranker approach even at moderate scale.

**Embedding Models.** The canonical embedding baseline, *all-MiniLM-L6-v2*, attains an average F1 of 0.37, supporting prior observations that rerankers generally outperform embedding models in retrieval, albeit at higher computational cost. However, newer embedding models such as *e5-large-v2*, *gte-modernbert-base*, and *gte-large-en-v1.5* achieve substantially higher F1 scores (0.60, 0.59, and 0.62, respectively), placing them on par with or even surpassing the best NLI cross-encoders. Notably, these embedding models lack cross-attention between documents and label verbalizers yet still deliver strong results at similar model sizes. For instance, *gte-large-en-v1.5* surpasses all NLI cross-encoders and all rerankers of comparable size, yet it still trails the top-performing *Qwen3-Reranker-8B* by roughly 10 F1 points. Scaling up embedding models does not yield the same improvements observed in rerankers; for example, *Qwen3-Embedding-8B* only marginally improves over its 0.6B variant (F1: 0.59 vs. 0.58).

**Instruction-tuned LLMs.** Instruction-tuned LLMs form a fourth family, evaluated in a strictly zero-shot setting with simple prompt templates. Their performance spans a wide range and reveals that both scale and model family play critical roles. Small models such as *gemma-3-270m-it* and *gemma-3-1b-it* perform comparably to base encoders (average F1 0.28 and 0.36, respectively), indicating that sub-billion parameter LLMs struggle as zero-shot classifiers. At the 3–4B parameter range, we observe substantial variation by model family: *Llama-3.2-3B-Instruct* and *Phi-4-mini-instruct* achieve only mid-0.40s F1, roughly matching weaker NLI cross-encoders and embedding baselines, whereas *Qwen3-4B* reaches 0.65 F1 despite comparable size. This performance gap suggests that at moderate scales, instruction-tuning quality and base model design matter more than parameter count alone. At 8B+ parameters, LLMs become consistently competitive: *Qwen3-8B* achieves 0.66 F1 with accuracy around 0.71. The strongest LLM, *Mistral-Nemo-Instruct-2407* (12B), attains 0.67 F1 and 0.71 accuracy, surpassing all embedding models and NLI cross-encoders, though it remains about 5 F1 points behind the specialized *Qwen3-Reranker-8B*. LLMs particularly excel on topic classification (F1 up to 0.69), while lagging somewhat on intent and emotion relative to the best rerankers and embedding models.

Figure 2(b) further elucidates scaling trends. All three families benefit from scaling, but with distinct regimes. Rerankers exhibit roughly monotonic gains with scale and form the top curve at all sizes, culminating in *Qwen3-Reranker-8B*. Embedding models improve rapidly up to a few hundred million parameters and then largely saturate around 0.60-0.62 F1. LLMs show the steepest scaling: performance rises slowly at sub-billion scales and then sharply between 3B and 8B, where they catch up with the best embeddings and approach the strongest reranker. Figure 3(a) plots model F1 score against normalized inference speed (inverse wall time) on a standard test set. The upper-right quadrant, bounded by the medians of both metrics, highlights models that best balance accuracy and efficiency. The majority of the models in this region are embedding models, indicating they offer the most favorable trade-off between performance and latency for practical deployments, with

*gte-reranker-modernbert-base* as the only reranker achieving comparable efficiency. Large LLMs, in contrast, tend to be accurate but slow: they cluster in the upper-left region of the plot, well outside the Pareto-efficient quadrant.

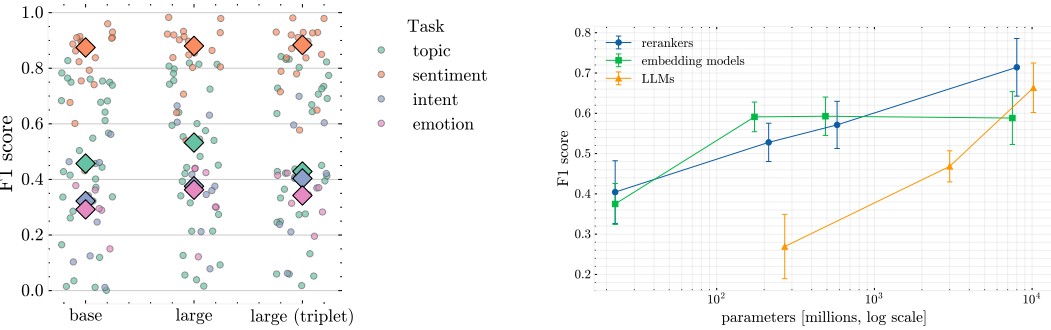

(a) NLI-based cross-encoders performance.      (b) Scaling across model sizes.

Figure 2: (a) Performance of NLI-based cross-encoders on BTZSC; points are individual datasets and diamonds mark task-wise medians, comparing model size (base vs. large) and loss type (binary vs. three-way). (b) Effect of scale on zero-shot performance: macro-$F_1$ vs. parameter count (log scale); bands show 95% CIs.

## 5.1 NLI PERFORMANCE AS A PROXY FOR ZERO-SHOT CLASSIFICATION

We also examine whether NLI task performance predicts zero-shot classification effectiveness. As shown in Figure 3(b), the relationship is strongly model-family dependent. For NLI-tuned cross-encoders, there is a clear, almost linear relationship: higher NLI AUROC consistently translates into higher F1 on BTZSC, reflecting the direct transfer of entailment supervision to zero-shot classification. Large LLMs follow a similar pattern, with models that perform better on NLI also achieving stronger BTZSC results, indicating that entailment-aligned reasoning capabilities in instruction-tuned LLMs remain predictive of ZSC quality. Rerankers, although not explicitly fine-tuned on NLI, still display a positive trend: better NLI AUROC is generally associated with higher BTZSC F1. At the same time, several rerankers attain strong classification performance despite only moderate NLI scores, suggesting that relevance-focused training captures discriminative task signals that standard NLI benchmarks do not fully reflect. Embedding models, by contrast, show tightly clustered NLI AUROC values but a wide spread in BTZSC F1. This lack of a clear monotonic relationship implies that once a basic level of NLI competence is reached, NLI performance is no longer a good proxy for zero-shot classification quality in embedding models. Instead, the structure of the embedding space and its ability to encode fine-grained topical distinctions are hypothesized to be the dominant factors.

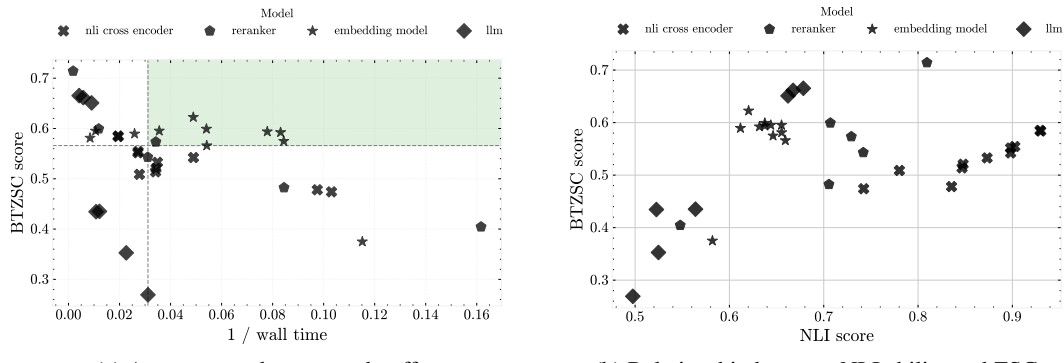

(a) Accuracy vs. latency trade-off.      (b) Relationship between NLI ability and ZSC.

Figure 3: (a) Performance–speed trade-off: macro-$F_1$ on BTZSC vs. normalized inference throughput (1/wall time) on a standard test set; the upper-right quadrant (split by metric medians) marks models with the best accuracy–efficiency balance. (b) NLI ability vs. zero-shot classification: AUROC on standard NLI benchmarks (x-axis) vs. BTZSC macro-$F_1$ (y-axis).

| Model | Topic | Sentiment | Intent | Emotion | Avg F1 | Avg Acc |
|---|---|---|---|---|---|---|
| **Base encoders** | | | | | | |
| bert-large-uncased | 0.34 (0.22) | 0.38 (0.06) | 0.15 (0.24) | 0.08 (0.11) | 0.30 (0.20) | 0.40 (0.26) |
| deberta-v3-large | 0.30 (0.23) | 0.34 (0.03) | 0.16 (0.26) | 0.05 (0.07) | 0.27 (0.20) | 0.36 (0.25) |
| ModernBERT-large | 0.30 (0.24) | 0.37 (0.06) | 0.14 (0.21) | 0.03 (0.04) | 0.27 (0.21) | 0.35 (0.24) |
| **NLI cross-encoders** | | | | | | |
| bart-large-mnli | 0.37 (0.23) | 0.84 (0.19) | 0.43 (0.16) | 0.41 (0.04) | 0.51 (0.28) | 0.53 (0.28) |
| nli-roberta-base | 0.40 (0.25) | 0.80 (0.15) | 0.30 (0.22) | 0.33 (0.02) | 0.49 (0.28) | 0.50 (0.28) |
| bert-base-uncased-nli | 0.43 (0.27) | 0.76 (0.17) | 0.30 (0.28) | 0.26 (0.15) | 0.49 (0.29) | 0.50 (0.28) |
| bert-large-uncased-nli | 0.49 (0.27) | 0.79 (0.10) | 0.35 (0.26) | 0.27 (0.21) | 0.53 (0.28) | 0.57 (0.28) |
| bert-large-uncased-nli-triplet | 0.49 (0.27) | 0.78 (0.12) | 0.35 (0.27) | 0.24 (0.06) | 0.52 (0.28) | 0.55 (0.26) |
| deberta-v3-base-nli | 0.49 (0.26) | 0.86 (0.10) | 0.31 (0.17) | 0.33 (0.06) | 0.55 (0.28) | 0.58 (0.26) |
| deberta-v3-large-nli | 0.48 (0.26) | 0.90 (0.06) | 0.48 (0.17) | 0.44 (0.00) | 0.59 (0.27) | 0.62 (0.25) |
| deberta-v3-large-nli-triplet | 0.50 (0.28) | 0.90 (0.07) | 0.45 (0.23) | 0.42 (0.01) | 0.60 (0.28) | 0.62 (0.27) |
| modernbert-base-nli | 0.49 (0.26) | 0.84 (0.14) | 0.27 (0.18) | 0.29 (0.01) | 0.53 (0.29) | 0.56 (0.29) |
| modernbert-large-nli | 0.47 (0.25) | 0.86 (0.16) | 0.40 (0.21) | 0.30 (0.00) | 0.55 (0.28) | 0.59 (0.27) |
| modernbert-large-nli-triplet | 0.45 (0.26) | 0.88 (0.12) | 0.41 (0.19) | 0.34 (0.04) | 0.55 (0.29) | 0.58 (0.27) |
| **Rerankers** | | | | | | |
| ms-marco-MiniLM-L6-v2 | 0.41 (0.14) | 0.59 (0.16) | 0.30 (0.29) | 0.19 (0.01) | 0.42 (0.20) | 0.46 (0.21) |
| gte-reranker-modernbert-base | 0.49 (0.14) | 0.82 (0.17) | 0.51 (0.15) | 0.42 (0.04) | 0.58 (0.20) | 0.62 (0.19) |
| bge-reranker-base | 0.42 (0.14) | 0.62 (0.15) | 0.47 (0.04) | 0.29 (0.00) | 0.47 (0.16) | 0.49 (0.14) |
| bge-reranker-large | 0.43 (0.19) | 0.78 (0.15) | 0.54 (0.07) | 0.37 (0.03) | 0.53 (0.22) | 0.56 (0.21) |
| Qwen3-Reranker-0.6B | 0.54 (0.24) | 0.80 (0.20) | 0.55 (0.07) | 0.45 (0.06) | 0.61 (0.23) | 0.64 (0.21) |
| Qwen3-Reranker-8B | 0.66 (0.19) | **0.92 (0.06)** | **0.70 (0.03)** | 0.49 (0.00) | **0.72 (0.19)** | **0.76 (0.15)** |
| **Embedding models** | | | | | | |
| all-MiniLM-L6-v2 | 0.41 (0.12) | 0.35 (0.04) | 0.41 (0.07) | 0.13 (0.03) | 0.37 (0.12) | 0.44 (0.14) |
| e5-base-v2 | 0.51 (0.20) | 0.83 (0.19) | 0.56 (0.08) | 0.40 (0.04) | 0.60 (0.23) | 0.62 (0.21) |
| e5-large-v2 | 0.50 (0.17) | 0.86 (0.17) | 0.55 (0.04) | 0.41 (0.04) | 0.60 (0.22) | 0.62 (0.20) |
| e5-mistral-7b-instruct | 0.41 (0.21) | 0.87 (0.13) | 0.65 (0.02) | 0.50 (0.00) | 0.58 (0.26) | 0.62 (0.24) |
| bge-base-en-v1.5 | 0.47 (0.20) | 0.82 (0.20) | 0.58 (0.05) | 0.36 (0.09) | 0.57 (0.24) | 0.59 (0.22) |
| bge-large-en-v1.5 | 0.42 (0.19) | 0.84 (0.19) | 0.58 (0.09) | 0.39 (0.06) | 0.55 (0.25) | 0.59 (0.24) |
| gte-base-en-v1.5 | 0.49 (0.23) | 0.83 (0.18) | 0.59 (0.07) | 0.37 (0.07) | 0.58 (0.24) | 0.61 (0.23) |
| gte-large-en-v1.5 | 0.54 (0.22) | 0.85 (0.18) | 0.59 (0.04) | 0.37 (0.04) | 0.62 (0.23) | 0.64 (0.22) |
| gte-modernbert-base | 0.45 (0.20) | 0.87 (0.14) | 0.62 (0.01) | 0.42 (0.04) | 0.59 (0.24) | 0.61 (0.23) |
| Qwen3-Embedding-0.6B | 0.49 (0.14) | 0.81 (0.17) | 0.56 (0.10) | 0.43 (0.07) | 0.58 (0.20) | 0.61 (0.18) |
| Qwen3-Embedding-8B | 0.44 (0.16) | 0.89 (0.09) | 0.59 (0.17) | **0.51 (0.05)** | 0.59 (0.23) | 0.64 (0.20) |
| **Instruction-tuned LLMs** | | | | | | |
| gemma-3-270m-it | 0.29 (0.21) | 0.42 (0.11) | 0.13 (0.22) | 0.04 (0.04) | 0.28 (0.21) | 0.31 (0.21) |
| gemma-3-1b-it | 0.34 (0.18) | 0.52 (0.14) | 0.24 (0.25) | 0.14 (0.02) | 0.36 (0.20) | 0.40 (0.21) |
| Llama-3.2-3B-Instruct | 0.44 (0.16) | 0.46 (0.09) | 0.41 (0.05) | 0.35 (0.03) | 0.43 (0.12) | 0.46 (0.11) |
| Qwen3-4B | 0.64 (0.23) | 0.88 (0.11) | 0.40 (0.04) | 0.37 (0.08) | 0.65 (0.25) | 0.70 (0.22) |
| Phi-4-mini-instruct | 0.44 (0.15) | 0.49 (0.13) | 0.37 (0.07) | 0.30 (0.01) | 0.43 (0.14) | 0.47 (0.14) |
| Qwen3-8B | 0.65 (0.23) | 0.90 (0.08) | 0.48 (0.17) | 0.32 (0.11) | 0.66 (0.25) | 0.71 (0.23) |
| Mistral-Nemo-Instruct-2407 | **0.69 (0.24)** | 0.84 (0.17) | 0.46 (0.17) | 0.36 (0.10) | 0.67 (0.25) | 0.71 (0.23) |

Table 2: Zero-shot classification results on BTZSC. We report macro F1 per task family, overall macro F1 (Avg F1), and micro accuracy (Avg Acc). Parentheses report the across-dataset standard deviation within each task family reflecting task heterogeneity. Bold indicates the best and underlining the second-best score per column; the best model within each family (based on overall macro F1) is also underlined.

## 6 CONCLUSION AND FUTURE WORK

This paper introduces BTZSC, a unified benchmark for zero-shot text classification that jointly evaluates NLI cross-encoders, embedding models, rerankers, and instruction-tuned LLMs. Across 22 datasets, modern rerankers achieve the best overall performance, while strong embedding models offer the most attractive accuracy-latency trade-off; NLI cross-encoders remain competitive but show diminishing returns with scale, and NLI scores predict zero-shot quality only within this family and for LLMs, not for embeddings. Instruction-tuned LLMs at 4–12B parameters form a fourth regime: clearly better than base encoders and small LLMs and competitive with strong embeddings and cross-encoders, yet still lagging behind the best rerankers at substantially higher cost. BTZSC, together with our released code and models, provides a reproducible testbed for future work on multilingual extensions, improved label verbalizations and prompts, and scaling up instruction-tuned and reranker models for realistic zero-shot deployment.

## REPRODUCIBILITY STATEMENT

To foster transparency and enable future work, we release the complete evaluation benchmark, including preprocessing and evaluation scripts, with the full codebase under an MIT license at `https://github.com/IliasAarab/btzsc`. The BTZSC datasets are available at `https://huggingface.co/datasets/btzsc/btzsc`, trained model checkpoints at `https://github.com/IliasAarab/btzsc`, and a live leaderboard at `https://huggingface.co/spaces/btzsc/btzsc-leaderboard`. All training configurations (optimizer, learning rate schedules, batch sizes, and early stopping criteria) are documented in the appendix. For robustness, training results are averaged over three independent runs with different random seeds. Evaluation was performed on a compute cluster with NVIDIA A100 80GB GPUs, using parallelized execution across models. All inference runs were carried out in `bfloat16` precision, and training, where applicable, employed mixed precision for efficiency. Pretrained checkpoints sourced from public libraries are properly cited and referenced. To our knowledge, no restrictions prevent full reproducibility of the results presented in this work.

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

## A  DATASETS OVERVIEW

BTZSC comprises 22 English single-label classification datasets spanning topic, sentiment, intent, and emotion, as described in Section 3 and Table 1. Each dataset is treated as a *separate, single-label task*: we never merge examples or label spaces across datasets, and all metrics are computed per dataset before aggregation.

All datasets are publicly available through Hugging Face Datasets[5] and listed in Table 3 together with their original sources and licenses. BTZSC reuses a subset of the dataset pool compiled by Laurer et al. (2023) for transfer learning across a broad range of domains and task types, selecting only datasets we deem sufficiently high-quality and well-suited to zero-shot text classification. Where applicable, we adopt their task definitions and label verbalizers.

### A.1  INSTANCE FORMAT AND SPLITS

For every dataset $D$, we standardise the Hugging Face representation to a simple pair

$$(x_i, y_i) \in \mathcal{X} \times \{0, \ldots, L_D - 1\},$$

where $x_i$ is a single input text string and $y_i$ is a categorical label index.

**Single text field.** Each original dataset may expose one or more textual fields (e.g. `text`, `sentence`, `utterance`, `comment_text`). We map these to a single canonical text field as follows:

- If there is a single obvious document field (e.g. `text`, `review`, `comment_text`), we use that field directly as $x_i$.
- If relevant information is split across multiple short fields (e.g. title + body, question + answer, or conversational context), we concatenate them in a fixed order with newline separators to form one string $x_i$. No dataset-specific prompts or instructions are injected.

This standardisation is performed once per dataset and then reused across all model families, so every model sees exactly the same input text for a given example.

**Single categorical label.** Each example carries a single gold label. We map the dataset-specific label field (e.g. `label`, `sentiment`, `topic`, `intent`, `emotion`) to an integer index $y_i \in \{0, \ldots, L_D - 1\}$, where $L_D$ is the number of classes in dataset $D$. The mapping between original label names and indices is fixed per dataset and shared across all models. Multi-label datasets are not included in BTZSC.

**Splits and zero-shot protocol.** For each dataset we evaluate *purely zero-shot* on the designated evaluation split. Whenever the Hugging Face dataset exposes an official `test` split, we use that split as BTZSC's test set. For datasets without a separate `test` split, we adopt the `dev` or `validation` split as test. No remixing or re-partitioning of examples is performed.

No labeled examples from any BTZSC dataset are used for training, hyper-parameter tuning, or model selection. All reported scores are computed on these test splits under the zero-shot protocol defined in the main text: models are only allowed to use their pretrained parameters and generic evaluation prompts/verbalizers, with no supervision from BTZSC labels.

---

[5] https://huggingface.co/datasets

## A.2 LABEL VERBALIZERS

Zero-shot classification is implemented via natural-language label descriptions. Following Laurer et al. (2023), each dataset $D$ is associated with:

- a set of class names (e.g. *positive*, *negative*, *neutral*, *toxic*, *non-toxic*), and

- a short, semantically informative verbalizer template describing how the label appears in context.

Concretely, for each dataset we define a template such as

*"The overall sentiment within the Amazon product review is {label}"*

where {label} is replaced by the class name (e.g. *positive* or *negative*). The exact templates and class names are inherited from the release of Laurer et al. (2023).

These verbalizers are shared across all model families:

- For NLI cross-encoders, the text $x_i$ acts as *premise* and each verbalized label as *hypothesis*.

- For embedding models, both $x_i$ and verbalized labels are encoded and compared via cosine similarity.

- For rerankers, $x_i$ is the query and verbalized labels are candidate "documents".

- For LLMs, verbalized labels appear as options in a multiple-choice prompt (Appendix C.3).

Crucially, we do *not* tune verbalizers per model or per dataset: the same set of label descriptions is used for all runs to ensure strict comparability.

## A.3 TOKENISATION AND TRUNCATION

All models use their own official tokenizer. For each batch, we determine a maximum sequence length

$$L_{\text{batch}} = \min\left(L_{\text{model}}, \max_i |x_i|\right),$$

where $L_{\text{model}}$ is the maximum context length supported by the model and $|x_i|$ is the tokenised length of input $x_i$ in that batch.

Inputs longer than $L_{\text{model}}$ are truncated at the model's hard limit; otherwise no manual truncation is applied. We use dynamic padding to the longest sequence in the batch and do not apply any additional dataset-specific preprocessing.

## A.4 EVALUATION AND AGGREGATION PROTOCOL

Evaluation proceeds in two stages.

**Per-dataset evaluation.** For every dataset $D$ we compute:

- macro-averaged $F_1$ over the $L_D$ classes,

- micro-averaged accuracy,

- macro-averaged recall, and

- macro-averaged precision

on the full test split of $D$. No labels or examples are shared across datasets, and there is no aggregation of label spaces.

**Aggregation across datasets.** To obtain the summary scores reported in Table 2, we only aggregate *dataset-level* metrics. For each task family $F \in \{\text{topic}, \text{sentiment}, \text{intent}, \text{emotion}\}$, we compute the family-wise macro-$F_1$ as the *unweighted mean* of macro-$F_1$ over all datasets $D \in F$. The same unweighted averaging is used for family-wise accuracy. Each dataset therefore contributes equally, independent of its size or class cardinality. We never pool examples across datasets when computing these aggregates: there is no global micro-averaging over all test instances. Instead, BTZSC deliberately treats each dataset as an independent testbed and uses simple unweighted averages to summarise performance across this collection.

## A.5 SOURCES AND LICENSES

Table 3 lists the original sources and licenses for all datasets used in BTZSC.

| Domain | Dataset | Source | License |
|--------|---------|--------|---------|
| **Emotion** | | | |
| dialogue | empathetic_dialogues | (Rashkin et al., 2019) | CC BY-NC 4.0 |
| social-media | dair_ai_emotion | (Saravia et al., 2018) | Research/education only |
| **Intent** | | | |
| banking | banking77 | (Casanueva et al., 2020) | CC BY 4.0 |
| social-media | biasframes_intent | (Sap et al., 2020) | CC BY 4.0 |
| assistant | massive_intent | (FitzGerald et al., 2022) | CC BY 4.0 |
| **Sentiment** | | | |
| apps | appreviews | (Grano et al., 2017) | Unknown |
| e-commerce | amazonpolarity | (Zhang et al., 2015a) | Apache-2.0 |
| finance | financialphrasebank | (Malo et al., 2014) | CC BY-NC-SA 3.0 |
| local-business | yelpreviews | (Zhang et al., 2015c) | ToU (non-commercial) |
| movies | imdb | (Maas et al., 2011) | IMDb Non-Commercial Terms |
| movies | rottentomatoes | (Pang & Lee, 2005) | CC0 1.0 |
| **Topic** | | | |
| education | trueteacher | (Gekhman et al., 2023) | CC BY-NC 4.0 |
| news | agnews | (Zhang et al., 2015a) | Non-commercial |
| politics | capsotu | (Jones et al., 2023; Laurer et al., 2023) | CC BY-NC-SA 4.0 |
| politics | manifesto | (Lehmann et al., 2024) | Terms of Use |
| qa-forum | yahootopics | (Zhang et al., 2015b) | Unknown |
| social-media | biasframes_offensive | (Sap et al., 2020) | CC BY 4.0 |
| social-media | biasframes_sex | (Sap et al., 2020) | CC BY 4.0 |
| wikipedia | wikitoxic_insult | (Wulczyn et al., 2017) | CC0 1.0 |
| wikipedia | wikitoxic_obscene | (Wulczyn et al., 2017) | CC0 1.0 |
| wikipedia | wikitoxic_threat | (Wulczyn et al., 2017) | CC0 1.0 |
| wikipedia | wikitoxic_toxicaggregated | (Wulczyn et al., 2017) | CC0 1.0 |

Table 3: Sources and licenses for BTZSC datasets. All datasets are used as single-label classification tasks and evaluated in a purely zero-shot setting on their respective test splits (Section A.1).

### A.5.1 DATASET DESCRIPTIONS

Below we provide brief descriptions of each dataset included in BTZSC, outlining the source domain, annotation scheme, and classification task.

**`dialogue_empathetic`.** EmpatheticDialogues is a corpus of multi-turn conversations in which one speaker describes a personal situation and the other responds in an empathetic way. Each dialogue turn is associated with one of several emotion categories (e.g., joy, sadness, fear), so the dataset can be used for emotion or topic-style classification of conversational text.

**`dair_ai_emotion`.** The `dair_ai_emotion` dataset corresponds to the `dair-ai/emotion` corpus, a collection of short English texts (originally social media posts) annotated with discrete emotion labels. Each instance is labeled with one of six basic emotions (anger, fear, joy, love, sadness, or surprise), making it a standard benchmark for single-label emotion classification in English.

**banking77.** Banking77 is a dataset of short customer queries to an online banking assistant (e.g., "I want to freeze my card"), each labeled with one of 77 fine-grained intent classes such as card issues, transfers, or account information. It is designed for intent classification in the banking and financial customer-support domain.

**biasframes_intent.** These splits are derived from the Social Bias Frames dataset, which contains English sentences about social groups (often from online platforms) annotated with rich labels describing the speaker's intent and implied meaning (e.g., whether they intend to offend, to joke, to express hatred, etc.). The intent view focuses on predicting those communicative-intent labels from the text alone.

**appreviews.** The app_reviews dataset consists of user reviews from mobile app stores (e.g., Google Play / Apple App Store), where each review text is paired with a sentiment-style label that reflects the user's overall evaluation of the app (for example negative vs. positive, or star-based ratings). It is used as a product-review sentiment classification benchmark.

**amazonpolarity.** Amazon Polarity is a large-scale sentiment dataset built from Amazon product reviews. Each example is a review text labeled as either positive or negative, based on the original star rating, and spans a wide range of product categories (books, electronics, etc.).

**financialphrasebank.** Financial PhraseBank contains short English snippets from financial news and company press releases, each labeled according to the sentiment of the text with respect to the target company's future performance (positive, neutral, or negative). It is a standard benchmark for fine-grained sentiment analysis in finance.

**yelpreviews.** The Yelp review datasets are collections of user-written reviews of local businesses (restaurants, shops, services) posted on Yelp. Depending on the variant, each review is labeled either with a binary polarity (positive vs. negative) or with one of several star-based rating categories, making it a benchmark for review sentiment classification.

**imdb.** The IMDB sentiment dataset consists of movie reviews from the Internet Movie Database, each labeled as positive or negative based on the overall opinion expressed. Reviews are relatively long and varied in style, so the dataset is often used to test document-level sentiment classification.

**rottentomatoes.** The Rotten Tomatoes movie review dataset contains short snippets of film reviews taken from the Rotten Tomatoes website, each annotated with a binary positive/negative sentiment label (in some variants, a finer-grained rating). It is a classic benchmark for sentence-level sentiment classification.

**massive.** MASSIVE ("Multilingual Amazon SLU Simulation for Slot filling, Intent classification, and Virtual assistant Evaluation") is a dataset of crowdsourced virtual-assistant utterances spanning many languages. Each utterance is labeled with an intent class and slot annotations; in this benchmark it is used for intent classification on assistant-style queries.

**trueteacher.** The TRUETEACHER dataset is a large collection of sentence pairs constructed to study and improve factual consistency in summarization. It is built from news summarization corpora where a powerful language model (FLAN-PaLM 540B) is used to generate candidate summaries and then label them as factually consistent or inconsistent with their source articles. The resulting pairs support training and evaluating models that distinguish faithful summaries from hallucinated or unsupported content.

**agnews.** AG News is a topic-classification dataset of news headlines and short descriptions collected from the AG's News corpus. Each article is categorized into one of four high-level topics: World, Sports, Business, or Sci/Tech.

**capsotu.** The CAPSOTU dataset is derived from the Comparative Agendas Project's State of the Union (SOTU) data.[6] The SOTU corpus breaks U.S. presidential State of the Union speeches into short quasi-sentences and codes each segment with detailed policy content categories.

**manifesto.** The Manifesto dataset is derived from political party election manifestos compiled by the Comparative Manifesto Project. Text segments (often sentences or quasi-sentences) are annotated with policy-topic categories such as economic policy, welfare, or foreign relations, enabling multi-class classification of political positions.

**yahootopics.** The Yahoo Topics (Yahoo! Answers) dataset contains questions and accompanying text from the Yahoo! Answers platform, each assigned to one of several broad topical categories (e.g., Society & Culture, Science & Mathematics, Sports, Business & Finance). It is used as a multi-class topic classification benchmark for user-generated Q&A text.

**biasframes_offensive.** This split of the Social Bias Frames data focuses on labels describing whether an utterance is offensive or not, and to what degree. The underlying texts are short statements about social groups, annotated for perceived offensiveness, so the task is to classify language according to its offensive content.

**biasframes_sex.** This variant of the Social Bias Frames dataset isolates labels related to sexism or gender-based bias. The texts are again statements about people or groups, annotated for whether they convey sexist stereotypes or implications, turning the task into detecting gender-related bias in language.

**wikitoxic_insult.** These splits are based on the Wikipedia Talk Labels toxicity datasets released in the Jigsaw toxicity challenges. The insult subset contains comments from Wikipedia talk pages, each annotated for whether it includes insulting language, and is used as a binary classification task for the presence of insults.

**wikitoxic_obscene.** The obscene split from the same Wikipedia toxicity data contains comments annotated for obscene or vulgar language. The classification task is to determine whether a given user comment uses obscene expressions or not.

**wikitoxic_threat.** The threat split consists of Wikipedia talk-page comments labeled according to whether they contain threats of violence or other threatening language. It is used to train and evaluate models on detecting threatening content in online discussions.

**wikitoxic_toxicaggregated.** The toxicaggregated variant combines several toxicity-related labels (e.g., toxic, severe toxic, obscene, insult, threat, identity hate) from the Wikipedia toxicity datasets into a single binary "toxic vs. non-toxic" label. This yields a broader notion of toxicity for classifying harmful comments in online conversations.

## B  MODELS OVERVIEW

Table 4 provides a comprehensive overview of all 38 models evaluated in this study, including their architecture, backbone, fine-tuning data, parameter count, and pooling strategy.

---

[6]See https://www.comparativeagendas.net/datasets_codebooks.

| Model | Yr | Arch. | Backbone | FT / train data | # P | Pool / dim |
|---|---|---|---|---|---|---|
| **Base encoders** | | | | | | |
| bert-large-uncased | 2018 | enc. | BERT | NA | 340M | - |
| deberta-v3-large | 2021 | enc. | DeBERTa v3 | NA | 304M | - |
| ModernBERT-large | 2024 | enc. | ModernBERT | NA | 395M | - |
| **NLI cross-encoders** | | | | | | |
| bart-large-mnli | 2020 | enc-dec. | BART | SNLI, MNLI | 406M | - |
| nli-roberta-base | 2020 | enc. | RoBERTa | SNLI, MNLI | 125M | - |
| bert-base-uncased-nli | — | enc. | BERT | MNLI, ANLI, WANLI, FEVERNLI, LINGNLI | 110M | - |
| bert-large-uncased-nli | — | enc. | BERT | same as above | 340M | - |
| bert-large-uncased-nli-triplet | — | enc. | BERT | same as above | 340M | - |
| deberta-v3-base-nli | — | enc. | DeBERTa v3 | same as above | 184M | - |
| deberta-v3-large-nli | — | enc. | DeBERTa v3 | same as above | 304M | - |
| deberta-v3-large-nli-triplet | — | enc. | DeBERTa v3 | same as above | 304M | - |
| modernbert-base-nli | — | enc. | ModernBERT | same as above | 149M | - |
| modernbert-large-nli | — | enc. | ModernBERT | same as above | 395M | - |
| modernbert-large-nli-triplet | — | enc. | ModernBERT | same as above | 395M | - |
| **Rerankers** | | | | | | |
| ms-marco-MiniLM-L6-v2 | 2021 | enc. | MiniLM | MS MARCO | 22.7M | - |
| gte-reranker-modernbert-base | 2024 | enc. | ModernBERT | large multiling. pairs | 149M | - |
| bge-reranker-base | 2023 | enc. | XLM-RoBERTa base | large multiling. pairs | 278M | - |
| bge-reranker-large | 2023 | enc. | XLM-RoBERTa large | large multiling. pairs | 560M | - |
| Qwen3-Reranker-0.6B | 2025 | dec. | Qwen3 | synthetic yes/no ranking | 0.6B | - |
| Qwen3-Reranker-8B | 2025 | dec. | Qwen3 | synthetic yes/no ranking | 8B | - |
| **Embedding models** | | | | | | |
| all-MiniLM-L6-v2 | 2021 | enc. | MiniLM | 1B paired sentences | 22.7M | mean / 384 |
| e5-base-v2 | 2023 | enc. | E5 (BERT) | 270M synthetic contrastive | 110M | mean / 768 |
| e5-large-v2 | 2023 | enc. | E5 (BERT) | same as above | 335M | mean / 1024 |
| e5-mistral-7b-instruct | 2024 | dec. | Mistral-7B | synthetic multiling. contrastive | 7B | last / 4096 |
| bge-base-en-v1.5 | 2023 | enc. | BGE (RoB.) | 1.5B pair data, contrastive | 137M | CLS / 768 |
| bge-large-en-v1.5 | 2023 | enc. | BGE (RoB.) | same as above | 434M | CLS / 1024 |
| gte-base-en-v1.5 | 2024 | enc.+ | GTE | MLM + contrastive pretrain | 137M | CLS / 768 |
| gte-large-en-v1.5 | 2024 | enc.+ | GTE | same as above | 434M | CLS / 1024 |
| gte-modernbert-base | 2024 | enc. | ModernBERT | same as above | 149M | CLS / 768 |
| Qwen3-Embedding-0.6B | 2025 | dec. | Qwen3 | synthetic multiling. contrastive | 0.6B | last / 1024 |
| Qwen3-Embedding-8B | 2025 | dec. | Qwen3 | synthetic multiling. contrastive | 8B | last / 4096 |
| **LLMs** | | | | | | |
| gemma-3-270m-it | 2025 | dec. | Gemma 3 | NA | 270M | - |
| gemma-3-1b-it | 2025 | dec. | Gemma 3 | NA | 1B | - |
| Llama-3.2-3B-Instruct | 2024 | dec. | Llama 3.2 | NA | 3.21B | - |
| Qwen3-4B | 2025 | dec. | Qwen3 | NA | 4.0B | - |
| Phi-4-mini-instruct | 2025 | dec. | Phi-4 | NA | 3.8B | - |
| Qwen3-8B | 2025 | dec. | Qwen3 | NA | 8.2B | - |
| Mistral-Nemo-Instruct-2407 | 2024 | dec. | Mistral-Nemo | NA | 12.2B | - |

Table 4: Architectural and training overview of the 38 models evaluated. Columns list publication year (Yr), encoder/decoder architecture (Arch.), backbone, principal fine-tuning data, parameter count (#P), and pooling strategy with embedding dimensionality.

## C EXPERIMENTAL SETUP

### C.1 TRAINING PROCEDURE FOR NLI CROSS-ENCODERS

Let a paired input sequence (premise $\|$ hypothesis) be tokenised as $\mathbf{x} = (x_0 = [\text{CLS}], x_1, \ldots, [\text{SEP}], \ldots, x_{S-1})$ and encoded by a pretrained Transformer backbone $f_\theta : \mathbb{N}^S \to \mathbb{R}^{S \times E}$ with hidden size $E$:

$$H = f_\theta(\mathbf{x}) \in \mathbb{R}^{S \times E}, \qquad h = H_0 \in \mathbb{R}^E \quad (\text{CLS row}).$$

A two-layer classification head with dropout $p = 0.1$ transforms $h$:

$$\tilde{h} = \text{Dropout}_{0.1}(h), \tag{1}$$

$$u = \text{GELU}\big(W_1 \tilde{h} + b_1\big), \quad W_1 \in \mathbb{R}^{E \times E}, \ b_1 \in \mathbb{R}^E, \tag{2}$$

$$z = \text{LayerNorm}(u), \tag{3}$$

$$\ell = W_2 z + b_2, \quad W_2 \in \mathbb{R}^{E \times C}, \ b_2 \in \mathbb{R}^C, \tag{4}$$

where $C$ is the number of label logits.

**Binary variant.** Here $C = 1$ and $\ell \in \mathbb{R}$ is an *entailment logit*. The probability of entailment is $\sigma(\ell) = \left(1 + e^{-\ell}\right)^{-1}$ and the model is optimised with binary cross-entropy:

$$\mathcal{L}_{\mathrm{BCE}}(y, \ell) = -y \log \sigma(\ell) - (1 - y) \log\left(1 - \sigma(\ell)\right), \quad y \in \{0, 1\}.$$

**Three-way variant (*triplet*).** Now $C = 3$ with logits $\ell = (\ell_{\mathrm{ent}}, \ell_{\mathrm{neut}}, \ell_{\mathrm{contra}})$. During *training* the standard multi-class cross-entropy is used:

$$\mathcal{L}_{\mathrm{CE}}(y, \ell) = -\log \frac{\exp(\ell_y)}{\sum_{c=1}^{3} \exp(\ell_c)}, \quad y \in \{1, 2, 3\}.$$

During *evaluation* the scalar entailment score is

$$s = \ell_{\mathrm{ent}} - \log\left(e^{\ell_{\mathrm{neut}}} + e^{\ell_{\mathrm{contra}}}\right),$$

which is the log-odds of ENTAILMENT versus the union of the other classes (with probability $\sigma(s)$).

**Validation signal.** Early stopping is triggered by the dev-set loss computed on an equal-sized, balanced union

$$\mathcal{D}_{\mathrm{dev}} = \mathrm{MNLI}_{\mathrm{m}} \cup \mathrm{MNLI}_{\mathrm{mm}} \cup \mathrm{ANLI}_{r1} \cup \mathrm{ANLI}_{r2} \cup \mathrm{ANLI}_{r3} \cup \mathrm{WANLI} \cup \mathrm{FEVERNLI} \cup \mathrm{LINGNLI}.$$

At every evaluation step the loss is measured, and training stops when this loss fails to decrease for 10 consecutive evaluations, or 3 epochs, whichever comes first. Evaluation is performed every $1\%$ of total steps.

**Optimiser and schedules.** Fine-tuning uses the PyTorch `AdamW` (Loshchilov & Hutter, 2019) optimiser with default settings ($\beta_1 = 0.9$, $\beta_2 = 0.999$, $\varepsilon = 10^{-8}$, weight-decay $= 0.01$). The learning rate employs a *linear warm-up* for the first $10\%$ of steps followed by *cosine decay*:

$$\eta_t = \begin{cases} \eta_0 \dfrac{t}{0.1T}, & 0 \leq t < 0.1T, \\[2ex] \frac{1}{2}\eta_0 \left(1 + \cos\dfrac{\pi(t - 0.1T)}{0.9T}\right), & 0.1T \leq t \leq T, \end{cases}$$

with separate initial rates for the backbone ($\eta_{\mathrm{enc}}$) and classification head ($\eta_{\mathrm{head}}$).

- **Large backbones:** $\eta_{\mathrm{enc}} = 8 \times 10^{-6}$, $\eta_{\mathrm{head}} = 4 \times 10^{-5}$.
- **Base backbones:** $\eta_{\mathrm{enc}} = 2 \times 10^{-5}$, $\eta_{\mathrm{head}} = 1 \times 10^{-4}$.

All models train for $E = 3$ epochs with mini-batch size $B = 32$ and no layer freezing.

## C.2 QWEN3 RERANKER

For every *query-document* pair we build a single decoder-only prompt of the form

$$P = \texttt{prefix} + \langle\text{Instruct}\rangle\colon I + \langle\text{Query}\rangle\colon q + \langle\text{Document}\rangle\colon d + \texttt{suffix}.$$

**Fixed strings.**

*Prefix*

```
<|im_start|>system
Judge whether the Document meets the requirements based on the Query
and the Instruct provided. Note that the answer can only be "yes" or "no".
<|im_end|>
<|im_start|>user
```

*Suffix*

```
<|im_end|>
<|im_start|>assistant
<think>

</think>
```

**Instructions** $I$.

*NLI retrieval*

```
Given a piece of text, retrieve the passage that entails the text the best.
```

*Label retrieval*

```
Given a piece of text, retrieve relevant label descriptions
that best match the text.
```

### C.2.1   BINARY DECISION VIA "YES/NO" TOKENS

Let $\tau_{\text{yes}}$ and $\tau_{\text{no}}$ be the token IDs that realise the strings "yes" and "no". Denote the final-step logit vector by $v = L_{S-1} \in \mathbb{R}^V$. We extract

$$v_{\tau_{\text{yes}}}, v_{\tau_{\text{no}}}$$

and compute the entailment probability as

$$p_{\text{yes}} = \frac{e^{v_{\tau_{\text{yes}}}}}{e^{v_{\tau_{\text{yes}}}} + e^{v_{\tau_{\text{no}}}}}$$

### C.3   INSTRUCTION-TUNED LLMS

For zero-shot classification with instruction-tuned LLMs, we frame the task as a multiple-choice problem. Each candidate label is assigned a unique single-token character option (e.g., "A", "B", "C", etc.). For every input text $x$ and label set $\mathcal{Y} = \{y_1, \ldots, y_K\}$, we construct a prompt of the form:

$$P = \texttt{system\_instructions} + \langle\text{Text}\rangle: x + \langle\text{Options}\rangle: \mathcal{O} + \texttt{suffix}$$

where $\mathcal{O}$ enumerates each label verbalizer with its assigned character option.

**Fixed strings.**

*System instructions*

```
You are a text classifier.
You will be given a text and several mutually exclusive options.
Each option is prefixed by a single letter (e.g. A, b, ...).
Your task is to choose the single best option.

IMPORTANT:
- Answer with EXACTLY ONE LETTER used to prefix the options.
- Do NOT output any words, punctuation, or explanation.
```

*Text and options format*

```
TEXT:
{text}

OPTIONS:
A) {label_verbalizer_1}
B) {label_verbalizer_2}
...
```

*Suffix*

```
Answer: The correct option is letter
```

### C.3.1 CLASSIFICATION VIA NEXT-TOKEN PROBABILITIES

Let $\mathcal{A} = \{a_1, \ldots, a_K\}$ be the set of single-token character options corresponding to the $K$ candidate labels. We tokenize each symbol $a_k$ to obtain its vocabulary index $\tau_{a_k}$.

Given the constructed prompt $P$, we perform a single forward pass through the decoder-only model to obtain the logit vector at the final position:

$$v = L_{S-1} \in \mathbb{R}^V$$

where $V$ is the vocabulary size and $S$ is the sequence length.

We extract the logits corresponding to the option characters and compute a softmax over this restricted set:

$$p(y_k \mid x) = \frac{\exp(v_{\tau_{a_k}})}{\sum_{j=1}^{K} \exp(v_{\tau_{a_j}})}$$

The predicted label is then:

$$\hat{y} = \underset{k \in \{1, \ldots, K\}}{\arg\max} \; p(y_k \mid x)$$

This approach requires only a single forward pass per unique input text (rather than one pass per text–label pair), making it substantially more efficient than per-label scoring methods while still leveraging the model's instruction-following capabilities.

## D DISAGGREGATED RESULTS

To better understand where different model families succeed or fail, we complement the main results with disaggregated scores by dataset and metric. Tables 5–8 report macro-averaged $F_1$, micro-averaged accuracy, and macro-averaged recall and precision for each BTZSC dataset and model.

Overall, the disaggregated view confirms the aggregate picture from the main benchmark. Base encoders perform poorly and inconsistently across datasets, with low $F_1$ and systematically weak recall and precision, especially on intent and emotion tasks. NLI-tuned cross-encoders and rerankers form the strongest families: they attain uniformly high $F_1$, recall, and precision on most topic and sentiment tasks, and maintain relatively strong performance on more challenging intent datasets. Embedding models and instruction-tuned LLMs sit in between: the best embeddings (e.g. GTE-, BGE-, e5-, and Qwen3-based models) and stronger LLMs (Qwen3 and Nemo) reach competitive $F_1$ scores, but the disaggregated metrics reveal systematic variation across domains.

For the embedding models, the hierarchy already visible in aggregate metrics becomes particularly clear. The baseline all-MiniLM-L6-v2 underperforms with average recall around 0.45 and precision around 0.50, and struggles notably on high-cardinality or nuanced datasets such as BANKING77, BIASFRAMES-INTENT, EMOTIONDIARY, and EMPATHETICDIALOGUES. In contrast, mid-tier models (e5-base/large, BGE-base, GTE-base, Qwen3-Embedding-0.6B) achieve average recall and precision in the low-to-mid 0.60s and deliver very strong performance on classical sentiment tasks (Amazon Polarity, IMDB, RottenTomatoes, Yelp, FinancialPhraseBank). Top-tier embeddings such as gte-large-en-v1.5 and Qwen3-Embedding-8B further improve both recall and precision (Avg R/P $\approx 0.67$–$0.70$) and show more stable behavior across topic, sentiment, and intent tasks. Their performance on emotion and social-media intent remains clearly below that on standard sentiment, but still outperforms weaker embeddings and base encoders.

The disaggregated scores also highlight dataset-level effects. Standard sentiment benchmarks (Amazon Polarity, IMDB, RottenTomatoes, Yelp, FinancialPhraseBank) are close to saturated for all strong models: NLI cross-encoders, rerankers, top embeddings, and strong LLMs all reach recall and precision around 0.9 or higher, indicating that these tasks are no longer particularly discriminative for modern architectures. In contrast, several BTZSC datasets expose sharp differences. The Manifesto dataset is consistently hard across families, with markedly lower recall and precision even for the strongest models, reflecting the difficulty of multi-class political text classification. BiasFrames (offensive/sex/intent) and WikiToxic variants likewise reveal non-trivial gaps: while top rerankers and embeddings achieve good performance, smaller or weaker models often show pronounced

asymmetries (e.g. reasonable recall but poor precision, or vice versa), especially on BIASFRAMES-INTENT. For intent classification, BANKING77 and MASSIVE again favour rerankers and strong embeddings, with instruction-tuned LLMs performing competitively but not surpassing them. Finally, the emotion datasets (EMOTIONDIARY, EMPATHETICDIALOGUES) are the most challenging slice for all families, with recall and precision typically 0.1–0.2 points lower than on sentiment and topic tasks, even for the strongest models.

Comparing recall and precision directly, we do not observe a systematic family-wide recall–precision trade-off. Top embeddings, rerankers, and LLMs tend to be reasonably balanced, with a mild tendency for the best embeddings and LLMs to exhibit slightly higher precision than recall on the hardest tasks. Rerankers, in particular, are recall-preserving on toxicity and social-media intent datasets, while maintaining acceptable precision, which is desirable for high-recall applications such as safety filtering. Taken together, these disaggregated results show that BTZSC can be used not only to rank models by a single headline metric, but also to characterise their error profiles across domains (news, political text, product reviews, social media, dialogue) and task types (topic, sentiment, intent, emotion). In this sense, BTZSC provides a nuanced testbed for analysing how different model families trade off recall and precision across a diverse classification landscape.

| Task | Topic | | | | | | | | | | | Sentiment | | | | | | Intent | | | Emotion | | |
|---|---|---|---|---|---|---|---|---|---|---|---|---|---|---|---|---|---|---|---|---|---|---|---|
| Model | AGN | BF-Off | BF-Sex | CAPS | MAN | TT | WT-Ins | WT-Obs | WT-Thr | WT-Agg | YT | AmzPol | AppR | FPB | IMDB | RT | Yelp | B77 | BF-Int | MASS | EmoD | Emp | Avg F1 |
| **Base encoders** | | | | | | | | | | | | | | | | | | | | | | | |
| bert-large-uncased | 0.17 | 0.48 | 0.48 | 0.03 | 0.01 | 0.33 | 0.51 | 0.58 | 0.49 | 0.53 | 0.08 | 0.42 | 0.36 | 0.28 | 0.38 | 0.47 | 0.36 | 0.02 | 0.42 | 0.00 | 0.16 | 0.00 | 0.30 |
| deberta-v3-large | 0.18 | 0.49 | 0.12 | 0.01 | 0.01 | 0.34 | 0.55 | 0.56 | 0.51 | 0.53 | 0.03 | 0.33 | 0.37 | 0.30 | 0.33 | 0.35 | 0.37 | 0.01 | 0.45 | 0.00 | 0.10 | 0.00 | 0.27 |
| ModernBERT-large | 0.10 | 0.53 | 0.46 | 0.01 | 0.01 | 0.36 | 0.46 | 0.58 | 0.13 | 0.60 | 0.08 | 0.33 | 0.44 | 0.30 | 0.34 | 0.36 | 0.43 | 0.02 | 0.38 | 0.02 | 0.06 | 0.01 | 0.27 |
| **NLI cross-encoders** | | | | | | | | | | | | | | | | | | | | | | | |
| bart-large-mnli | 0.71 | 0.37 | 0.07 | 0.33 | 0.09 | 0.51 | 0.33 | 0.69 | 0.08 | 0.55 | 0.27 | 0.93 | 0.92 | 0.47 | 0.93 | 0.83 | 0.96 | 0.28 | 0.60 | 0.41 | 0.44 | 0.38 | 0.51 |
| nli-roberta-base | 0.69 | 0.47 | 0.18 | 0.14 | 0.02 | 0.46 | 0.54 | 0.71 | 0.14 | 0.70 | 0.35 | 0.89 | 0.89 | 0.50 | 0.83 | 0.80 | 0.89 | 0.07 | 0.51 | 0.32 | 0.34 | 0.32 | 0.49 |
| bert-base-uncased-nli | 0.68 | 0.57 | 0.43 | 0.01 | 0.00 | 0.34 | 0.76 | 0.75 | 0.26 | 0.61 | 0.34 | 0.86 | 0.84 | 0.43 | 0.79 | 0.75 | 0.89 | 0.01 | 0.56 | 0.32 | 0.36 | 0.15 | 0.49 |
| bert-large-uncased-nli | 0.74 | 0.54 | 0.58 | 0.13 | 0.02 | 0.39 | 0.81 | 0.78 | 0.55 | 0.64 | 0.21 | 0.84 | 0.85 | 0.64 | 0.80 | 0.71 | 0.90 | 0.08 | 0.61 | 0.38 | 0.42 | 0.12 | 0.53 |
| bert-large-uncased-nli-triplet | 0.73 | 0.64 | 0.43 | 0.13 | 0.02 | 0.36 | 0.73 | 0.83 | 0.39 | 0.77 | 0.33 | 0.84 | 0.85 | 0.58 | 0.78 | 0.72 | 0.90 | 0.06 | 0.60 | 0.37 | 0.28 | 0.20 | 0.52 |
| deberta-v3-base-nli | 0.76 | 0.50 | 0.44 | 0.17 | 0.04 | 0.33 | 0.68 | 0.83 | 0.37 | 0.78 | 0.46 | 0.90 | 0.91 | 0.68 | 0.91 | 0.82 | 0.93 | 0.12 | 0.46 | 0.34 | 0.38 | 0.29 | 0.55 |
| deberta-v3-large-nli | 0.81 | 0.54 | 0.27 | 0.21 | 0.06 | 0.34 | 0.60 | 0.82 | 0.32 | 0.80 | 0.53 | 0.92 | 0.93 | 0.80 | 0.90 | 0.85 | 0.98 | 0.35 | 0.66 | 0.42 | 0.44 | 0.44 | 0.59 |
| deberta-v3-large-nli-triplet | 0.83 | 0.64 | 0.27 | 0.25 | 0.06 | 0.41 | 0.69 | 0.84 | 0.44 | 0.82 | 0.28 | 0.93 | 0.93 | 0.79 | 0.93 | 0.84 | 0.98 | 0.24 | 0.70 | 0.40 | 0.42 | 0.41 | 0.60 |
| modernbert-base-nli | 0.74 | 0.55 | 0.64 | 0.12 | 0.02 | 0.42 | 0.75 | 0.68 | 0.29 | 0.75 | 0.40 | 0.91 | 0.91 | 0.60 | 0.89 | 0.74 | 0.96 | 0.10 | 0.46 | 0.25 | 0.28 | 0.30 | 0.53 |
| modernbert-large-nli | 0.76 | 0.42 | 0.44 | 0.04 | 0.09 | 0.37 | 0.59 | 0.81 | 0.45 | 0.72 | 0.48 | 0.93 | 0.92 | 0.54 | 0.91 | 0.86 | 0.98 | 0.21 | 0.63 | 0.36 | 0.30 | 0.30 | 0.55 |
| modernbert-large-nli-triplet | 0.71 | 0.45 | 0.34 | 0.05 | 0.06 | 0.41 | 0.67 | 0.80 | 0.42 | 0.73 | 0.25 | 0.93 | 0.92 | 0.65 | 0.91 | 0.87 | 0.98 | 0.21 | 0.60 | 0.42 | 0.37 | 0.31 | 0.55 |
| **Rerankers** | | | | | | | | | | | | | | | | | | | | | | | |
| ms-marco-MiniLM-L6-v2 | 0.40 | 0.62 | 0.47 | 0.30 | 0.07 | 0.39 | 0.50 | 0.51 | 0.45 | 0.47 | 0.30 | 0.68 | 0.71 | 0.28 | 0.62 | 0.58 | 0.65 | 0.24 | 0.61 | 0.05 | 0.20 | 0.18 | 0.42 |
| gte-reranker-modernbert-base | 0.68 | 0.52 | 0.60 | 0.42 | 0.17 | 0.58 | 0.53 | 0.56 | 0.43 | 0.55 | 0.37 | 0.91 | 0.92 | 0.50 | 0.84 | 0.80 | 0.96 | 0.65 | 0.36 | 0.54 | 0.45 | 0.39 | 0.58 |
| bge-reranker-base | 0.63 | 0.47 | 0.30 | 0.40 | 0.16 | 0.47 | 0.41 | 0.55 | 0.27 | 0.57 | 0.41 | 0.66 | 0.79 | 0.35 | 0.69 | 0.57 | 0.65 | 0.48 | 0.50 | 0.43 | 0.30 | 0.29 | 0.47 |
| bge-reranker-large | 0.73 | 0.53 | 0.20 | 0.48 | 0.16 | 0.44 | 0.38 | 0.53 | 0.15 | 0.60 | 0.53 | 0.87 | 0.89 | 0.49 | 0.80 | 0.76 | 0.88 | 0.57 | 0.59 | 0.45 | 0.39 | 0.34 | 0.53 |
| Qwen3-Reranker-0.6B | 0.79 | 0.57 | 0.08 | 0.53 | 0.27 | 0.34 | 0.74 | 0.80 | 0.50 | 0.79 | 0.55 | 0.91 | 0.89 | 0.41 | 0.88 | 0.78 | 0.95 | 0.63 | 0.50 | 0.53 | 0.49 | 0.41 | 0.61 |
| Qwen3-Reranker-8B | 0.79 | 0.77 | 0.64 | 0.66 | 0.33 | 0.36 | 0.82 | 0.88 | 0.58 | 0.86 | 0.61 | 0.96 | 0.93 | 0.82 | 0.95 | 0.90 | 0.98 | 0.69 | 0.74 | 0.67 | 0.49 | 0.48 | 0.72 |
| **Embedding models** | | | | | | | | | | | | | | | | | | | | | | | |
| all-MiniLM-L6-v2 | 0.49 | 0.48 | 0.51 | 0.48 | 0.15 | 0.40 | 0.32 | 0.50 | 0.26 | 0.51 | 0.36 | 0.35 | 0.41 | 0.31 | 0.34 | 0.34 | 0.33 | 0.43 | 0.47 | 0.33 | 0.11 | 0.15 | 0.37 |
| e5-base-v2 | 0.76 | 0.59 | 0.20 | 0.53 | 0.21 | 0.44 | 0.74 | 0.64 | 0.31 | 0.67 | 0.55 | 0.93 | 0.93 | 0.46 | 0.90 | 0.84 | 0.95 | 0.62 | 0.60 | 0.47 | 0.43 | 0.37 | 0.60 |
| e5-large-v2 | 0.79 | 0.52 | 0.39 | 0.50 | 0.22 | 0.47 | 0.51 | 0.66 | 0.29 | 0.69 | 0.52 | 0.94 | 0.91 | 0.52 | 0.93 | 0.85 | 0.98 | 0.58 | 0.54 | 0.51 | 0.44 | 0.38 | 0.60 |
| e5-mistral-7b-instruct | 0.77 | 0.47 | 0.08 | 0.62 | 0.30 | 0.42 | 0.40 | 0.33 | 0.09 | 0.37 | 0.64 | 0.94 | 0.93 | 0.62 | 0.91 | 0.84 | 0.98 | 0.65 | 0.67 | 0.63 | 0.50 | 0.50 | 0.58 |
| bge-base-en-v1.5 | 0.63 | 0.57 | 0.16 | 0.54 | 0.20 | 0.48 | 0.46 | 0.68 | 0.17 | 0.72 | 0.51 | 0.93 | 0.90 | 0.43 | 0.90 | 0.81 | 0.94 | 0.64 | 0.57 | 0.53 | 0.43 | 0.30 | 0.57 |
| bge-large-en-v1.5 | 0.77 | 0.46 | 0.13 | 0.56 | 0.26 | 0.40 | 0.44 | 0.44 | 0.11 | 0.50 | 0.57 | 0.95 | 0.92 | 0.46 | 0.93 | 0.82 | 0.95 | 0.68 | 0.54 | 0.53 | 0.44 | 0.35 | 0.55 |
| gte-base-en-v1.5 | 0.75 | 0.63 | 0.10 | 0.57 | 0.21 | 0.35 | 0.61 | 0.74 | 0.22 | 0.65 | 0.56 | 0.90 | 0.93 | 0.47 | 0.85 | 0.84 | 0.97 | 0.66 | 0.60 | 0.51 | 0.42 | 0.32 | 0.58 |
| gte-large-en-v1.5 | 0.74 | 0.47 | 0.21 | 0.55 | 0.28 | 0.40 | 0.75 | 0.82 | 0.38 | 0.82 | 0.56 | 0.95 | 0.91 | 0.49 | 0.94 | 0.87 | 0.93 | 0.63 | 0.56 | 0.57 | 0.40 | 0.34 | 0.62 |
| gte-modernbert-base | 0.76 | 0.59 | 0.13 | 0.48 | 0.24 | 0.47 | 0.63 | 0.52 | 0.15 | 0.48 | 0.54 | 0.95 | 0.92 | 0.64 | 0.91 | 0.82 | 0.96 | 0.64 | 0.63 | 0.61 | 0.44 | 0.39 | 0.59 |
| Qwen3-Embedding-0.6B | 0.66 | 0.52 | 0.32 | 0.53 | 0.24 | 0.47 | 0.52 | 0.54 | 0.32 | 0.66 | 0.55 | 0.90 | 0.87 | 0.49 | 0.88 | 0.76 | 0.96 | 0.64 | 0.45 | 0.59 | 0.48 | 0.38 | 0.58 |
| Qwen3-Embedding-8B | 0.77 | 0.54 | 0.26 | 0.55 | 0.32 | 0.41 | 0.31 | 0.39 | 0.27 | 0.47 | 0.59 | 0.94 | 0.92 | 0.72 | 0.95 | 0.86 | 0.96 | 0.71 | 0.40 | 0.65 | 0.54 | 0.47 | 0.59 |
| **Instruction-tuned LLMs** | | | | | | | | | | | | | | | | | | | | | | | |
| gemma-3-270m-it | 0.18 | 0.54 | 0.26 | 0.03 | 0.00 | 0.44 | 0.47 | 0.49 | 0.20 | 0.51 | 0.03 | 0.46 | 0.48 | 0.21 | 0.47 | 0.47 | 0.43 | 0.00 | 0.38 | 0.00 | 0.06 | 0.01 | 0.28 |
| gemma-3-1b-it | 0.36 | 0.48 | 0.44 | 0.16 | 0.01 | 0.39 | 0.50 | 0.55 | 0.21 | 0.52 | 0.17 | 0.52 | 0.68 | 0.25 | 0.56 | 0.57 | 0.53 | 0.10 | 0.52 | 0.08 | 0.13 | 0.16 | 0.36 |
| Llama-3.2-3B-Instruct | 0.67 | 0.57 | 0.30 | 0.44 | 0.10 | 0.46 | 0.50 | 0.52 | 0.25 | 0.53 | 0.46 | 0.51 | 0.51 | 0.28 | 0.51 | 0.49 | 0.44 | 0.39 | 0.46 | 0.37 | 0.33 | 0.37 | 0.43 |
| Qwen3-4B | 0.82 | 0.69 | 0.64 | 0.55 | 0.17 | 0.34 | 0.85 | 0.89 | 0.67 | 0.87 | 0.52 | 0.94 | 0.92 | 0.66 | 0.92 | 0.87 | 0.98 | 0.37 | 0.45 | 0.38 | 0.43 | 0.32 | 0.65 |
| Phi-4-mini-instruct | 0.57 | 0.58 | 0.26 | 0.46 | 0.15 | 0.47 | 0.53 | 0.53 | 0.25 | 0.56 | 0.47 | 0.56 | 0.55 | 0.23 | 0.56 | 0.53 | 0.52 | 0.42 | 0.41 | 0.28 | 0.30 | 0.30 | 0.43 |
| Qwen3-8B | 0.85 | 0.72 | 0.67 | 0.53 | 0.18 | 0.34 | 0.84 | 0.89 | 0.72 | 0.86 | 0.55 | 0.94 | 0.92 | 0.75 | 0.94 | 0.86 | 0.98 | 0.39 | 0.67 | 0.39 | 0.40 | 0.24 | 0.66 |
| Mistral-Nemo-Instruct-2407 | 0.84 | 0.79 | 0.81 | 0.51 | 0.18 | 0.41 | 0.88 | 0.91 | 0.83 | 0.85 | 0.59 | 0.94 | 0.90 | 0.75 | 0.95 | 0.54 | 0.97 | 0.35 | 0.65 | 0.36 | 0.44 | 0.29 | 0.67 |

Table 5: Zero-shot classification results on BTZSC by dataset (macro-averaged F1). Abbreviations: AGN = AGNEWS, BF-Off / BF-Sex = BiasFrames (offensive / sex), CAPS = CAPSOTU, MAN = Manifesto, TT = TrueTeacher, WT-Ins / WT-Obs / WT-Thr / WT-Agg = WikiToxic (insult / obscene / threat / toxic aggregated), YT = Yahoo Topics, AmzPol = Amazon Polarity, AppR = AppReviews, FPB = FinancialPhraseBank, RT = RottenTomatoes, Yelp = YelpReviews, B77 = Banking77, BF-Int = BiasFrames (intent), MASS = MASSIVE (intent), EmoD = EmotionDiary, Emp = EmpatheticDialogues.

| Task | Topic / toxicity | | | | | | | | | | | Sentiment | | | | | | Intent | | | Emotion | | |
|---|---|---|---|---|---|---|---|---|---|---|---|---|---|---|---|---|---|---|---|---|---|---|---|
| Model | AGN | BF-Off | BF-Sex | CAPS | MAN | TT | WT-Ins | WT-Obs | WT-Thr | WT-Agg | YT | AmzPol | AppR | FPB | IMDB | RT | Yelp | B77 | BF-Int | MASS | EmoD | Emp | Avg Acc |
| **Base encoders** | | | | | | | | | | | | | | | | | | | | | | | |
| bert-large-uncased | 0.26 | 0.57 | 0.91 | 0.09 | 0.02 | 0.49 | 0.58 | 0.58 | 0.78 | 0.55 | 0.13 | 0.48 | 0.51 | 0.56 | 0.52 | 0.52 | 0.52 | 0.03 | 0.44 | 0.01 | 0.20 | 0.03 | 0.40 |
| deberta-v3-large | 0.28 | 0.58 | 0.12 | 0.01 | 0.04 | 0.51 | 0.65 | 0.64 | 0.86 | 0.58 | 0.11 | 0.49 | 0.51 | 0.40 | 0.49 | 0.50 | 0.52 | 0.03 | 0.47 | 0.00 | 0.20 | 0.03 | 0.36 |
| ModernBERT-large | 0.24 | 0.53 | 0.72 | 0.03 | 0.01 | 0.51 | 0.46 | 0.59 | 0.14 | 0.60 | 0.15 | 0.49 | 0.50 | 0.57 | 0.51 | 0.50 | 0.51 | 0.03 | 0.48 | 0.02 | 0.14 | 0.02 | 0.35 |
| **NLI cross-encoders** | | | | | | | | | | | | | | | | | | | | | | | |
| bart-large-mnli | 0.73 | 0.57 | 0.08 | 0.39 | 0.09 | 0.51 | 0.42 | 0.69 | 0.08 | 0.61 | 0.30 | 0.93 | 0.92 | 0.42 | 0.93 | 0.83 | 0.96 | 0.29 | 0.63 | 0.43 | 0.50 | 0.39 | 0.53 |
| nli-roberta-base | 0.70 | 0.58 | 0.19 | 0.23 | 0.02 | 0.48 | 0.56 | 0.71 | 0.14 | 0.70 | 0.41 | 0.89 | 0.89 | 0.45 | 0.83 | 0.80 | 0.89 | 0.08 | 0.51 | 0.36 | 0.34 | 0.32 | 0.50 |
| bert-base-uncased-nli | 0.69 | 0.57 | 0.53 | 0.02 | 0.02 | 0.49 | 0.76 | 0.77 | 0.29 | 0.65 | 0.38 | 0.86 | 0.84 | 0.40 | 0.79 | 0.76 | 0.89 | 0.02 | 0.56 | 0.28 | 0.39 | 0.15 | 0.50 |
| bert-large-uncased-nli | 0.75 | 0.60 | 0.76 | 0.22 | 0.02 | 0.49 | 0.81 | 0.79 | 0.75 | 0.67 | 0.25 | 0.84 | 0.86 | 0.65 | 0.81 | 0.72 | 0.90 | 0.08 | 0.61 | 0.35 | 0.55 | 0.13 | 0.57 |
| bert-large-uncased-nli-triplet | 0.73 | 0.67 | 0.53 | 0.24 | 0.02 | 0.49 | 0.73 | 0.83 | 0.50 | 0.78 | 0.42 | 0.84 | 0.85 | 0.59 | 0.79 | 0.73 | 0.90 | 0.07 | 0.60 | 0.37 | 0.30 | 0.21 | 0.55 |
| deberta-v3-base-nli | 0.76 | 0.60 | 0.55 | 0.23 | 0.05 | 0.49 | 0.68 | 0.83 | 0.46 | 0.78 | 0.53 | 0.90 | 0.91 | 0.67 | 0.91 | 0.82 | 0.93 | 0.13 | 0.51 | 0.33 | 0.38 | 0.32 | 0.58 |
| deberta-v3-large-nli | 0.81 | 0.63 | 0.30 | 0.23 | 0.11 | 0.50 | 0.61 | 0.82 | 0.37 | 0.80 | 0.60 | 0.92 | 0.93 | 0.82 | 0.90 | 0.85 | 0.98 | 0.36 | 0.67 | 0.44 | 0.49 | 0.46 | 0.62 |
| deberta-v3-large-nli-triplet | 0.83 | 0.67 | 0.30 | 0.27 | 0.09 | 0.51 | 0.69 | 0.84 | 0.57 | 0.82 | 0.37 | 0.93 | 0.93 | 0.82 | 0.93 | 0.84 | 0.98 | 0.24 | 0.70 | 0.42 | 0.44 | 0.42 | 0.62 |
| modernbert-base-nli | 0.75 | 0.62 | 0.84 | 0.18 | 0.03 | 0.50 | 0.75 | 0.71 | 0.33 | 0.75 | 0.44 | 0.91 | 0.91 | 0.61 | 0.89 | 0.75 | 0.96 | 0.09 | 0.52 | 0.28 | 0.28 | 0.29 | 0.56 |
| modernbert-large-nli | 0.76 | 0.59 | 0.55 | 0.09 | 0.10 | 0.52 | 0.61 | 0.81 | 0.59 | 0.73 | 0.54 | 0.93 | 0.92 | 0.70 | 0.91 | 0.86 | 0.98 | 0.24 | 0.63 | 0.39 | 0.28 | 0.31 | 0.59 |
| modernbert-large-nli-triplet | 0.72 | 0.60 | 0.40 | 0.13 | 0.08 | 0.51 | 0.67 | 0.80 | 0.54 | 0.74 | 0.29 | 0.93 | 0.92 | 0.73 | 0.91 | 0.87 | 0.98 | 0.22 | 0.60 | 0.44 | 0.37 | 0.34 | 0.58 |
| **Rerankers** | | | | | | | | | | | | | | | | | | | | | | | |
| ms-marco-MiniLM-L6-v2 | 0.42 | 0.62 | 0.75 | 0.32 | 0.14 | 0.52 | 0.53 | 0.55 | 0.65 | 0.50 | 0.33 | 0.69 | 0.71 | 0.29 | 0.62 | 0.58 | 0.65 | 0.22 | 0.62 | 0.04 | 0.20 | 0.18 | 0.46 |
| gte-reranker-modernbert-base | 0.71 | 0.52 | 0.85 | 0.48 | 0.26 | 0.60 | 0.55 | 0.57 | 0.60 | 0.55 | 0.39 | 0.91 | 0.92 | 0.47 | 0.84 | 0.80 | 0.96 | 0.63 | 0.47 | 0.57 | 0.52 | 0.39 | 0.62 |
| bge-reranker-base | 0.65 | 0.48 | 0.35 | 0.44 | 0.27 | 0.47 | 0.46 | 0.55 | 0.31 | 0.57 | 0.47 | 0.66 | 0.79 | 0.35 | 0.69 | 0.58 | 0.65 | 0.49 | 0.50 | 0.45 | 0.35 | 0.28 | 0.49 |
| bge-reranker-large | 0.74 | 0.58 | 0.21 | 0.52 | 0.32 | 0.45 | 0.44 | 0.56 | 0.16 | 0.62 | 0.59 | 0.87 | 0.89 | 0.48 | 0.80 | 0.76 | 0.88 | 0.56 | 0.60 | 0.47 | 0.41 | 0.34 | 0.56 |
| Qwen3-Reranker-0.6B | 0.80 | 0.64 | 0.08 | 0.59 | 0.40 | 0.50 | 0.75 | 0.81 | 0.70 | 0.79 | 0.61 | 0.91 | 0.89 | 0.38 | 0.88 | 0.78 | 0.95 | 0.62 | 0.56 | 0.52 | 0.55 | 0.42 | 0.64 |
| Qwen3-Reranker-8B | 0.80 | 0.78 | 0.82 | 0.68 | 0.46 | 0.52 | 0.82 | 0.88 | 0.78 | 0.86 | 0.67 | 0.96 | 0.93 | 0.84 | 0.95 | 0.90 | 0.98 | 0.67 | 0.75 | 0.72 | 0.56 | 0.49 | 0.76 |
| **Embedding models** | | | | | | | | | | | | | | | | | | | | | | | |
| all-MiniLM-L6-v2 | 0.50 | 0.50 | 0.77 | 0.53 | 0.30 | 0.49 | 0.41 | 0.50 | 0.30 | 0.53 | 0.38 | 0.49 | 0.53 | 0.42 | 0.49 | 0.50 | 0.49 | 0.44 | 0.53 | 0.34 | 0.11 | 0.17 | 0.44 |
| e5-base-v2 | 0.77 | 0.62 | 0.21 | 0.59 | 0.36 | 0.49 | 0.74 | 0.65 | 0.36 | 0.68 | 0.62 | 0.93 | 0.93 | 0.43 | 0.90 | 0.84 | 0.95 | 0.62 | 0.62 | 0.49 | 0.51 | 0.41 | 0.62 |
| e5-large-v2 | 0.79 | 0.52 | 0.50 | 0.55 | 0.34 | 0.48 | 0.54 | 0.66 | 0.33 | 0.69 | 0.56 | 0.94 | 0.91 | 0.45 | 0.93 | 0.85 | 0.98 | 0.56 | 0.54 | 0.51 | 0.49 | 0.43 | 0.62 |
| e5-mistral-7b-instruct | 0.78 | 0.60 | 0.08 | 0.62 | 0.50 | 0.48 | 0.46 | 0.43 | 0.09 | 0.51 | 0.71 | 0.94 | 0.93 | 0.57 | 0.92 | 0.84 | 0.98 | 0.65 | 0.68 | 0.67 | 0.55 | 0.54 | 0.62 |
| bge-base-en-v1.5 | 0.65 | 0.60 | 0.17 | 0.58 | 0.34 | 0.48 | 0.50 | 0.68 | 0.17 | 0.73 | 0.59 | 0.93 | 0.90 | 0.39 | 0.90 | 0.81 | 0.94 | 0.64 | 0.58 | 0.56 | 0.52 | 0.35 | 0.59 |
| bge-large-en-v1.5 | 0.77 | 0.57 | 0.13 | 0.61 | 0.43 | 0.47 | 0.49 | 0.50 | 0.11 | 0.57 | 0.63 | 0.95 | 0.92 | 0.41 | 0.94 | 0.82 | 0.95 | 0.68 | 0.58 | 0.52 | 0.55 | 0.39 | 0.59 |
| gte-base-en-v1.5 | 0.76 | 0.64 | 0.10 | 0.59 | 0.36 | 0.48 | 0.62 | 0.74 | 0.23 | 0.67 | 0.62 | 0.90 | 0.93 | 0.41 | 0.85 | 0.84 | 0.97 | 0.65 | 0.60 | 0.54 | 0.50 | 0.36 | 0.61 |
| gte-large-en-v1.5 | 0.75 | 0.48 | 0.22 | 0.59 | 0.39 | 0.45 | 0.75 | 0.82 | 0.47 | 0.82 | 0.62 | 0.95 | 0.91 | 0.47 | 0.94 | 0.87 | 0.93 | 0.63 | 0.56 | 0.58 | 0.45 | 0.36 | 0.64 |
| gte-modernbert-base | 0.76 | 0.63 | 0.13 | 0.52 | 0.38 | 0.47 | 0.63 | 0.55 | 0.16 | 0.55 | 0.60 | 0.95 | 0.92 | 0.64 | 0.91 | 0.82 | 0.96 | 0.63 | 0.63 | 0.65 | 0.50 | 0.40 | 0.61 |
| Qwen3-Embedding-0.6B | 0.68 | 0.56 | 0.38 | 0.55 | 0.40 | 0.47 | 0.55 | 0.55 | 0.38 | 0.67 | 0.60 | 0.90 | 0.87 | 0.45 | 0.88 | 0.76 | 0.96 | 0.64 | 0.53 | 0.63 | 0.53 | 0.40 | 0.61 |
| Qwen3-Embedding-8B | 0.78 | 0.64 | 0.30 | 0.52 | 0.47 | 0.51 | 0.41 | 0.47 | 0.30 | 0.56 | 0.64 | 0.94 | 0.92 | 0.69 | 0.95 | 0.86 | 0.96 | 0.70 | 0.56 | 0.71 | 0.59 | 0.50 | 0.64 |
| **Instruction-tuned LLMs** | | | | | | | | | | | | | | | | | | | | | | | |
| gemma-3-270m-it | 0.24 | 0.59 | 0.29 | 0.09 | 0.00 | 0.50 | 0.49 | 0.51 | 0.22 | 0.54 | 0.08 | 0.51 | 0.51 | 0.27 | 0.52 | 0.50 | 0.47 | 0.01 | 0.41 | 0.01 | 0.11 | 0.04 | 0.31 |
| gemma-3-1b-it | 0.39 | 0.52 | 0.58 | 0.20 | 0.04 | 0.46 | 0.52 | 0.57 | 0.22 | 0.56 | 0.19 | 0.59 | 0.69 | 0.25 | 0.61 | 0.60 | 0.60 | 0.14 | 0.56 | 0.09 | 0.19 | 0.22 | 0.40 |
| Llama-3.2-3B-Instruct | 0.68 | 0.60 | 0.35 | 0.50 | 0.17 | 0.50 | 0.50 | 0.53 | 0.29 | 0.56 | 0.50 | 0.54 | 0.54 | 0.30 | 0.54 | 0.51 | 0.48 | 0.41 | 0.47 | 0.37 | 0.46 | 0.40 | 0.46 |
| Qwen3-4B | 0.82 | 0.71 | 0.83 | 0.60 | 0.29 | 0.51 | 0.86 | 0.89 | 0.88 | 0.87 | 0.56 | 0.94 | 0.92 | 0.62 | 0.92 | 0.87 | 0.98 | 0.40 | 0.58 | 0.40 | 0.54 | 0.34 | 0.70 |
| Phi-4-mini-instruct | 0.56 | 0.64 | 0.28 | 0.45 | 0.30 | 0.50 | 0.55 | 0.55 | 0.28 | 0.60 | 0.45 | 0.62 | 0.60 | 0.22 | 0.61 | 0.58 | 0.60 | 0.43 | 0.44 | 0.31 | 0.34 | 0.31 | 0.47 |
| Qwen3-8B | 0.85 | 0.75 | 0.86 | 0.57 | 0.29 | 0.51 | 0.84 | 0.89 | 0.91 | 0.86 | 0.58 | 0.94 | 0.92 | 0.76 | 0.94 | 0.86 | 0.98 | 0.40 | 0.70 | 0.41 | 0.54 | 0.26 | 0.71 |
| Mistral-Nemo-Instruct-2407 | 0.84 | 0.79 | 0.95 | 0.60 | 0.26 | 0.54 | 0.89 | 0.91 | 0.96 | 0.85 | 0.65 | 0.94 | 0.90 | 0.71 | 0.95 | 0.61 | 0.97 | 0.39 | 0.69 | 0.39 | 0.55 | 0.32 | 0.71 |

Table 6: Zero-shot classification results on BTZSC by dataset (micro-averaged accuracy). Abbreviations: AGN = AGNEWS, BF-Off / BF-Sex = BiasFrames (offensive / sex), CAPS = CAPSOTU, MAN = Manifesto, TT = TrueTeacher, WT-Ins / WT-Obs / WT-Thr / WT-Agg = WikiToxic (insult / obscene / threat / toxic aggregated), YT = Yahoo Topics, AmzPol = Amazon Polarity, AppR = AppReviews, FPB = FinancialPhraseBank, RT = RottenTomatoes, Yelp = YelpReviews, B77 = Banking77, BF-Int = BiasFrames (intent), MASS = MASSIVE (intent), EmoD = EmotionDiary, Emp = EmpatheticDialogues.

| Task | Topic / toxicity | | | | | | | | | | | Sentiment | | | | | | Intent | | | Emotion | | |
|---|---|---|---|---|---|---|---|---|---|---|---|---|---|---|---|---|---|---|---|---|---|---|---|
| Model | AGN | BF-Off | BF-Sex | CAPS | MAN | TT | WT-Ins | WT-Obs | WT-Thr | WT-Agg | YT | AmzPol | AppR | FPB | IMDB | RT | Yelp | B77 | BF-Int | MASS | EmoD | Emp | Avg R |
| **Base encoders** | | | | | | | | | | | | | | | | | | | | | | | |
| bert-large-uncased | 0.26 | 0.53 | 0.49 | 0.10 | 0.02 | 0.50 | 0.52 | 0.58 | 0.55 | 0.55 | 0.12 | 0.47 | 0.51 | 0.33 | 0.51 | 0.52 | 0.51 | 0.03 | 0.46 | 0.03 | 0.18 | 0.03 | 0.35 |
| deberta-v3-large | 0.27 | 0.53 | 0.50 | 0.04 | 0.02 | 0.50 | 0.58 | 0.58 | 0.52 | 0.57 | 0.09 | 0.50 | 0.51 | 0.39 | 0.50 | 0.50 | 0.51 | 0.03 | 0.49 | 0.00 | 0.18 | 0.03 | 0.36 |
| ModernBERT-large | 0.25 | 0.55 | 0.50 | 0.07 | 0.01 | 0.50 | 0.47 | 0.59 | 0.49 | 0.61 | 0.14 | 0.50 | 0.50 | 0.33 | 0.50 | 0.50 | 0.52 | 0.04 | 0.52 | 0.03 | 0.18 | 0.02 | 0.36 |
| **NLI cross-encoders** | | | | | | | | | | | | | | | | | | | | | | | |
| bart-large-mnli | 0.73 | 0.50 | 0.51 | 0.37 | 0.10 | 0.52 | 0.52 | 0.73 | 0.52 | 0.62 | 0.28 | 0.93 | 0.92 | 0.67 | 0.93 | 0.83 | 0.96 | 0.30 | 0.61 | 0.43 | 0.48 | 0.38 | 0.58 |
| nli-roberta-base | 0.71 | 0.53 | 0.56 | 0.18 | 0.02 | 0.48 | 0.63 | 0.72 | 0.55 | 0.71 | 0.36 | 0.89 | 0.89 | 0.65 | 0.83 | 0.80 | 0.89 | 0.07 | 0.53 | 0.33 | 0.46 | 0.32 | 0.55 |
| bert-base-uncased-nli | 0.69 | 0.58 | 0.73 | 0.05 | 0.02 | 0.50 | 0.79 | 0.75 | 0.62 | 0.64 | 0.34 | 0.86 | 0.84 | 0.60 | 0.79 | 0.76 | 0.89 | 0.01 | 0.57 | 0.31 | 0.42 | 0.15 | 0.54 |
| bert-large-uncased-nli | 0.75 | 0.56 | 0.82 | 0.17 | 0.04 | 0.50 | 0.84 | 0.78 | 0.85 | 0.67 | 0.22 | 0.84 | 0.85 | 0.76 | 0.81 | 0.72 | 0.90 | 0.08 | 0.61 | 0.36 | 0.43 | 0.13 | 0.58 |
| bert-large-uncased-nli-triplet | 0.74 | 0.64 | 0.73 | 0.19 | 0.04 | 0.50 | 0.77 | 0.84 | 0.74 | 0.78 | 0.37 | 0.84 | 0.85 | 0.71 | 0.78 | 0.73 | 0.90 | 0.07 | 0.61 | 0.43 | 0.35 | 0.20 | 0.58 |
| deberta-v3-base-nli | 0.77 | 0.55 | 0.74 | 0.19 | 0.08 | 0.50 | 0.73 | 0.83 | 0.72 | 0.79 | 0.47 | 0.90 | 0.91 | 0.80 | 0.91 | 0.82 | 0.93 | 0.13 | 0.54 | 0.34 | 0.42 | 0.31 | 0.61 |
| deberta-v3-large-nli | 0.82 | 0.59 | 0.63 | 0.23 | 0.11 | 0.50 | 0.68 | 0.84 | 0.67 | 0.80 | 0.54 | 0.92 | 0.93 | 0.84 | 0.90 | 0.85 | 0.98 | 0.38 | 0.69 | 0.44 | 0.48 | 0.45 | 0.65 |
| deberta-v3-large-nli-triplet | 0.83 | 0.64 | 0.63 | 0.30 | 0.09 | 0.52 | 0.74 | 0.85 | 0.77 | 0.82 | 0.33 | 0.93 | 0.93 | 0.80 | 0.93 | 0.84 | 0.98 | 0.26 | 0.70 | 0.44 | 0.47 | 0.42 | 0.65 |
| modernbert-base-nli | 0.75 | 0.58 | 0.85 | 0.19 | 0.05 | 0.50 | 0.78 | 0.68 | 0.65 | 0.75 | 0.40 | 0.91 | 0.91 | 0.64 | 0.89 | 0.75 | 0.96 | 0.09 | 0.55 | 0.26 | 0.34 | 0.28 | 0.58 |
| modernbert-large-nli | 0.76 | 0.52 | 0.75 | 0.08 | 0.14 | 0.51 | 0.67 | 0.81 | 0.79 | 0.74 | 0.49 | 0.93 | 0.92 | 0.51 | 0.91 | 0.86 | 0.98 | 0.24 | 0.64 | 0.38 | 0.35 | 0.30 | 0.60 |
| modernbert-large-nli-triplet | 0.72 | 0.54 | 0.67 | 0.08 | 0.08 | 0.51 | 0.73 | 0.81 | 0.75 | 0.75 | 0.26 | 0.93 | 0.92 | 0.61 | 0.91 | 0.87 | 0.98 | 0.22 | 0.61 | 0.45 | 0.42 | 0.32 | 0.60 |
| **Rerankers** | | | | | | | | | | | | | | | | | | | | | | | |
| ms-marco-MiniLM-L6-v2 | 0.42 | 0.64 | 0.49 | 0.36 | 0.10 | 0.51 | 0.50 | 0.52 | 0.60 | 0.50 | 0.30 | 0.69 | 0.71 | 0.49 | 0.63 | 0.58 | 0.66 | 0.22 | 0.61 | 0.06 | 0.27 | 0.18 | 0.46 |
| gte-reranker-modernbert-base | 0.71 | 0.52 | 0.69 | 0.48 | 0.21 | 0.60 | 0.54 | 0.56 | 0.62 | 0.55 | 0.39 | 0.91 | 0.92 | 0.65 | 0.84 | 0.80 | 0.96 | 0.67 | 0.50 | 0.57 | 0.48 | 0.39 | 0.62 |
| bge-reranker-base | 0.66 | 0.47 | 0.59 | 0.52 | 0.19 | 0.47 | 0.54 | 0.57 | 0.60 | 0.58 | 0.42 | 0.66 | 0.79 | 0.50 | 0.70 | 0.58 | 0.65 | 0.51 | 0.50 | 0.53 | 0.34 | 0.27 | 0.53 |
| bge-reranker-large | 0.74 | 0.55 | 0.58 | 0.57 | 0.17 | 0.45 | 0.53 | 0.61 | 0.56 | 0.62 | 0.53 | 0.87 | 0.89 | 0.58 | 0.81 | 0.76 | 0.88 | 0.58 | 0.59 | 0.57 | 0.48 | 0.33 | 0.60 |
| Qwen3-Reranker-0.6B | 0.79 | 0.60 | 0.50 | 0.59 | 0.31 | 0.50 | 0.75 | 0.80 | 0.75 | 0.79 | 0.54 | 0.91 | 0.89 | 0.62 | 0.88 | 0.78 | 0.95 | 0.66 | 0.54 | 0.61 | 0.52 | 0.42 | 0.67 |
| Qwen3-Reranker-8B | 0.80 | 0.76 | 0.88 | 0.69 | 0.35 | 0.51 | 0.85 | 0.88 | 0.88 | 0.86 | 0.60 | 0.96 | 0.93 | 0.80 | 0.95 | 0.90 | 0.98 | 0.72 | 0.74 | 0.74 | 0.51 | 0.49 | 0.76 |
| **Embedding models** | | | | | | | | | | | | | | | | | | | | | | | |
| all-MiniLM-L6-v2 | 0.51 | 0.55 | 0.58 | 0.50 | 0.17 | 0.49 | 0.50 | 0.51 | 0.58 | 0.54 | 0.35 | 0.50 | 0.54 | 0.32 | 0.51 | 0.50 | 0.50 | 0.44 | 0.56 | 0.39 | 0.23 | 0.16 | 0.45 |
| e5-base-v2 | 0.77 | 0.59 | 0.51 | 0.64 | 0.25 | 0.49 | 0.75 | 0.68 | 0.63 | 0.68 | 0.56 | 0.93 | 0.93 | 0.60 | 0.90 | 0.84 | 0.95 | 0.65 | 0.61 | 0.58 | 0.45 | 0.41 | 0.65 |
| e5-large-v2 | 0.79 | 0.52 | 0.57 | 0.56 | 0.25 | 0.48 | 0.61 | 0.70 | 0.64 | 0.70 | 0.56 | 0.94 | 0.91 | 0.68 | 0.93 | 0.85 | 0.98 | 0.59 | 0.55 | 0.60 | 0.50 | 0.42 | 0.65 |
| e5-mistral-7b-instruct | 0.78 | 0.54 | 0.51 | 0.69 | 0.32 | 0.48 | 0.55 | 0.51 | 0.53 | 0.52 | 0.64 | 0.94 | 0.93 | 0.75 | 0.91 | 0.84 | 0.98 | 0.69 | 0.67 | 0.70 | 0.54 | 0.54 | 0.66 |
| bge-base-en-v1.5 | 0.65 | 0.57 | 0.51 | 0.64 | 0.23 | 0.48 | 0.58 | 0.72 | 0.56 | 0.73 | 0.59 | 0.93 | 0.90 | 0.65 | 0.90 | 0.81 | 0.94 | 0.67 | 0.57 | 0.64 | 0.47 | 0.34 | 0.64 |
| bge-large-en-v1.5 | 0.77 | 0.52 | 0.52 | 0.66 | 0.31 | 0.48 | 0.57 | 0.56 | 0.51 | 0.58 | 0.58 | 0.95 | 0.92 | 0.64 | 0.93 | 0.82 | 0.95 | 0.70 | 0.56 | 0.62 | 0.48 | 0.39 | 0.64 |
| gte-base-en-v1.5 | 0.76 | 0.63 | 0.51 | 0.64 | 0.25 | 0.49 | 0.68 | 0.77 | 0.59 | 0.68 | 0.56 | 0.90 | 0.93 | 0.63 | 0.85 | 0.84 | 0.97 | 0.68 | 0.60 | 0.61 | 0.45 | 0.36 | 0.65 |
| gte-large-en-v1.5 | 0.75 | 0.50 | 0.58 | 0.64 | 0.33 | 0.45 | 0.79 | 0.84 | 0.72 | 0.83 | 0.55 | 0.95 | 0.91 | 0.69 | 0.94 | 0.87 | 0.93 | 0.65 | 0.57 | 0.66 | 0.45 | 0.36 | 0.68 |
| gte-modernbert-base | 0.76 | 0.60 | 0.52 | 0.61 | 0.29 | 0.48 | 0.69 | 0.60 | 0.56 | 0.56 | 0.54 | 0.95 | 0.92 | 0.66 | 0.91 | 0.82 | 0.96 | 0.66 | 0.63 | 0.69 | 0.49 | 0.40 | 0.65 |
| Qwen3-Embedding-0.6B | 0.68 | 0.53 | 0.56 | 0.65 | 0.28 | 0.47 | 0.62 | 0.59 | 0.68 | 0.68 | 0.54 | 0.90 | 0.87 | 0.65 | 0.88 | 0.76 | 0.96 | 0.67 | 0.50 | 0.67 | 0.55 | 0.41 | 0.64 |
| Qwen3-Embedding-8B | 0.78 | 0.59 | 0.46 | 0.65 | 0.40 | 0.50 | 0.51 | 0.59 | 0.63 | 0.57 | 0.58 | 0.94 | 0.92 | 0.78 | 0.95 | 0.86 | 0.96 | 0.74 | 0.52 | 0.72 | 0.60 | 0.50 | 0.67 |
| **Instruction-tuned LLMs** | | | | | | | | | | | | | | | | | | | | | | | |
| gemma-3-270m-it | 0.23 | 0.55 | 0.48 | 0.05 | 0.02 | 0.50 | 0.54 | 0.56 | 0.47 | 0.55 | 0.07 | 0.50 | 0.51 | 0.33 | 0.52 | 0.50 | 0.47 | 0.01 | 0.43 | 0.01 | 0.14 | 0.04 | 0.34 |
| gemma-3-1b-it | 0.38 | 0.49 | 0.65 | 0.15 | 0.03 | 0.46 | 0.59 | 0.62 | 0.56 | 0.57 | 0.23 | 0.58 | 0.69 | 0.46 | 0.60 | 0.60 | 0.59 | 0.15 | 0.54 | 0.11 | 0.20 | 0.21 | 0.43 |
| Llama-3.2-3B-Instruct | 0.68 | 0.57 | 0.52 | 0.48 | 0.12 | 0.49 | 0.55 | 0.57 | 0.50 | 0.56 | 0.46 | 0.54 | 0.54 | 0.37 | 0.53 | 0.51 | 0.47 | 0.44 | 0.48 | 0.40 | 0.35 | 0.39 | 0.48 |
| Qwen3-4B | 0.82 | 0.69 | 0.86 | 0.58 | 0.21 | 0.50 | 0.87 | 0.89 | 0.91 | 0.87 | 0.51 | 0.94 | 0.92 | 0.74 | 0.92 | 0.87 | 0.98 | 0.42 | 0.54 | 0.43 | 0.40 | 0.34 | 0.69 |
| Phi-4-mini-instruct | 0.56 | 0.60 | 0.51 | 0.48 | 0.17 | 0.50 | 0.62 | 0.60 | 0.60 | 0.61 | 0.46 | 0.61 | 0.60 | 0.41 | 0.60 | 0.58 | 0.59 | 0.45 | 0.46 | 0.29 | 0.32 | 0.32 | 0.50 |
| Qwen3-8B | 0.85 | 0.72 | 0.90 | 0.55 | 0.22 | 0.50 | 0.86 | 0.89 | 0.92 | 0.86 | 0.53 | 0.94 | 0.92 | 0.84 | 0.94 | 0.86 | 0.98 | 0.43 | 0.68 | 0.43 | 0.39 | 0.26 | 0.70 |
| Mistral-Nemo-Instruct-2407 | 0.84 | 0.79 | 0.85 | 0.52 | 0.20 | 0.53 | 0.88 | 0.92 | 0.93 | 0.85 | 0.60 | 0.94 | 0.90 | 0.79 | 0.95 | 0.61 | 0.97 | 0.40 | 0.67 | 0.40 | 0.43 | 0.30 | 0.69 |

Table 7: Zero-shot classification results on BTZSC by dataset (macro-averaged recall). Abbreviations: AGN = AGNEWS, BF-Off / BF-Sex = BiasFrames (offensive / sex), CAPS = CAPSOTU, MAN = Manifesto, TT = TrueTeacher, WT-Ins / WT-Obs / WT-Thr / WT-Agg = WikiToxic (insult / obscene / threat / toxic aggregated), YT = Yahoo Topics, AmzPol = Amazon Polarity, AppR = AppReviews, FPB = FinancialPhraseBank, RT = RottenTomatoes, Yelp = YelpReviews, B77 = Banking77, BF-Int = BiasFrames (intent), MASS = MASSIVE (intent), EmoD = EmotionDiary, Emp = EmpatheticDialogues.

| Task | Topic / toxicity | | | | | | | | | | | Sentiment | | | | | | Intent | | | Emotion | | |
|---|---|---|---|---|---|---|---|---|---|---|---|---|---|---|---|---|---|---|---|---|---|---|---|
| Model | AGN | BF-Off | BF-Sex | CAPS | MAN | TT | WT-Ins | WT-Obs | WT-Thr | WT-Agg | YT | AmzPol | AppR | FPB | IMDB | RT | Yelp | B77 | BF-Int | MASS | EmoD | Emp | Avg P |
| **Base encoders** | | | | | | | | | | | | | | | | | | | | | | | |
| bert-large-uncased | 0.18 | 0.55 | 0.47 | 0.04 | 0.03 | 0.25 | 0.53 | 0.58 | 0.51 | 0.56 | 0.11 | 0.45 | 0.67 | 0.38 | 0.55 | 0.53 | 0.58 | 0.02 | 0.45 | 0.00 | 0.19 | 0.02 | 0.35 |
| deberta-v3-large | 0.38 | 0.56 | 0.50 | 0.01 | 0.02 | 0.25 | 0.66 | 0.66 | 0.51 | 0.62 | 0.04 | 0.24 | 0.58 | 0.32 | 0.24 | 0.53 | 0.58 | 0.02 | 0.49 | 0.00 | 0.10 | 0.00 | 0.33 |
| ModernBERT-large | 0.06 | 0.55 | 0.50 | 0.02 | 0.02 | 0.53 | 0.47 | 0.58 | 0.50 | 0.61 | 0.07 | 0.24 | 0.49 | 0.29 | 0.26 | 0.50 | 0.56 | 0.05 | 0.59 | 0.04 | 0.07 | 0.01 | 0.32 |
| **NLI cross-encoders** | | | | | | | | | | | | | | | | | | | | | | | |
| bart-large-mnli | 0.77 | 0.78 | 0.53 | 0.59 | 0.22 | 0.52 | 0.62 | 0.76 | 0.52 | 0.74 | 0.53 | 0.93 | 0.92 | 0.64 | 0.93 | 0.84 | 0.96 | 0.47 | 0.65 | 0.56 | 0.48 | 0.48 | 0.66 |
| nli-roberta-base | 0.72 | 0.58 | 0.53 | 0.26 | 0.09 | 0.48 | 0.70 | 0.72 | 0.52 | 0.73 | 0.46 | 0.89 | 0.89 | 0.61 | 0.83 | 0.80 | 0.89 | 0.13 | 0.53 | 0.47 | 0.47 | 0.43 | 0.58 |
| bert-base-uncased-nli | 0.73 | 0.58 | 0.55 | 0.06 | 0.01 | 0.45 | 0.78 | 0.77 | 0.53 | 0.71 | 0.45 | 0.87 | 0.84 | 0.61 | 0.80 | 0.77 | 0.89 | 0.03 | 0.57 | 0.54 | 0.45 | 0.36 | 0.56 |
| bert-large-uncased-nli | 0.76 | 0.59 | 0.59 | 0.32 | 0.07 | 0.50 | 0.82 | 0.80 | 0.57 | 0.75 | 0.49 | 0.85 | 0.86 | 0.63 | 0.84 | 0.76 | 0.90 | 0.17 | 0.61 | 0.53 | 0.46 | 0.33 | 0.60 |
| bert-large-uncased-nli-triplet | 0.75 | 0.68 | 0.55 | 0.18 | 0.06 | 0.48 | 0.79 | 0.83 | 0.54 | 0.79 | 0.41 | 0.86 | 0.85 | 0.61 | 0.82 | 0.77 | 0.90 | 0.11 | 0.61 | 0.45 | 0.40 | 0.34 | 0.58 |
| deberta-v3-base-nli | 0.76 | 0.61 | 0.55 | 0.38 | 0.13 | 0.39 | 0.77 | 0.83 | 0.54 | 0.79 | 0.51 | 0.90 | 0.91 | 0.68 | 0.91 | 0.82 | 0.93 | 0.23 | 0.58 | 0.49 | 0.46 | 0.46 | 0.62 |
| deberta-v3-large-nli | 0.81 | 0.70 | 0.54 | 0.41 | 0.11 | 0.66 | 0.74 | 0.83 | 0.53 | 0.81 | 0.56 | 0.93 | 0.93 | 0.78 | 0.91 | 0.86 | 0.98 | 0.45 | 0.71 | 0.57 | 0.47 | 0.56 | 0.67 |
| deberta-v3-large-nli-triplet | 0.83 | 0.67 | 0.54 | 0.48 | 0.18 | 0.56 | 0.78 | 0.84 | 0.55 | 0.83 | 0.39 | 0.93 | 0.93 | 0.78 | 0.93 | 0.84 | 0.98 | 0.35 | 0.70 | 0.55 | 0.50 | 0.54 | 0.67 |
| modernbert-base-nli | 0.77 | 0.64 | 0.62 | 0.29 | 0.08 | 0.51 | 0.78 | 0.71 | 0.53 | 0.75 | 0.52 | 0.91 | 0.91 | 0.63 | 0.89 | 0.80 | 0.96 | 0.20 | 0.60 | 0.39 | 0.44 | 0.44 | 0.61 |
| modernbert-large-nli | 0.77 | 0.68 | 0.56 | 0.18 | 0.14 | 0.62 | 0.74 | 0.81 | 0.55 | 0.78 | 0.53 | 0.94 | 0.93 | 0.76 | 0.92 | 0.88 | 0.98 | 0.29 | 0.64 | 0.47 | 0.47 | 0.47 | 0.64 |
| modernbert-large-nli-triplet | 0.77 | 0.70 | 0.54 | 0.17 | 0.21 | 0.52 | 0.76 | 0.81 | 0.54 | 0.80 | 0.41 | 0.93 | 0.92 | 0.72 | 0.92 | 0.88 | 0.98 | 0.36 | 0.63 | 0.53 | 0.49 | 0.45 | 0.64 |
| **Rerankers** | | | | | | | | | | | | | | | | | | | | | | | |
| ms-marco-MiniLM-L6-v2 | 0.50 | 0.64 | 0.50 | 0.36 | 0.13 | 0.57 | 0.50 | 0.52 | 0.52 | 0.49 | 0.33 | 0.71 | 0.72 | 0.43 | 0.64 | 0.58 | 0.67 | 0.44 | 0.61 | 0.14 | 0.33 | 0.35 | 0.49 |
| gte-reranker-modernbert-base | 0.76 | 0.52 | 0.58 | 0.46 | 0.24 | 0.61 | 0.54 | 0.56 | 0.52 | 0.55 | 0.56 | 0.92 | 0.93 | 0.62 | 0.86 | 0.83 | 0.96 | 0.68 | 0.53 | 0.58 | 0.45 | 0.49 | 0.63 |
| bge-reranker-base | 0.67 | 0.47 | 0.52 | 0.41 | 0.18 | 0.47 | 0.61 | 0.58 | 0.52 | 0.58 | 0.45 | 0.66 | 0.79 | 0.44 | 0.70 | 0.58 | 0.65 | 0.57 | 0.50 | 0.44 | 0.32 | 0.40 | 0.52 |
| bge-reranker-large | 0.77 | 0.56 | 0.53 | 0.51 | 0.24 | 0.45 | 0.62 | 0.68 | 0.52 | 0.64 | 0.54 | 0.87 | 0.89 | 0.51 | 0.81 | 0.76 | 0.89 | 0.66 | 0.59 | 0.44 | 0.44 | 0.49 | 0.61 |
| Qwen3-Reranker-0.6B | 0.82 | 0.69 | 0.51 | 0.55 | 0.31 | 0.25 | 0.74 | 0.80 | 0.55 | 0.79 | 0.58 | 0.91 | 0.89 | 0.63 | 0.88 | 0.78 | 0.95 | 0.66 | 0.57 | 0.56 | 0.49 | 0.47 | 0.65 |
| Qwen3-Reranker-8B | 0.84 | 0.78 | 0.62 | 0.71 | 0.40 | 0.76 | 0.84 | 0.88 | 0.58 | 0.86 | 0.65 | 0.96 | 0.93 | 0.85 | 0.95 | 0.91 | 0.98 | 0.74 | 0.79 | 0.71 | 0.50 | 0.58 | 0.76 |
| **Embedding models** | | | | | | | | | | | | | | | | | | | | | | | |
| all-MiniLM-L6-v2 | 0.63 | 0.57 | 0.53 | 0.52 | 0.19 | 0.46 | 0.53 | 0.51 | 0.52 | 0.55 | 0.55 | 0.51 | 0.70 | 0.35 | 0.74 | 0.46 | 0.24 | 0.56 | 0.64 | 0.42 | 0.46 | 0.36 | 0.50 |
| e5-base-v2 | 0.78 | 0.61 | 0.50 | 0.51 | 0.25 | 0.49 | 0.74 | 0.71 | 0.53 | 0.70 | 0.57 | 0.93 | 0.93 | 0.53 | 0.90 | 0.84 | 0.95 | 0.65 | 0.62 | 0.50 | 0.45 | 0.50 | 0.64 |
| e5-large-v2 | 0.80 | 0.52 | 0.52 | 0.52 | 0.28 | 0.48 | 0.69 | 0.73 | 0.53 | 0.72 | 0.57 | 0.94 | 0.91 | 0.65 | 0.93 | 0.85 | 0.98 | 0.70 | 0.55 | 0.55 | 0.47 | 0.48 | 0.65 |
| e5-mistral-7b-instruct | 0.82 | 0.68 | 0.53 | 0.62 | 0.36 | 0.46 | 0.70 | 0.66 | 0.52 | 0.72 | 0.66 | 0.94 | 0.93 | 0.67 | 0.92 | 0.84 | 0.98 | 0.70 | 0.68 | 0.65 | 0.53 | 0.59 | 0.69 |
| bge-base-en-v1.5 | 0.68 | 0.59 | 0.51 | 0.53 | 0.25 | 0.48 | 0.69 | 0.74 | 0.52 | 0.75 | 0.56 | 0.93 | 0.91 | 0.60 | 0.90 | 0.82 | 0.95 | 0.68 | 0.58 | 0.54 | 0.50 | 0.40 | 0.64 |
| bge-large-en-v1.5 | 0.77 | 0.54 | 0.52 | 0.55 | 0.29 | 0.45 | 0.70 | 0.68 | 0.51 | 0.68 | 0.60 | 0.95 | 0.92 | 0.61 | 0.93 | 0.83 | 0.95 | 0.74 | 0.58 | 0.54 | 0.51 | 0.48 | 0.65 |
| gte-base-en-v1.5 | 0.76 | 0.63 | 0.52 | 0.58 | 0.24 | 0.43 | 0.73 | 0.78 | 0.52 | 0.74 | 0.58 | 0.91 | 0.93 | 0.60 | 0.88 | 0.85 | 0.97 | 0.68 | 0.60 | 0.54 | 0.43 | 0.37 | 0.65 |
| gte-large-en-v1.5 | 0.78 | 0.51 | 0.53 | 0.56 | 0.36 | 0.43 | 0.80 | 0.83 | 0.54 | 0.85 | 0.59 | 0.95 | 0.91 | 0.57 | 0.94 | 0.88 | 0.94 | 0.68 | 0.57 | 0.59 | 0.47 | 0.43 | 0.67 |
| gte-modernbert-base | 0.76 | 0.63 | 0.51 | 0.49 | 0.29 | 0.48 | 0.73 | 0.67 | 0.52 | 0.67 | 0.56 | 0.95 | 0.92 | 0.67 | 0.91 | 0.83 | 0.96 | 0.69 | 0.63 | 0.62 | 0.48 | 0.45 | 0.66 |
| Qwen3-Embedding-0.6B | 0.68 | 0.54 | 0.52 | 0.54 | 0.28 | 0.47 | 0.71 | 0.61 | 0.53 | 0.74 | 0.57 | 0.90 | 0.87 | 0.60 | 0.89 | 0.77 | 0.96 | 0.68 | 0.51 | 0.59 | 0.48 | 0.43 | 0.63 |
| Qwen3-Embedding-8B | 0.81 | 0.72 | 0.49 | 0.59 | 0.35 | 0.51 | 0.70 | 0.71 | 0.53 | 0.76 | 0.64 | 0.94 | 0.92 | 0.72 | 0.95 | 0.86 | 0.96 | 0.76 | 0.71 | 0.65 | 0.55 | 0.55 | 0.70 |
| **Instruction-tuned LLMs** | | | | | | | | | | | | | | | | | | | | | | | |
| gemma-3-270m-it | 0.25 | 0.57 | 0.49 | 0.04 | 0.00 | 0.50 | 0.56 | 0.58 | 0.49 | 0.57 | 0.12 | 0.50 | 0.52 | 0.36 | 0.53 | 0.50 | 0.45 | 0.00 | 0.41 | 0.00 | 0.04 | 0.01 | 0.34 |
| gemma-3-1b-it | 0.50 | 0.49 | 0.53 | 0.29 | 0.02 | 0.43 | 0.65 | 0.66 | 0.52 | 0.62 | 0.30 | 0.70 | 0.72 | 0.43 | 0.67 | 0.64 | 0.71 | 0.18 | 0.55 | 0.20 | 0.20 | 0.24 | 0.46 |
| Llama-3.2-3B-Instruct | 0.73 | 0.59 | 0.50 | 0.49 | 0.24 | 0.49 | 0.56 | 0.58 | 0.50 | 0.58 | 0.51 | 0.55 | 0.55 | 0.36 | 0.54 | 0.51 | 0.46 | 0.54 | 0.48 | 0.46 | 0.42 | 0.48 | 0.51 |
| Qwen3-4B | 0.85 | 0.72 | 0.62 | 0.62 | 0.22 | 0.75 | 0.86 | 0.89 | 0.63 | 0.87 | 0.64 | 0.95 | 0.92 | 0.71 | 0.93 | 0.87 | 0.98 | 0.48 | 0.69 | 0.46 | 0.57 | 0.40 | 0.71 |
| Phi-4-mini-instruct | 0.74 | 0.67 | 0.50 | 0.55 | 0.24 | 0.50 | 0.70 | 0.63 | 0.52 | 0.69 | 0.58 | 0.71 | 0.70 | 0.38 | 0.69 | 0.66 | 0.71 | 0.53 | 0.44 | 0.40 | 0.39 | 0.41 | 0.56 |
| Qwen3-8B | 0.86 | 0.79 | 0.64 | 0.61 | 0.23 | 0.25 | 0.85 | 0.89 | 0.66 | 0.87 | 0.64 | 0.95 | 0.93 | 0.73 | 0.94 | 0.88 | 0.98 | 0.52 | 0.73 | 0.48 | 0.56 | 0.42 | 0.70 |
| Mistral-Nemo-Instruct-2407 | 0.85 | 0.78 | 0.79 | 0.61 | 0.25 | 0.70 | 0.88 | 0.91 | 0.77 | 0.85 | 0.64 | 0.94 | 0.90 | 0.79 | 0.95 | 0.77 | 0.97 | 0.45 | 0.76 | 0.47 | 0.55 | 0.40 | 0.73 |

Table 8: Zero-shot classification results on BTZSC by dataset (macro-averaged precision). Abbreviations: AGN = AGNEWS, BF-Off / BF-Sex = BiasFrames (offensive / sex), CAPS = CAPSOTU, MAN = Manifesto, TT = TrueTeacher, WT-Ins / WT-Obs / WT-Thr / WT-Agg = WikiToxic (insult / obscene / threat / toxic aggregated), YT = Yahoo Topics, AmzPol = Amazon Polarity, AppR = AppReviews, FPB = FinancialPhraseBank, RT = RottenTomatoes, Yelp = YelpReviews, B77 = Banking77, BF-Int = BiasFrames (intent), MASS = MASSIVE (intent), EmoD = EmotionDiary, Emp = EmpatheticDialogues.

# E    COMPARISON WITH MTEB

To assess whether our conclusions depend on the specific dataset composition of BTZSC, we re-evaluate the same set of models on the English classification tasks from MTEB v2 (Enevoldsen et al., 2025) (Amazon Counterfactual, MASSIVE Sentiment, MTOP Domain, ToxicConversations, IMDB, TweetSentiment Extraction, Banking77, MASSIVE Intent). Table 9 reports macro-$F_1$ scores, Table 10 accuracies, Table 11 recall, and Table 12 precision. We treat BTZSC and MTEB as two independent evaluators that induce rankings over the same models and compare their behavior both in terms of average performance and the resulting rank orderings.

**Global agreement.**    Across all models, macro-$F_1$ on BTZSC and MTEB is strongly aligned. The Kendall rank correlation coefficient (Kendall & Gibbons, 1990) between the BTZSC and MTEB Avg-$F_1$ rankings is high and positive ($\tau = 0.69$, $p \approx 1.3 \times 10^{-9}$), indicating substantial agreement in how the two benchmarks order models. The family-wise picture mirrors our main BTZSC results: base encoders perform worst, NLI cross-encoders substantially improve over them, modern rerankers and instruction-tuned LLMs are competitive, and contemporary embedding models attain the highest average macro-$F_1$ across both benchmarks. The top reranker, *Qwen3-Reranker-8B*, is the single best model on both BTZSC and MTEB, while smaller rerankers such as *Qwen3-Reranker-0.6B* already outperform all NLI cross-encoders in macro-$F_1$ on both suites.

**Rank consistency within model families.**    When restricting the correlation analysis to individual families, we still observe substantial agreement. For NLI cross-encoders, the BTZSC–MTEB rank correlation is $\tau = 0.64$ ($p \approx 0.0057$), showing that models that are strong on BTZSC tend to remain strong on MTEB. Rerankers show almost perfect concordance ($\tau \approx 1.0$, $p \approx 0.0028$), with both benchmarks inducing essentially the same ordering from the older *ms-marco-MiniLM-L6-v2* up to *Qwen3-Reranker-8B*. Instruction-tuned LLMs also exhibit a sizable positive correlation ($\tau = 0.62$, $p \approx 0.069$), reflecting a consistent picture where very small LLMs perform poorly, and 4–8B models are competitive but do not surpass the best reranker. For embedding models, the correlation is positive but more moderate ($\tau = 0.31$, $p \approx 0.22$): both benchmarks clearly favour modern embeddings (e5, BGE, GTE, Qwen-Embedding) over older baselines, but the fine-grained ordering within this family is somewhat benchmark-dependent.

**Embedding models and dataset composition.**    The embedding family illustrates well how dataset mix shapes absolute scores while leaving the main qualitative conclusions intact. Averaged over the eleven embedding models evaluated on both benchmarks, the family-level macro-$F_1$ is $0.57$ on BTZSC and $0.63$ on MTEB, i.e. MTEB is systematically more "forgiving" to embeddings by roughly six $F_1$ points. This gap is largely explained by task composition.

On BTZSC, embedding models achieve mean task-wise macro-$F_1$ of $0.47$ on topic classification, $0.80$ on sentiment, $0.57$ on intent, and $0.39$ on emotion. On MTEB, grouping the eight datasets into sentiment-like tasks (Amazon Counterfactual, MASSIVE Sentiment, IMDB, TweetSentiment Extraction), intent-like tasks (Banking77, MASSIVE Intent), and topic-like tasks (MTOP Domain, ToxicConversations), the same embedding models reach on average $0.59$ macro-$F_1$ on sentiment, $0.58$ on intent, and $0.74$ on topic. Thus, relative to BTZSC, MTEB exposes embeddings to (i) much easier topic-style problems (mid 0.70s vs. mid 0.40s) and (ii) no emotion tasks at all. Emotion classification is consistently difficult for all families on BTZSC (embeddings around $0.39$; NLI cross-encoders and rerankers around $0.33$-$0.37$), and the absence of this label family in MTEB removes a systematic downward pull on the macro averages. Conversely, MTEB's topic-style datasets (MTOP Domain and ToxicConversations) appear particularly well aligned with embedding-based semantic similarity, yielding high scores that boost the family's average.

Importantly, BTZSC does not uniquely penalise embeddings: on BTZSC they are the strongest family on intent and emotion and competitive on topic and sentiment; MTEB simply provides a task mix—easier topic-style classification, no emotion—in which their strengths are accentuated. The fact that modern embeddings outperform NLI cross-encoders on average and remain a few points below the best reranker holds on both BTZSC and MTEB. What differs is primarily the absolute level at which they plateau (high-0.50s on BTZSC vs. low- to mid-0.60s on MTEB) and the precise ordering among closely matched embedding architectures. These observations indicate that BTZSC not only yields findings that are consistent with MTEBs classification suite, but also

provides a richer testbed for more nuanced performance analysis across tasks and domains, thanks to its explicit coverage of sentiment, topic, intent, and emotion with varying granularities.

| Model | AmzCf | MASS-S | MTOP-D | ToxicConv | IMDB | TweetSentExt | B77 | MASS-I | Avg F1 |
|---|---|---|---|---|---|---|---|---|---|
| **Base encoders** | | | | | | | | | |
| bert-large-uncased | 0.25 | 0.03 | 0.08 | 0.44 | 0.38 | 0.41 | 0.02 | 0.00 | 0.20 |
| deberta-v3-large | 0.51 | 0.02 | 0.07 | 0.49 | 0.33 | 0.31 | 0.01 | 0.00 | 0.22 |
| ModernBERT-large | 0.45 | 0.07 | 0.11 | 0.43 | 0.34 | 0.16 | 0.02 | 0.02 | 0.20 |
| **NLI cross-encoders** | | | | | | | | | |
| bart-large-mnli | 0.15 | 0.54 | 0.84 | 0.50 | 0.93 | 0.54 | 0.28 | 0.41 | 0.52 |
| nli-roberta-base | 0.22 | 0.25 | 0.41 | 0.66 | 0.83 | 0.52 | 0.07 | 0.32 | 0.41 |
| bert-base-uncased-nli | 0.20 | 0.23 | 0.64 | 0.52 | 0.79 | 0.59 | 0.01 | 0.32 | 0.41 |
| bert-large-uncased-nli | 0.42 | 0.40 | 0.42 | 0.47 | 0.80 | 0.61 | 0.08 | 0.38 | 0.45 |
| bert-large-uncased-nli-triplet | 0.38 | 0.31 | 0.49 | 0.53 | 0.78 | 0.61 | 0.06 | 0.37 | 0.44 |
| deberta-v3-base-nli | 0.15 | 0.48 | 0.68 | 0.69 | 0.91 | 0.54 | 0.12 | 0.34 | 0.49 |
| deberta-v3-large-nli | 0.16 | 0.51 | 0.74 | 0.73 | 0.90 | 0.62 | 0.35 | 0.42 | 0.55 |
| deberta-v3-large-nli-triplet | 0.34 | 0.56 | 0.49 | 0.72 | 0.93 | 0.58 | 0.24 | 0.40 | 0.53 |
| modernbert-base-nli | 0.61 | 0.35 | 0.45 | 0.68 | 0.89 | 0.54 | 0.10 | 0.25 | 0.48 |
| modernbert-large-nli | 0.42 | 0.42 | 0.59 | 0.69 | 0.91 | 0.64 | 0.21 | 0.36 | 0.53 |
| modernbert-large-nli-triplet | 0.39 | 0.50 | 0.64 | 0.70 | 0.91 | **0.66** | 0.21 | 0.42 | 0.56 |
| **Rerankers** | | | | | | | | | |
| ms-marco-MiniLM-L6-v2 | 0.53 | 0.08 | 0.19 | 0.53 | 0.62 | 0.32 | 0.24 | 0.05 | 0.32 |
| gte-reranker-modernbert-base | 0.56 | 0.53 | 0.59 | 0.54 | 0.84 | 0.53 | 0.65 | 0.54 | 0.60 |
| bge-reranker-base | 0.39 | 0.44 | 0.73 | 0.56 | 0.69 | 0.53 | 0.48 | 0.43 | 0.53 |
| bge-reranker-large | 0.46 | 0.48 | 0.81 | 0.63 | 0.80 | 0.52 | 0.57 | 0.45 | 0.59 |
| Qwen3-Reranker-0.6B | 0.26 | 0.66 | 0.79 | 0.64 | 0.88 | 0.51 | 0.63 | 0.53 | 0.61 |
| Qwen3-Reranker-8B | 0.46 | **0.77** | 0.80 | 0.74 | **0.95** | 0.62 | 0.69 | **0.67** | **0.71** |
| **Embedding models** | | | | | | | | | |
| all-MiniLM-L6-v2 | 0.15 | 0.48 | 0.63 | 0.42 | 0.34 | 0.40 | 0.43 | 0.33 | 0.40 |
| e5-base-v2 | 0.33 | 0.55 | 0.81 | 0.64 | 0.90 | 0.54 | 0.62 | 0.47 | 0.61 |
| e5-large-v2 | 0.57 | 0.58 | 0.80 | 0.57 | 0.93 | 0.51 | 0.58 | 0.51 | 0.63 |
| e5-mistral-7b-instruct | 0.23 | 0.70 | 0.87 | 0.64 | 0.91 | 0.60 | 0.65 | 0.63 | 0.65 |
| bge-base-en-v1.5 | 0.47 | 0.54 | 0.82 | 0.61 | 0.90 | 0.58 | 0.64 | 0.53 | 0.64 |
| bge-large-en-v1.5 | 0.39 | 0.56 | 0.86 | 0.69 | 0.93 | 0.57 | 0.68 | 0.53 | 0.65 |
| gte-base-en-v1.5 | 0.43 | 0.61 | 0.88 | 0.69 | 0.85 | 0.56 | 0.66 | 0.51 | 0.65 |
| gte-large-en-v1.5 | 0.34 | 0.62 | 0.85 | **0.81** | 0.94 | 0.61 | 0.63 | 0.57 | 0.67 |
| gte-modernbert-base | 0.32 | 0.60 | 0.86 | 0.74 | 0.91 | 0.65 | 0.64 | 0.61 | 0.67 |
| Qwen3-Embedding-0.6B | 0.50 | 0.57 | 0.85 | 0.69 | 0.88 | 0.55 | 0.64 | 0.59 | 0.66 |
| Qwen3-Embedding-8B | 0.23 | 0.70 | **0.91** | 0.72 | **0.95** | 0.50 | **0.71** | 0.65 | 0.67 |
| **Instruction-tuned LLMs** | | | | | | | | | |
| gemma-3-270m-it | 0.34 | 0.03 | 0.05 | 0.47 | 0.47 | 0.24 | 0.00 | 0.00 | 0.20 |
| gemma-3-1b-it | 0.32 | 0.18 | 0.23 | 0.52 | 0.56 | 0.37 | 0.10 | 0.08 | 0.30 |
| Llama-3.2-3B-Instruct | 0.35 | 0.53 | 0.55 | 0.50 | 0.51 | 0.36 | 0.39 | 0.37 | 0.44 |
| Qwen3-4B | **0.80** | 0.65 | 0.78 | 0.46 | 0.92 | 0.61 | 0.37 | 0.38 | 0.62 |
| Phi-4-mini-instruct | 0.35 | 0.53 | 0.56 | 0.47 | 0.56 | 0.35 | 0.42 | 0.28 | 0.44 |
| Qwen3-8B | 0.66 | 0.66 | 0.82 | 0.46 | 0.94 | 0.62 | 0.39 | 0.39 | 0.62 |
| Mistral-Nemo-Instruct-2407 | 0.73 | 0.64 | 0.75 | 0.49 | **0.95** | 0.61 | 0.35 | 0.36 | 0.61 |

Table 9: Zero-shot classification results (macro-averaged F1) on eight English classification datasets from MTEB v2 plus their average (Avg F1). Bold denotes the best and underlining the second-best score in each column. Best model in each family is underlined.

| Model | AmzCf | MASS-S | MTOP-D | ToxicConv | IMDB | TweetSE | B77 | MASS-I | Avg Acc |
|---|---|---|---|---|---|---|---|---|---|
| **Base encoders** | | | | | | | | | |
| bert-large-uncased | 0.25 | 0.05 | 0.15 | 0.53 | 0.52 | 0.41 | 0.03 | 0.01 | 0.25 |
| deberta-v3-large | 0.66 | 0.03 | 0.09 | 0.50 | 0.49 | 0.32 | 0.03 | 0.00 | 0.27 |
| ModernBERT-large | 0.83 | 0.09 | 0.13 | 0.50 | 0.51 | 0.33 | 0.03 | 0.02 | **0.30** |
| **NLI cross-encoders** | | | | | | | | | |
| bart-large-mnli | 0.18 | 0.52 | 0.84 | 0.57 | 0.93 | 0.62 | 0.29 | 0.43 | 0.55 |
| nli-roberta-base | 0.23 | 0.28 | 0.43 | 0.66 | 0.83 | 0.61 | 0.08 | 0.36 | 0.43 |
| bert-base-uncased-nli | 0.21 | 0.21 | 0.62 | 0.58 | 0.79 | 0.64 | 0.02 | 0.28 | 0.42 |
| bert-large-uncased-nli | 0.43 | 0.38 | 0.41 | 0.55 | 0.81 | **0.65** | 0.08 | 0.35 | 0.46 |
| bert-large-uncased-nli-triplet | 0.39 | 0.30 | 0.46 | 0.59 | 0.79 | **0.65** | 0.07 | 0.37 | 0.45 |
| deberta-v3-base-nli | 0.17 | 0.49 | 0.65 | 0.69 | 0.91 | 0.63 | 0.13 | 0.33 | 0.50 |
| deberta-v3-large-nli | 0.18 | 0.54 | 0.75 | 0.73 | 0.90 | 0.67 | 0.36 | 0.44 | **0.57** |
| deberta-v3-large-nli-triplet | 0.34 | 0.57 | 0.51 | 0.72 | 0.93 | 0.64 | 0.24 | 0.42 | 0.55 |
| modernbert-base-nli | 0.68 | 0.36 | 0.47 | 0.68 | 0.89 | 0.62 | 0.09 | 0.28 | 0.51 |
| modernbert-large-nli | 0.43 | 0.40 | 0.59 | 0.69 | 0.91 | 0.66 | 0.24 | 0.39 | 0.54 |
| modernbert-large-nli-triplet | 0.40 | 0.48 | 0.67 | 0.70 | 0.91 | 0.68 | 0.22 | 0.44 | 0.56 |
| **Rerankers** | | | | | | | | | |
| ms-marco-MiniLM-L6-v2 | 0.76 | 0.08 | 0.23 | 0.53 | 0.62 | 0.42 | 0.22 | 0.04 | 0.36 |
| gte-reranker-modernbert-base | 0.67 | 0.54 | 0.60 | 0.59 | 0.84 | 0.62 | 0.63 | 0.57 | 0.63 |
| bge-reranker-base | 0.42 | 0.46 | 0.77 | 0.57 | 0.69 | 0.59 | 0.49 | 0.45 | 0.55 |
| bge-reranker-large | 0.50 | 0.50 | 0.82 | 0.64 | 0.80 | 0.61 | 0.56 | 0.47 | 0.61 |
| Qwen3-Reranker-0.6B | 0.26 | 0.66 | 0.80 | 0.65 | 0.88 | 0.62 | 0.62 | 0.52 | 0.63 |
| Qwen3-Reranker-8B | 0.47 | **0.75** | 0.81 | 0.74 | **0.95** | 0.65 | **0.67** | **0.72** | **0.72** |
| **Embedding models** | | | | | | | | | |
| all-MiniLM-L6-v2 | 0.17 | 0.47 | 0.66 | 0.49 | 0.49 | 0.40 | 0.44 | 0.34 | 0.43 |
| e5-base-v2 | 0.33 | 0.55 | 0.84 | 0.65 | 0.90 | 0.60 | 0.62 | 0.49 | 0.62 |
| e5-large-v2 | 0.66 | 0.58 | 0.83 | 0.61 | 0.93 | 0.59 | 0.56 | 0.51 | 0.66 |
| e5-mistral-7b-instruct | 0.24 | 0.71 | 0.88 | 0.66 | 0.92 | 0.64 | 0.65 | 0.67 | 0.67 |
| bge-base-en-v1.5 | 0.51 | 0.54 | 0.83 | 0.64 | 0.90 | 0.61 | 0.64 | 0.56 | 0.66 |
| bge-large-en-v1.5 | 0.39 | 0.58 | 0.88 | 0.69 | 0.94 | 0.62 | 0.68 | 0.52 | 0.66 |
| gte-base-en-v1.5 | 0.43 | 0.62 | 0.89 | 0.71 | 0.85 | 0.63 | 0.65 | 0.54 | 0.66 |
| gte-large-en-v1.5 | 0.34 | 0.63 | 0.86 | **0.81** | 0.94 | 0.66 | 0.63 | 0.58 | 0.68 |
| gte-modernbert-base | 0.32 | 0.61 | 0.88 | 0.74 | 0.91 | 0.67 | 0.63 | 0.65 | 0.68 |
| Qwen3-Embedding-0.6B | 0.54 | 0.60 | 0.87 | 0.69 | 0.88 | 0.58 | 0.64 | 0.63 | 0.68 |
| Qwen3-Embedding-8B | 0.24 | 0.69 | **0.91** | 0.73 | **0.95** | 0.53 | **0.70** | 0.71 | 0.68 |
| **Instruction-tuned LLMs** | | | | | | | | | |
| gemma-3-270m-it | 0.34 | 0.07 | 0.07 | 0.50 | 0.52 | 0.32 | 0.01 | 0.01 | 0.23 |
| gemma-3-1b-it | 0.33 | 0.20 | 0.27 | 0.54 | 0.61 | 0.43 | 0.14 | 0.09 | 0.33 |
| Llama-3.2-3B-Instruct | 0.35 | 0.54 | 0.55 | 0.52 | 0.54 | 0.44 | 0.41 | 0.37 | 0.46 |
| Qwen3-4B | **0.87** | 0.65 | 0.79 | 0.56 | 0.92 | 0.66 | 0.40 | 0.40 | **0.66** |
| Phi-4-mini-instruct | 0.35 | 0.51 | 0.52 | 0.50 | 0.61 | 0.44 | 0.43 | 0.31 | 0.46 |
| Qwen3-8B | 0.71 | 0.65 | 0.82 | 0.56 | 0.94 | 0.67 | 0.40 | 0.41 | 0.65 |
| Mistral-Nemo-Instruct-2407 | 0.85 | 0.66 | 0.76 | 0.58 | **0.95** | 0.65 | 0.39 | 0.39 | 0.65 |

Table 10: Zero-shot classification results (micro accuracy) on the 8 English MTEB-v2 classification tasks. Bold denotes the best and underlining the second-best score in each column. Best model in each family is underlined.

| Model | AmzCf | MASS-S | MTOP-D | ToxicConv | IMDB | TweetSE | B77 | MASS-I | Avg Acc |
|---|---|---|---|---|---|---|---|---|---|
| **Base encoders** | | | | | | | | | |
| bert-large-uncased | 0.49 | 0.06 | 0.11 | 0.53 | 0.51 | 0.42 | 0.03 | 0.03 | 0.27 |
| deberta-v3-large | 0.52 | 0.05 | 0.08 | 0.50 | 0.50 | 0.32 | 0.03 | 0.00 | 0.25 |
| ModernBERT-large | 0.50 | 0.10 | 0.15 | 0.50 | 0.50 | 0.33 | 0.04 | 0.03 | 0.27 |
| **NLI cross-encoders** | | | | | | | | | |
| bart-large-mnli | 0.50 | 0.55 | 0.84 | 0.57 | 0.93 | 0.62 | 0.30 | 0.43 | 0.59 |
| nli-roberta-base | 0.51 | 0.28 | 0.44 | 0.66 | 0.83 | 0.61 | 0.07 | 0.33 | 0.47 |
| bert-base-uncased-nli | 0.51 | 0.29 | 0.62 | 0.58 | 0.79 | 0.63 | 0.01 | 0.31 | 0.47 |
| bert-large-uncased-nli | 0.57 | 0.39 | 0.43 | 0.55 | 0.81 | 0.65 | 0.08 | 0.36 | 0.48 |
| bert-large-uncased-nli-triplet | 0.60 | 0.33 | 0.48 | 0.59 | 0.78 | 0.65 | 0.07 | 0.43 | 0.49 |
| deberta-v3-base-nli | 0.50 | 0.50 | 0.67 | 0.69 | 0.91 | 0.62 | 0.13 | 0.34 | 0.55 |
| deberta-v3-large-nli | 0.51 | 0.57 | 0.77 | 0.73 | 0.90 | 0.66 | 0.38 | 0.44 | 0.62 |
| deberta-v3-large-nli-triplet | 0.59 | 0.62 | 0.52 | 0.72 | 0.93 | 0.64 | 0.26 | 0.44 | 0.59 |
| modernbert-base-nli | 0.71 | 0.44 | 0.47 | 0.68 | 0.89 | 0.61 | 0.09 | 0.26 | 0.52 |
| modernbert-large-nli | 0.57 | 0.44 | 0.56 | 0.69 | 0.91 | 0.66 | 0.24 | 0.38 | 0.56 |
| modernbert-large-nli-triplet | 0.57 | 0.52 | 0.64 | 0.70 | 0.91 | **0.68** | 0.22 | 0.45 | 0.59 |
| **Rerankers** | | | | | | | | | |
| ms-marco-MiniLM-L6-v2 | 0.52 | 0.13 | 0.23 | 0.53 | 0.63 | 0.41 | 0.22 | 0.06 | 0.34 |
| gte-reranker-modernbert-base | 0.60 | 0.61 | 0.60 | 0.59 | 0.84 | 0.61 | 0.67 | 0.57 | 0.64 |
| bge-reranker-base | 0.49 | 0.54 | 0.75 | 0.57 | 0.70 | 0.58 | 0.51 | 0.53 | 0.58 |
| bge-reranker-large | 0.55 | 0.58 | 0.82 | 0.64 | 0.81 | 0.61 | 0.58 | 0.57 | 0.64 |
| Qwen3-Reranker-0.6B | 0.55 | 0.71 | 0.80 | 0.65 | 0.88 | 0.61 | 0.66 | 0.61 | 0.68 |
| Qwen3-Reranker-8B | 0.67 | **0.80** | 0.82 | 0.74 | **0.95** | 0.65 | 0.72 | **0.74** | **0.76** |
| **Embedding models** | | | | | | | | | |
| all-MiniLM-L6-v2 | 0.50 | 0.58 | 0.66 | 0.49 | 0.51 | 0.40 | 0.44 | 0.39 | 0.50 |
| e5-base-v2 | 0.56 | 0.63 | 0.83 | 0.65 | 0.90 | 0.59 | 0.65 | 0.58 | 0.67 |
| e5-large-v2 | 0.63 | 0.66 | 0.81 | 0.61 | 0.93 | 0.59 | 0.59 | 0.60 | 0.68 |
| e5-mistral-7b-instruct | 0.53 | 0.75 | 0.88 | 0.66 | 0.91 | 0.64 | 0.69 | 0.70 | 0.72 |
| bge-base-en-v1.5 | 0.56 | 0.64 | 0.83 | 0.64 | 0.90 | 0.61 | 0.67 | 0.64 | 0.69 |
| bge-large-en-v1.5 | 0.62 | 0.66 | 0.87 | 0.69 | 0.93 | 0.62 | 0.70 | 0.62 | 0.72 |
| gte-base-en-v1.5 | 0.65 | 0.68 | 0.89 | 0.71 | 0.85 | 0.62 | 0.68 | 0.61 | 0.71 |
| gte-large-en-v1.5 | 0.59 | 0.71 | 0.86 | **0.81** | 0.94 | 0.66 | 0.65 | 0.66 | 0.74 |
| gte-modernbert-base | 0.54 | 0.69 | 0.88 | 0.74 | 0.91 | 0.67 | 0.66 | 0.69 | 0.72 |
| Qwen3-Embedding-0.6B | 0.62 | 0.68 | 0.86 | 0.69 | 0.88 | 0.58 | 0.67 | 0.67 | 0.71 |
| Qwen3-Embedding-8B | 0.54 | 0.78 | **0.92** | 0.73 | **0.95** | 0.53 | **0.74** | 0.72 | 0.74 |
| **Instruction-tuned LLMs** | | | | | | | | | |
| gemma-3-270m-it | 0.49 | 0.07 | 0.08 | 0.50 | 0.52 | 0.31 | 0.01 | 0.01 | 0.25 |
| gemma-3-1b-it | 0.54 | 0.19 | 0.27 | 0.53 | 0.60 | 0.43 | 0.15 | 0.11 | 0.35 |
| Llama-3.2-3B-Instruct | 0.51 | 0.58 | 0.58 | 0.52 | 0.53 | 0.44 | 0.44 | 0.40 | 0.50 |
| Qwen3-4B | **0.85** | 0.69 | 0.80 | 0.56 | 0.92 | 0.65 | 0.42 | 0.43 | 0.67 |
| Phi-4-mini-instruct | 0.54 | 0.54 | 0.52 | 0.50 | 0.60 | 0.43 | 0.45 | 0.29 | 0.49 |
| Qwen3-8B | 0.79 | 0.69 | 0.83 | 0.56 | 0.94 | 0.66 | 0.43 | 0.43 | 0.67 |
| Mistral-Nemo-Instruct-2407 | 0.71 | 0.65 | 0.77 | 0.58 | **0.95** | 0.64 | 0.40 | 0.40 | 0.64 |

Table 11: Zero-shot classification results (macro-averaged recall) on the 8 MTEB (English, v2) classification datasets and their average (Avg Recall). Bold denotes the best and underlining the second-best score in each column. Best model in each family is underlined.

| Model | Amazon C. | Massive S. | MTOP | Toxic | IMDB | Tweet S. | Banking77 | Massive I. | Avg Prec |
|---|---|---|---|---|---|---|---|---|---|
| **Base encoders** | | | | | | | | | |
| bert-large-uncased | 0.49 | 0.04 | 0.11 | 0.60 | 0.55 | 0.46 | 0.02 | 0.00 | 0.28 |
| deberta-v3-large | 0.51 | 0.08 | 0.07 | 0.51 | 0.24 | 0.32 | 0.02 | 0.00 | 0.22 |
| ModernBERT-large | 0.41 | 0.08 | 0.22 | 0.49 | 0.26 | 0.11 | 0.05 | 0.04 | 0.21 |
| **NLI cross-encoders** | | | | | | | | | |
| bart-large-mnli | 0.59 | 0.67 | 0.84 | 0.68 | 0.93 | 0.64 | 0.47 | 0.56 | 0.67 |
| nli-roberta-base | 0.53 | 0.44 | 0.46 | 0.66 | 0.83 | 0.61 | 0.13 | 0.47 | 0.52 |
| bert-base-uncased-nli | 0.53 | 0.34 | 0.76 | 0.65 | 0.80 | 0.67 | 0.03 | 0.54 | 0.54 |
| bert-large-uncased-nli | 0.55 | 0.61 | 0.65 | 0.65 | 0.84 | 0.66 | 0.17 | 0.53 | 0.58 |
| bert-large-uncased-nli-triplet | 0.57 | 0.57 | 0.70 | 0.67 | 0.82 | 0.66 | 0.11 | 0.45 | 0.57 |
| deberta-v3-base-nli | 0.09 | 0.62 | 0.79 | 0.69 | 0.91 | 0.65 | 0.23 | 0.49 | 0.56 |
| deberta-v3-large-nli | 0.59 | 0.61 | 0.79 | 0.74 | 0.91 | 0.69 | 0.45 | 0.57 | 0.67 |
| deberta-v3-large-nli-triplet | 0.59 | 0.65 | 0.64 | 0.72 | 0.93 | 0.68 | 0.35 | 0.55 | 0.64 |
| modernbert-base-nli | 0.63 | 0.37 | 0.63 | 0.69 | 0.89 | 0.64 | 0.20 | 0.39 | 0.55 |
| modernbert-large-nli | 0.55 | 0.64 | 0.67 | 0.69 | 0.92 | 0.68 | 0.29 | 0.47 | 0.61 |
| modernbert-large-nli-triplet | 0.55 | 0.64 | 0.73 | 0.70 | 0.92 | 0.69 | 0.36 | 0.53 | 0.64 |
| **Rerankers** | | | | | | | | | |
| ms-marco-MiniLM-L6-v2 | 0.53 | 0.21 | 0.50 | 0.53 | 0.64 | 0.48 | 0.44 | 0.14 | 0.43 |
| gte-reranker-modernbert-base | 0.56 | 0.59 | 0.72 | 0.65 | 0.86 | 0.64 | 0.68 | 0.58 | 0.66 |
| bge-reranker-base | 0.50 | 0.44 | 0.74 | 0.57 | 0.70 | 0.56 | 0.57 | 0.44 | 0.56 |
| bge-reranker-large | 0.53 | 0.50 | 0.82 | 0.65 | 0.81 | 0.58 | 0.66 | 0.44 | 0.63 |
| Qwen3-Reranker-0.6B | 0.58 | 0.67 | 0.82 | 0.66 | 0.88 | 0.67 | 0.66 | 0.56 | 0.69 |
| Qwen3-Reranker-8B | 0.62 | **0.79** | 0.84 | 0.74 | **0.95** | 0.65 | 0.74 | **0.71** | **0.76** |
| **Embedding models** | | | | | | | | | |
| all-MiniLM-L6-v2 | 0.09 | 0.55 | 0.69 | 0.48 | 0.74 | 0.45 | 0.56 | 0.42 | 0.50 |
| e5-base-v2 | 0.56 | 0.59 | 0.82 | 0.68 | 0.90 | 0.59 | 0.65 | 0.50 | 0.66 |
| e5-large-v2 | 0.58 | 0.61 | 0.82 | 0.67 | 0.93 | 0.57 | 0.70 | 0.55 | 0.68 |
| e5-mistral-7b-instruct | 0.57 | 0.72 | 0.87 | 0.71 | 0.92 | 0.64 | 0.70 | 0.65 | 0.72 |
| bge-base-en-v1.5 | 0.54 | 0.60 | 0.83 | 0.71 | 0.90 | 0.59 | 0.68 | 0.54 | 0.67 |
| bge-large-en-v1.5 | 0.60 | 0.62 | 0.87 | 0.69 | 0.93 | 0.61 | 0.74 | 0.54 | 0.70 |
| gte-base-en-v1.5 | 0.61 | 0.64 | 0.89 | 0.75 | 0.88 | 0.63 | 0.68 | 0.54 | 0.70 |
| gte-large-en-v1.5 | 0.59 | 0.66 | 0.86 | **0.81** | 0.94 | 0.67 | 0.68 | 0.59 | 0.73 |
| gte-modernbert-base | 0.53 | 0.60 | 0.86 | 0.74 | 0.91 | 0.66 | 0.69 | 0.62 | 0.70 |
| Qwen3-Embedding-0.6B | 0.57 | 0.59 | 0.85 | 0.71 | 0.89 | 0.60 | 0.68 | 0.65 | 0.68 |
| Qwen3-Embedding-8B | 0.59 | 0.72 | **0.90** | 0.76 | **0.95** | **0.74** | **0.76** | 0.65 | **0.76** |
| **Instruction-tuned LLMs** | | | | | | | | | |
| gemma-3-270m-it | 0.50 | 0.02 | 0.04 | 0.50 | 0.53 | 0.21 | 0.00 | 0.00 | 0.23 |
| gemma-3-1b-it | 0.53 | 0.25 | 0.40 | 0.54 | 0.67 | 0.41 | 0.18 | 0.20 | 0.40 |
| Llama-3.2-3B-Instruct | 0.50 | 0.56 | 0.59 | 0.52 | 0.54 | 0.43 | 0.54 | 0.46 | 0.52 |
| Qwen3-4B | **0.78** | 0.67 | 0.80 | 0.73 | 0.93 | 0.67 | 0.74 | 0.46 | 0.69 |
| Phi-4-mini-instruct | 0.53 | 0.62 | 0.65 | 0.50 | 0.69 | 0.49 | 0.53 | 0.40 | 0.55 |
| Qwen3-8B | 0.67 | 0.68 | 0.83 | 0.73 | 0.94 | 0.70 | 0.52 | 0.48 | 0.69 |
| Mistral-Nemo-Instruct-2407 | 0.75 | 0.69 | 0.79 | 0.72 | **0.95** | 0.65 | 0.45 | 0.47 | 0.68 |

Table 12: Zero-shot classification results (macro-averaged precision) on BTZSC for the eight MTEB (EN, v2) classification datasets. We report per-dataset precision and overall average precision (Avg Prec). Bold denotes the best and underlining the second-best score in each column. Best model in each family is underlined.

