# OpenReview forum: "BTZSC: A Benchmark for Zero-Shot Text Classification Across Cross-Encoders, Embedding Models, Rerankers and LLMs"
_ICLR.cc/2026/Conference — ICLR 2026 Poster_

### Official Review · Reviewer_tZ9R · 2025-10-30

**Soundness:** 3
**Presentation:** 2
**Contribution:** 3
**Rating:** 4
**Confidence:** 3

**Summary:**

This paper introduces BTZSC, a comprehensive benchmark for evaluating zero-shot text classification performance across three major model paradigms: NLI-based cross-encoders, embedding models, and rerankers. BTZSC unifies 22 English datasets covering sentiment, topic, intent, and emotion classification under a text-label semantic matching framework, where each label is verbalized as a natural-language sentence. Through systematic experiments with 31 models, the paper finds that rerankers achieve the highest zero-shot accuracy, while strong embedding models offer better efficiency-performance trade-offs. The benchmark further analyzes scaling trends, NLI transferability, and latency-accuracy relationships, providing new insights into how different architectures generalize under true zero-shot conditions.

**Strengths:**

1) The benchmark systematically evaluates three major paradigms for zero-shot text classification under a unified evaluation protocol. This breadth of comparison provides valuable insight into how different architectures trade off accuracy, scalability, and cost.
2) The benchmark integrates 22 publicly available datasets spanning multiple task types, which enables a relatively broad assessment of zero-shot performance across heterogeneous domains.

**Weaknesses:**

1) The paper claims to be "true zero-shot", but the 22 datasets used (such as AGNews, IMDb, AmazonPolarity, etc.) are all from public corpora that may have been seen during model pre-training. This means that the model may have indirectly learned the task distribution or category semantics, and strictly speaking, it is impossible to verify the model's migration ability under "completely new tasks".
2) The benchmark relies solely on macro-averaged F1 as the evaluation metric. While this simplifies cross-dataset comparison, it introduces several methodological limitations. F1 alone cannot reveal important trade-offs between precision and recall, nor can it account for task difficulty or class imbalance. Moreover, reranker models are evaluated only on top-1 predictions, ignoring their ranking quality. The absence of per-class variance and statistical significance testing further limits interpretability.

**Questions:**

1) Is the full benchmark publicly released or planned to be open-sourced? If not, could you clarify the reasons or timeline for public availability?
2) How do you ensure that the datasets used were not partially included in the pre-training corpora of the evaluated models (such as BERT, DeBERTa, GTE, E5, or Qwen)?
3) Why did you choose to rely solely on macro-averaged F1, and did you evaluate whether other metrics lead to consistent rankings across models?

---

> ### Author Response · Authors · 2025-11-20
> **Author response to Reviewer tZ9R– initial reply (I/II)**
>
> _The paper claims to be "true zero-shot", but the 22 datasets used (such as AGNews, IMDb, AmazonPolarity, etc.) are all from public corpora that may have been seen during model pre-training. This means that the model may have indirectly learned the task distribution or category semantics, and strictly speaking, it is impossible to verify the model's migration ability under "completely new tasks":_
>
> We agree that, in the large-pretraining regime, we cannot guarantee that our test corpora are completely unseen: the datasets we use (AGNews, IMDb, AmazonPolarity, etc.) are public and may have been partially included in some models’ pre-training data. Our intention with “true zero-shot” was to emphasise that **no labelled examples from BTZSC are used for training or tuning**; models are evaluated purely by matching documents to natural-language label verbalizers, without any supervised adaptation on these tasks.
>
> In the revised version we will (i) make this definition explicit in the introduction, by clearly defining what we mean by “zero-shot” in BTZSC (no fine-tuning or labelled examples from the target tasks; evaluation only via document–label semantic matching), and (ii) add a limitation paragraph noting that, as with other modern benchmarks such as MTEB, we cannot fully exclude pre-training exposure to some underlying corpora. This is a general challenge in the current era of large pre-trained models, but benchmarks remain useful and widely adopted under the explicit understanding that “zero-shot” refers to the absence of *task-specific supervised training*, not guaranteed corpus novelty. For transparency, we will also state that we have checked the publicly documented pre-training corpora and supervised training data for the models we include, and we avoid models that explicitly list any of our datasets as part of their supervised training data, while acknowledging that undocumented overlap may still exist.

---

> > ### Author Response · Authors · 2025-11-20
> > **Author response to Reviewer tZ9R– initial reply (II/II)**
> >
> > _The benchmark relies solely on macro-averaged F1 as the evaluation metric. While this simplifies cross-dataset comparison, it introduces several methodological limitations. F1 alone cannot reveal important trade-offs between precision and recall, nor can it account for task difficulty or class imbalance. Moreover, reranker models are evaluated only on top-1 predictions, ignoring their ranking quality. The absence of per-class variance and statistical significance testing further limits interpretability:_
> >
> > Our design intentionally uses a single primary metric for cross-dataset comparability. As stated in §4, we adopt **macro-F1** as the primary metric because it gives equal weight to each class and is appropriate across binary and multi-class settings with varying label cardinalities, and we already report average (micro) **accuracy** as a complementary headline number (Table 2), with standard deviations across datasets in parentheses.
> >
> > We agree that additional views can be informative, and we will make the evaluation slightly richer in the revision without changing the core protocol:
> >
> > * **Precision/recall trade-offs.** In the single-label setting, macro-F1 already combines macro-precision and macro-recall, but we acknowledge that seeing these components explicitly can be useful. In the revised version we will add macro-precision and macro-recall averages per task family and model family in the appendix, while keeping macro-F1 as the primary headline metric.
> >
> > * **Task difficulty and imbalance.** The current paper already analyses difficulty by task family and label granularity (e.g., Figure 2(a) and the discussion of sentiment vs emotion/intent), and Table 1 reports label counts and domains. To give a more fine-grained view, we will add a full disaggregated results table in the appendix reporting macro-F1 for **every model–dataset combination**, so readers can inspect performance per task in more detail and, using the released code, obtain per-class metrics if desired.
> >
> > * **Reranker ranking quality.** Our focus in BTZSC is *classification*, selecting a single label per example, so we evaluate rerankers in the same top-1 regime as cross-encoders and embeddings to keep families directly comparable. We agree that ranking metrics (e.g., NDCG or top-k accuracy over the label list) are valuable for applications that use rerankers as full label rankers, and we will clarify this scope and point to ranking-based evaluation of BTZSC as a natural extension beyond the present classification focus.
> >
> > * **Significance and variance.** Providing per-class variance and formal significance tests for all 31 models × 22 datasets would be unwieldy in the main text. Instead, we rely on sizeable and consistent gaps between families (e.g., rerankers vs embeddings). By releasing disaggregated scores and code, we enable practitioners to compute per-class statistics and perform detailed significance testing for specific model pairs or tasks where needed.
> >
> > Overall, we see macro-F1 plus accuracy as reasonable primary metrics for BTZSC, with the planned appendix tables and released code offering additional detail, while keeping the benchmark centred on cross-family zero-shot classification rather than on designing a new metric suite. If there are particular aspects of the metric design or additional analyses that you consider crucial for your final recommendation, we would very much appreciate a brief indication so we can address them most directly.

---

> > > ### Comment · Reviewer_tZ9R · 2025-11-26
> > >
> > > Thank you to the authors for their thoughtful and detailed response to my initial review. I have read your proposed revisions and appreciate the concrete steps you have outlined to address the concerns raised.

---

> > > > ### Author Response · Authors · 2025-12-02
> > > > **Reply to tZ9R**
> > > >
> > > > Thank you very much for your careful follow-up and for considering our response. In the revised version, we have implemented the main changes outlined in our rebuttal: clearer exposition of the benchmark construction and evaluation protocol, added LLM-as-classifier baselines, richer and more disaggregated reporting (including additional metrics), and a dedicated comparison to the MTEB-v2 classification subset. We hope these concrete revisions address your initial concerns and make the contribution of BTZSC clearer and more useful to the community.

---

### Official Review · Reviewer_Muat · 2025-10-30

**Soundness:** 2
**Presentation:** 3
**Contribution:** 3
**Rating:** 4
**Confidence:** 3

**Summary:**

This paper introduces BTZSC, a benchmark comprising 22 datasets to evaluate zero-shot text classification performance across three major model families: NLI-based cross-encoders, embedding models, and rerankers. The authors conduct a broad empirical comparison across 31 models and report that rerankers achieve the highest accuracy, while embedding models offer the best efficiency-performance trade-off.

**Strengths:**

The paper systematically covers a wide range of datasets and model types, making it one of the most comprehensive empirical studies on zero-shot text classification to date. The release of the BTZSC benchmark and codebase supports reproducibility and could serve as a useful evaluation tool for the community.

**Weaknesses:**

1. The work lacks technical novelty, where no new algorithm, model, or training strategy is proposed.
2. The core contribution lies in benchmark, which is largely incremental given the existence of MTEB and similar efforts. The experimental insights are mostly intuitive or expected (e.g., rerankers scale better, embedding models are efficient), and analysis is superficial.
3. The evaluation design largely mirrors existing work, such as label verbalization and metric selection, with limited innovation in methodology.
4. The discussion around trade-offs and scaling is descriptive but lacks theoretical depth.

**Questions:**

Please refer to weaknesses regarding the lack of technical contributions and analytical depth.

---

> ### Author Response · Authors · 2025-11-20
> **Author response to Reviewer Muat – initial reply (I/II)**
>
> _The work lacks technical novelty, where no new algorithm, model, or training strategy is proposed:_
>
> We agree that we do not introduce a new algorithm or model; BTZSC is a benchmark and empirical study paper. We see this as analogous to works like MTEB: despite not proposing new architectures, such benchmarks have proved highly valuable for both the research community and the broader ecosystem by standardizing evaluation and revealing strengths and weaknesses of existing methods.
>
> In contrast to MTEB, which focuses primarily on **embedding models** evaluated via **supervised fine-tuning / linear probes**, BTZSC is explicitly designed for **zero-shot classification** and covers a broader set of architectures that are actually used for low-latency classification in practice: NLI cross-encoders, dual-encoder embeddings, and rerankers. We will clarify this positioning more explicitly in the introduction and related-work sections, emphasizing that the intended contribution of BTZSC lies in its benchmark design and empirical scope—unified label-verbalization and scoring across three model families and 22 tasks—rather than in proposing a new model. If you see any specific reframing of the contribution that would better reflect this and influence your recommendation, we would be grateful to know.
>
> _The core contribution lies in benchmark, which is largely incremental given the existence of MTEB and similar efforts. The experimental insights are mostly intuitive or expected (e.g., rerankers scale better, embedding models are efficient), and analysis is superficial:_
>
> We appreciate this perspective and agree that benchmarks must justify their added value carefully. While BTZSC is indeed a benchmark (we do not propose a new model), we believe it is not merely incremental over MTEB and similar efforts.
>
> MTEB is designed primarily for embedding models under supervised evaluation (linear probes / fine-tuning or few-shot adapters on labelled train sets), and its infrastructure does not natively target the pure zero-shot document–label matching regime we study. BTZSC, by contrast, is explicitly built for zero-shot classification across a broader set of architectures used in practice: NLI cross-encoders, dual-encoder embeddings, and rerankers, all under a unified label-verbalization and scoring protocol on 22 tasks spanning topic, sentiment, intent, and emotion. To our knowledge, there is currently no benchmark that systematically compares these three paradigms in this zero-shot label-matching setting with a shared evaluation pipeline.
>
> We agree that some headline findings (e.g., “rerankers are strongest but slower; embeddings are fastest”) are intuitively plausible, but BTZSC quantifies and nuances these intuitions. The main text already highlights, for example, the strong variation in difficulty across task families (near-saturation on sentiment vs substantially lower F1 on fine-grained emotion/intent), the early saturation of embedding models with scale relative to rerankers, and the different behaviour of model families on long vs short texts. In the revised version we will make these patterns more explicit in the discussion and, importantly, we will add a full disaggregated results table in the appendix reporting macro-F1 for every model–dataset combination. This will both document the non-trivial variation across tasks in more detail and enable further analysis by readers, going beyond the aggregated metrics shown in the main tables and figures. If there are particular analyses whose absence you consider decisive for your score, we would greatly appreciate a brief pointer so we can prioritize them.
>
> _The evaluation design largely mirrors existing work, such as label verbalization and metric selection, with limited innovation in methodology:_
>
> This is a fair observation: our evaluation design intentionally builds on established practice rather than introducing new metrics or verbalization schemes. We deliberately reuse standard ingredients (macro-F1 as the primary metric, accuracy as a complement, and well-established label verbalizers) so that BTZSC is directly comparable to prior zero-shot classification work and easy to adopt in existing pipelines.
>
> The intended methodological contribution is therefore not to propose a new metric or prompting scheme, but to **standardize and extend** this evaluation protocol across three model families (NLI cross-encoders, dual-encoder embeddings, rerankers) and a broader, more diverse set of tasks in a purely zero-shot setting. In the revised version we will clarify this design choice more explicitly and emphasize that we see BTZSC’s value in its unified, cross-family evaluation protocol and breadth of coverage, rather than in introducing new evaluation primitives. If there is any specific methodological refinement you regard as critical for your overall recommendation, we would be glad to hear it.

---

> > ### Author Response · Authors · 2025-11-20
> > **Author response to Reviewer Muat – initial reply (II/II)**
> >
> > _The discussion around trade-offs and scaling is descriptive but lacks theoretical depth:_
> >
> > We agree that our treatment of trade-offs and scaling is primarily empirical rather than theoretical; BTZSC is intended as a measurement-oriented benchmark paper, not a theory paper. Our goal is to quantify how different model families behave under a unified zero-shot protocol, rather than to develop a formal scaling theory.
> >
> > That said, we can make the empirical narrative more structured and informative. In the revised version, we will (i) reorganize the scaling section around explicit questions (e.g., how each family’s performance changes with size across task families, where embedding models saturate, and how this relates to training data and supervision), and (ii) more clearly articulate hypotheses linking the observed patterns to differences in supervision signal (NLI vs retrieval vs generic contrastive training) and data curation, while pointing to existing work on scaling laws and representation learning as possible explanations. A full theoretical account is beyond the scope of this benchmark, but we see BTZSC as providing the kind of detailed empirical evidence that such theory could build on, and we will clarify this positioning in the discussion. If there are particular aspects of the trade-off/scaling analysis that you view as decisive for your final score, we would appreciate a short indication so we can address them directly.

---

> ### Comment · Reviewer_Muat · 2025-11-21
>
> Thank you for the detailed clarifications. I acknowledge that I have read both the rebuttal and the reviews from other members of the Reviewers. Here are my suggestions:
>
> 1. Highlight empirical insights more directly and informative (instead in tables) would make your paper contributions easier to appreciate.
> 2. Additionally, more implementation details regarding the construction of the dataset and the evaluation procedure are needed further providing.
>
> Overall, the authors have addressed some of concerns, and I have increased my score to "6: marginally above the acceptance threshold".

---

> > ### Author Response · Authors · 2025-12-02
> > **Reply to Muat**
> >
> > Thank you very much for the thoughtful follow-up and for updating your score. In the revision, we have (i) foregrounded the main empirical insights more explicitly in the narrative of Sections 4–5, highlighting key patterns and trade-offs directly in the text rather than only in tables, and (ii) substantially expanded the implementation details of dataset construction and evaluation: Section 3 with Appendix A now clearly describes how raw datasets become BTZSC tasks and how aggregation works, Appendices A and C further provide a detailed account of preprocessing, verbalizers, splits, tokenization, and the zero-shot scoring protocol for all model families.

---

### Official Review · Reviewer_yjwX · 2025-10-31

**Soundness:** 3
**Presentation:** 3
**Contribution:** 3
**Rating:** 6
**Confidence:** 4

**Summary:**

The paper proposes BTZSC, a benchmark for true zero‑shot text classification that compares three families: NLI‑tuned cross‑encoders, embedding models, and rerankers; over 22 English datasets spanning topic, sentiment, intent, and emotion. The primary metric is macro‑F1 (with micro accuracy also reported). The study evaluates 31 models (public and custom), analyzes scaling and the NLI to ZSC transfer, and reports that: (i) rerankers, notably Qwen3‑Reranker‑8B, achieve the highest overall performance (macro‑F1 = 0.72), (ii) strong embeddings (e.g., GTE‑large‑en‑v1.5) “close the gap” and tend to offer the best accuracy–latency trade‑off, and (iii) NLI cross‑encoders are plateauing while scaling primarily benefits rerankers. See Abstract; dataset suite and selection in §3, Table 1 (p. 4), Figure 1 (p. 5); model families in §3.2 (pp. 5–6) and Table 4 (p. 15); results in Table 2 (p. 9); and analyses in Figure 2 (p. 8) and Figure 3 (p. 8).

**Strengths:**

- Representative, diverse suite. 22 datasets across four task families; Table 1 and Figure 1 quantify class cardinalities, lengths, and lexical overlap (pp. 4–5).
- Clean zero‑shot protocol across families. Unified scoring: NLI log‑odds entailment; embedding cosine; reranker yes/no token probability with a fixed template (§4, Appendix C.3).
- Clear headline results. Qwen3‑Reranker‑8B tops macro‑F1 (0.72); GTE‑large‑en‑v1.5 is competitively high among embeddings; Figure 2(b) shows reranker‑favoring scaling; Figure 3(a) visualizes the speed–accuracy frontier (p. 8; p. 9).
- NLI to ZSC analysis. Figure 3(b) offers a useful lens on transfer for cross‑encoders and contrasts to embeddings/rerankers (p. 8).

**Weaknesses:**

1. No zero‑shot LLM‑as‑classifier baseline. The Introduction acknowledges the approach but omits it from evaluation; 8B–32B LLMs are feasible on the stated hardware and would complete Figure 3(a) (pp. 2, 8, 10).
2. Embedding template opacity. §4 lacks per‑model instruction/template details for E5/GTE/BGE/Qwen‑Embedding; performance is often template‑sensitive (p. 6). These models can be instructed.
3. Verbalizer multiplicity untested. Single label verbalization per class; no template‑averaging/paraphrase robustness (p. 6; Appendix A notes reusing Laurer et al. 2023 verbalizers). Authors can consider zero-shot augmentations on verbalizers, with a L2-normalization of the average embedding of augmentations like CLIP, or a k-NN approach where majority voting across different augmented forms of verbalizers is used.
4. Minor inconsistencies. banking77 listed as 77 classes in text (p. 3) vs 72 in Table 1 (p. 4); and the “second‑highest overall 0.64” prose refers to accuracy after defining macro‑F1 as primary (pp. 4, 7, 9).
5. For completion, it would be interesting to verify whether the conclusions of this paper are reproducible using the same models and evaluation methodology on the classification tasks of MTEB instead. However, such an analysis is missing.

**Questions:**

Aren't the used reranking models cross-encoders?

---

> ### Author Response · Authors · 2025-11-20
> **Author response to Reviewer yjwX – initial reply (I/II)**
>
> _Zero-shot LLM-as-classifier baseline (Fig. 3(a)):_
>
> We agree that including LLM-as-classifier baselines would make the comparison more complete and better connect our encoder-focused results to current practice. Our primary focus in BTZSC is on encoder-based architectures (NLI cross-encoders, dual-encoder embeddings, and rerankers), which remain the dominant choice in settings with strict latency, cost, and on-prem deployment constraints. Nevertheless, generative LLMs are an important complementary family for zero-shot classification.
>
> In the revised version, we **will add zero-shot LLM-as-classifier baselines** with parameter counts comparable to our largest models (≈8B), evaluated under the same label-verbalization protocol. Concretely, we will prompt each model with a fixed instruction to choose exactly one label from the BTZSC label set and will evaluate macro-F1, accuracy, and average latency. We will add these results to the speed–accuracy frontier (Figure 3(a)), so that LLMs appear alongside rerankers and embedding models. This will clarify when reasonably sized generative models match or exceed encoder-based methods in accuracy, and when encoder-style models remain preferable due to efficiency and deployment constraints. If you see any particular LLM configuration as decisive for your final score, we would appreciate guidance so we can include it.
>
> _Embedding template opacity. §4 lacks per-model instruction/template details for E5/GTE/BGE/Qwen-Embedding; performance is often template-sensitive (p. 6). These models can be instructed:_
>
> Thank you for pointing this out. We agree that template choices are important for reproducibility. In the current draft, Appendix A only states that we reuse the label verbalizers from Laurer et al. (2023), and Appendix C.3 fully specifies the Qwen3-Reranker prompt, but we do not make the instruction/template strings for the embedding models sufficiently explicit.
>
> In the revised version, we will (i) add a short paragraph to §4 clarifying that for each embedding family (E5, BGE, GTE, Qwen-Embedding) we follow the official instruction templates recommended in the original papers and model cards, and (ii) include a table in the appendix listing the exact input templates used for texts and label verbalizers for each model family. This will remove ambiguity and make our embedding setup fully reproducible. If there are specific template details you consider critical for reproducibility or for your recommendation, we would welcome a brief indication so we can highlight them.
>
> _Verbalizer multiplicity untested. Single label verbalization per class; no template-averaging/paraphrase robustness (p. 6; Appendix A notes reusing Laurer et al. 2023 verbalizers). Authors can consider zero-shot augmentations on verbalizers, with a L2-normalization of the average embedding of augmentations like CLIP, or a k-NN approach where majority voting across different augmented forms of verbalizers is used:_
>
> We agree that robustness to label verbalization is an important question. In the current draft we deliberately reuse the single verbalizer set from Laurer et al. (2023), which has already been adopted in several prior ZSC works, but we do not explore alternative verbalizations.
>
> In the revised version, we will add a small ablation study to probe this sensitivity: for a representative embedding model and a subset of BTZSC tasks (covering topic, sentiment, and fine-grained labels), we will construct several paraphrased verbalizers per class and compare (i) our current single-verbalizer setup and (ii) a CLIP-style variant that uses the L2-normalized average embedding of multiple verbalizers per label, as suggested. We will report the change in macro-F1 and the stability of model rankings across these variants, and summarize the findings in the main text. A more exhaustive exploration of k-NN voting and larger paraphrase sets is interesting but beyond the scope of this benchmark paper; we view it as a natural direction for follow-up work on verbalizer learning and robustness. If there is a particular robustness analysis you would consider especially impactful for your final recommendation, we would be glad to focus on it.

---

> > ### Author Response · Authors · 2025-11-20
> > **Author response to Reviewer yjwX – initial reply (II/II)**
> >
> > _Minor inconsistencies. banking77 listed as 77 classes in text (p. 3) vs 72 in Table 1 (p. 4); and the “second-highest overall 0.64” prose refers to accuracy after defining macro-F1 as primary (pp. 4, 7, 9):_
> >
> > Thank you for pointing out these inconsistencies. In the revised version we will correct the Banking77 class count so that the text and Table 1 are fully consistent, and we will rephrase the sentence about the “second-highest overall 0.64” to state explicitly that this value refers to average **accuracy**, while macro-F1 remains our primary headline metric. These inconsistencies do not affect any of the results or conclusions, but they should indeed be fixed and we will do so in the updated version.
> >
> > _For completion, it would be interesting to verify whether the conclusions of this paper are reproducible using the same models and evaluation methodology on the classification tasks of MTEB instead. However, such an analysis is missing:_
> >
> > BTZSC is indeed inspired by MTEB-style evaluations. Several of our tasks are drawn from the same underlying sources as the MTEB classification track (e.g., Banking77, IMDb, MASSIVE, Amazon reviews), but BTZSC is deliberately more extensive and tailored to the zero-shot setting: it comprises 22 single-label classification tasks across topic, sentiment, intent, and emotion, and evaluates three model families (NLI cross-encoders, embedding models, rerankers) under a unified label-verbalization protocol.
> >
> > In the revised version we will clarify this relationship in the related-work section and position BTZSC as complementary to MTEB: MTEB is designed primarily for embedding models under supervised evaluation (linear probes / fine-tuning on labelled train sets), whereas BTZSC targets a pure zero-shot document–label matching regime across multiple model paradigms. A full re-run of our methodology on the MTEB classification tasks would be an interesting extension but is beyond the scope of this paper. If you feel that any more explicit comparison to MTEB is crucial for your recommendation, we would appreciate a short note so we can address it in the discussion.
> >
> > _Aren't the used reranking models cross-encoders?_
> >
> > Yes, the reranking models we use are cross-encoder architectures, as explicitly mentioned in line 071. In our terminology, “NLI cross-encoders” and “rerankers” are distinguished not so much by architecture as by **training signal and data curation**: NLI models are fine-tuned on natural-language inference datasets and applied to document–label pairs, whereas rerankers are trained as relevance models on query–document pairs using ranking losses and carefully curated hard negatives. In the revised version we will make this distinction clearer and adjust the wording around “model families” so that it does not suggest architectural differences where there are none.

---

> > > ### Comment · Reviewer_yjwX · 2025-11-22
> > >
> > > Thank you very much for your rebuttal. I am looking forward to see the revisions suggested being implemented in the manuscript, or if not possible, then in the comments here. Regarding the MTEB point, I would appreciate to see whether the conclusions of the study still hold true for the classification subset of MTEB(v2) which is pretty small and should run very fast.

---

> > > > ### Author Response · Authors · 2025-12-02
> > > > **MTEB comparison**
> > > >
> > > > Thank you again for the helpful suggestion. We have now run our full zero-shot evaluation protocol on the English classification subset of MTEB(v2) and added the results to the revised manuscript (Appendix E, Tables 9–12). Qualitatively, the main conclusions of the paper continue to hold on MTEB: rerankers remain strongest in pure accuracy, strong embeddings and some smaller LLMs offer the best efficiency–accuracy trade-offs, and NLI-style cross-encoders plateau earlier. We also discuss where BTZSC provides additional coverage beyond the relatively small and less diverse MTEB classification subset.

---

### Official Review · Reviewer_9fKe · 2025-11-01

**Soundness:** 3
**Presentation:** 3
**Contribution:** 2
**Rating:** 4
**Confidence:** 3

**Summary:**

The paper proposes a comprehensive benchmark for Zero-Shot Text Classification including 22 datasets, with a focus on comparing three types of approaches: NLI, sentence transformers, and re-rankers. The proposed benchmark is designed to cover task diversity, class granularity, diversity of text domain and length. The paper proposes a primary metric, Micro F1, and the use of NLI performance as a proxy for zero-shot generalization capabilities. Empirical results indicate reranker models achieve the highest overall accuracy, while strong embedding models offer the most favorable balance between speed and accuracy.

**Strengths:**

I liked the proposal of a comprehensive evaluation dataset and the clear explanations of the three zero-shot approaches: NLI, sentence transformers, and re-rankers. The evaluation is systematic and includes many model variants. I also found the analysis of scaling across model sizes informative.

**Weaknesses:**

1. Given this is a dataset/benchmark paper, I'm surprised by how little detail it offers about the construction of the dataset and evaluation procedure. I am having a hard time connecting the underlying datasets and how the final BTZSC evaluation procedure works.
2. The paper primarily focus on encoder-based architectures, I wonder how do generative models perform on such tasks, and would a reasonably sized generative model effectively solve these tasks? What distinct value does this benchmark provide relative to generative models?

**Questions:**

1. How are the datasets aggregated? Are they not directly aggregated or the labels are aggregated by domain?
2. I wonder how do generative models perform on such tasks, and would a reasonably sized generative model effectively solve these tasks? What distinct value does this benchmark provide relative to generative models?
3. Given this dataset, what are some challenges in existing methods and promising directions for future work?

---

> ### Author Response · Authors · 2025-11-20
> **Author response to Reviewer 9fKe – initial reply (I/II)**
>
> _Given this is a dataset/benchmark paper, I'm surprised by how little detail it offers about the construction of the dataset and evaluation procedure. I am having a hard time connecting the underlying datasets and how the final BTZSC evaluation procedure works_:
>
> Thank you for pointing this out. We agree that the current version could explain the end-to-end pipeline from the underlying datasets to the BTZSC evaluation more clearly.
>
> BTZSC treats each of the 22 datasets as a separate single-label classification task. For each dataset, we use the official test split (or the standard Hugging Face split), standardize each example to a single text field and categorical label, map labels to natural-language verbalizers following Laurer et al. (2023), and evaluate all models purely in zero-shot mode on these test sets. For every dataset we compute macro-F1 and accuracy, and then report the unweighted average across datasets (overall and per task family), as shown in Table 2.
>
> In the revised version we will:
>
> * add a dedicated subsection *“From raw datasets to BTZSC tasks”* in §3 that summarizes the construction and aggregation protocol step-by-step;
> * introduce a short *“Zero-shot scoring protocol”* paragraph in §4 that explicitly describes how NLI cross-encoders, embedding models, and rerankers—and, in the revised version, generative LLMs—are applied to each dataset under a unified document–label matching framework; and
> * extend Appendix A with a table listing, for each dataset, the split used, the text and label fields, and any preprocessing (e.g., truncation).
>
> We believe these additions will make the benchmark construction and evaluation procedure much clearer. If there are aspects of this clarification that you view as particularly decisive for your final recommendation, we would appreciate a brief indication so we can prioritize them in the revision.
>
> _The paper primarily focus on encoder-based architectures, I wonder how do generative models perform on such tasks, and would a reasonably sized generative model effectively solve these tasks? What distinct value does this benchmark provide relative to generative models?_
>
> We intentionally focus BTZSC on encoder-based architectures (NLI cross-encoders, dual-encoder embeddings, and rerankers) because these remain the dominant choice in settings where throughput, latency, cost, and on-prem deployment constraints make large generative LLMs difficult to use as general-purpose classifiers. In such scenarios, it is often infeasible to call a 30–70B LLM for every document–label pair, while encoder pipelines with cached label embeddings or NLI heads are still widely deployed.
>
> That said, we agree that it is important to situate these families relative to generative LLMs. In the revised version, we will therefore add zero-shot LLM-as-classifier baselines with parameter counts comparable to our largest models (≈8B), evaluated under the same label-verbalization protocol. Concretely, we will prompt each model with a fixed instruction to choose exactly one label from the BTZSC label set, and we will report macro-F1, accuracy, and average latency on the 22 datasets. We will integrate these points into the speed–accuracy frontier in Figure 3(a), so that readers can directly compare encoder-based models and reasonably sized generative LLMs in terms of both accuracy and efficiency. This will clarify both the value of BTZSC for encoder-style methods and the regimes in which LLMs are or are not competitive under realistic deployment constraints. If there are specific LLM settings you consider especially informative for your recommendation, we would be grateful to know so we can focus our additions accordingly.

---

> ### Author Response · Authors · 2025-11-20
> **Author response to Reviewer 9fKe – initial reply (II/II)**
>
> _How are the datasets aggregated? Are they not directly aggregated or the labels are aggregated by domain?_
>
> BTZSC does **not** aggregate examples or labels across datasets. Each of the 22 datasets in Table 1 is treated as a separate single-label classification task with its original label set. We standardize each example to a text field $x$ and categorical label $y \in \mathcal{Y}_D$, and map each label $y$ to a natural-language verbalizer $v(y)$. Models are evaluated in zero-shot mode **separately on each dataset**, and we compute macro-F1 and accuracy per dataset.
>
> For reporting, we then take the **unweighted average** of these per-dataset scores, either
>
> * across all 22 datasets (overall score), or
> * within each task family (topic, sentiment, intent, emotion) for the analyses in §4.
>
> We never merge label spaces across datasets or aggregate labels “by domain”; the topic/sentiment/intent/emotion grouping is used only for stratified reporting. In the revised version, we will make this aggregation protocol explicit in a dedicated paragraph in §3 and in an extended table in Appendix A. If you believe any particular clarification around aggregation is critical to your overall assessment, we would appreciate a short note so we can address it explicitly.
>
> _Given this dataset, what are some challenges in existing methods and promising directions for future work?_
>
> Thank you for this question. We agree that BTZSC is useful not only as a benchmark, but also as a lens on the limitations of current zero-shot methods. In the revised version we will add a short *“Challenges and future directions”* paragraph in the conclusion that draws out three main lessons from our results:
>
> * **Fine-grained and subjective tasks remain difficult.**
>   While sentiment and coarse topic classification are close to saturation for the best models, performance on emotion and fine-grained intent tasks is substantially lower (cf. Table 2 and Fig. 2(a)). This suggests a need for methods that better handle subtle, overlapping label semantics, for example via architectures or objectives tailored to fine-grained affect/stance, or via richer label descriptions/definitions.
>
> * **Scaling alone is not enough; an efficiency–accuracy gap persists.**
>   Our scaling analysis shows that rerankers benefit most from increased model size and achieve the highest macro-F1, but at a significant latency cost, whereas embedding models plateau earlier but offer much better throughput (Fig. 2(b), Fig. 3(a)). This points to hybrid and distillation-based approaches as promising directions: approximating reranker-level accuracy with dual-encoder-level efficiency, or designing adaptive pipelines that only invoke heavy rerankers on difficult examples, similar to how they are typically used to rerank top-K candidates in IR settings.
>
> * **Substantial dataset-level heterogeneity and robustness gaps.**
>   Even within a single task family, our per-dataset scores show large variance: the strongest models are close to saturated on some review-style sentiment datasets, yet still struggle on social-media, political, or banking-intent datasets, as well as on high-cardinality label spaces. This indicates that current ZSC methods remain brittle to domain shift, label cardinality, and annotation artefacts. In the revised version we will make this heterogeneity more visible by adding a disaggregated results table in the appendix (macro-F1 for every model–dataset pair) and by highlighting that BTZSC motivates methods that explicitly target domain-robust ZSC and more fine-grained analyses as a function of domain, document length, or label frequency.
>
> We will integrate these points into the conclusion to clarify what BTZSC reveals about the challenges faced by current zero-shot classifiers and to highlight concrete methodological directions beyond extending the benchmark itself. If there are particular challenges or future directions you regard as especially important for your recommendation, we would be glad to know so we can emphasize them clearly.

---

> > ### Comment · Reviewer_9fKe · 2025-11-25
> > **Thanks for the response**
> >
> > I'd like to thank the authors for the response. I'm looking forward to the results of generative models.
> >
> > I'm also wondering if there are inherent label noise in existing datasets which could complicate evaluation—maybe insights from inspecting samples where models (especially generative) and label disagree.

---

> > > ### Author Response · Authors · 2025-12-02
> > > **Potential label noise**
> > >
> > > Thank you for the follow-up and for your interest in the generative models. The latest results have now been integrated into the revised manuscript, and the LLM baselines are fully included in the main tables and figures.
> > >
> > > Regarding label noise: we took your suggestion seriously and ran a small qualitative inspection of disagreement cases across several datasets, including examples where generative models and the gold labels diverge. We do indeed observe some level of semantic ambiguity in a few datasets. For instance, in AGNews we find items labeled as “Sci/Tech” whose content is arguably also close to “Business,” and similar borderline cases between “Business” and “World” news. In some intent and topic datasets, a given input could plausibly fit more than one label when judged purely on semantics.
> > >
> > > We emphasize that all BTZSC datasets are public, widely used NLP benchmarks, and we use them “as is,” which makes it especially interesting to see these imperfections in such standard resources. In small-scale experiments where we removed or relabeled the most clearly problematic items, our main conclusions and model-family rankings remained unchanged (though we cannot claim to have exhaustively corrected all noisy examples). A more systematic study of label noise and ambiguous cases, especially for fine-grained or subjective labels, remains an important direction for future work, both within and beyond BTZSC.

---

### Author Response · Authors · 2025-12-02
**Final Revisions and Author Response Summary (I/III)**

Thank you to the reviewers for the constructive feedback and for the updated scores during the discussion phase. Thank you also to the AC for taking the time to read the rebuttal and revised manuscript. Since only the AC will see the final version, we briefly summarize how we addressed the main concerns across all reviews and where we consciously limited scope.

---

**1. Dataset construction and evaluation protocol**
*Concern: Insufficient detail on how raw datasets become BTZSC tasks and how evaluation/aggregation is performed (R9fKe, RMu, RtZ9R).*

We substantially clarified how BTZSC is built and how evaluation is carried out:

* §3 (“Benchmark for Textual Zero-Shot Classification (BTZSC)”) now explicitly defines zero-shot in BTZSC as: no fine-tuning or labelled examples from BTZSC tasks are used for training or model selection; models are evaluated only via document–label semantic matching. We also clarify that in this context “zero-shot” refers to the absence of task-specific supervised training, not guaranteed corpus novelty.
* §3 has been rewritten to clearly describe the benchmark design: task diversity, label cardinality, domain diversity, document length range, and the fact that BTZSC comprises 22 English single-label tasks. Table 1 and Figure 1 remain as high-level summaries.
* Appendix A (“Datasets Overview”) has been substantially expanded:

  * §§A.1–A.3 describe the instance format and splits, label verbalizers, tokenization and truncation, and the evaluation/aggregation protocol. We now spell out that each dataset is treated as a separate single-label task, that we standardize to a single text field and categorical label per example, and that all metrics are computed per dataset before aggregation.
  * §§A.4–A.5 document sources and licenses and provide short descriptions for every dataset, including domain, annotation scheme, and task type.
* Appendix C (“Experimental Setup”) has been expanded to explicitly describe the zero-shot scoring protocol for all model families (NLI cross-encoders, embedding models, rerankers, and LLMs), and §4 now points to these details.

This directly addresses the concern that the benchmark construction and evaluation pipeline were hard to follow in the original draft and makes the end-to-end pipeline, from raw Hugging Face datasets to BTZSC tasks and aggregated metrics, much more transparent.

---

**2. LLM-as-classifier baselines**
*Concern: Missing comparison to generative models; unclear positioning relative to LLMs (R9fKe, RyjwX).*

To better situate encoder-based models relative to generative LLMs:

* We added zero-shot LLM-as-classifier baselines with parameter counts comparable to our largest models (≈8–12B), plus smaller models. In total, 7 LLMs from different families and sizes (2 <1B, 3 ~4B, 2 in the 8–12B range) are now included.
* All LLMs are evaluated under the same label-verbalization protocol, with a fixed instruction to choose exactly one label from the BTZSC label set.
* We updated:

  * Table 2 (main benchmark table),
  * Figure 2(b) (scaling behaviour, now including LLMs),
  * Figure 3(a) (speed–accuracy frontier, now with LLM points),
  * Figure 3(b) (NLI–ZSC transfer, with LLMs where applicable),
    and revised the narrative in §4–5 accordingly.
* Appendix C now documents in detail the LLM evaluation protocol, including prompts and inference settings.

This directly addresses the “missing zero-shot LLM baseline” weakness and allows a more complete comparison of encoder-style models vs reasonably sized LLMs under realistic latency constraints.

---

**3. Reproducibility: templates, metrics, and disaggregated scores**
*Concern: Template opacity for embeddings; macro-F1 alone seen as too limited; lack of detailed per-dataset results (RyjwX, RtZ9R).*

To improve transparency and reproducibility:

* **Instruction templates.** §4 now states that for each embedding family (E5, BGE, GTE, Qwen-Embedding, etc.) we follow the official instruction templates from the original papers/model cards. Appendix C was expanded with the exact input templates used for more complex models.
* **Metrics.** Macro-F1 remains the primary headline metric, with average micro-accuracy as a complementary overall score. To provide more detail without changing the core protocol:

  * Appendix D (“Disaggregated results”) and Tables 5–8 report macro-F1, macro-precision, macro-recall, and micro-accuracy for every model–dataset combination, so readers can inspect per-task behaviour and, using the released code, compute additional per-class statistics if needed.
  * Appendix D also includes an analysis at a more granular level, with a focus on precision–recall trade-offs across model families and task types.

This addresses concerns about template opacity and the perceived limitations of reporting only macro-F1 in the main text.

---

> ### Author Response · Authors · 2025-12-02
> **Final Revisions and Author Response Summary (II/III)**
>
> **4. Relation to MTEB and cross-benchmark validation**
> *Concern: Perceived incremental contribution over MTEB; no evidence that conclusions transfer to MTEB classification tasks (RyjwX, RMu).*
>
> The original draft positioned BTZSC conceptually relative to MTEB but did not validate whether our conclusions transfer. In the revision we did both:
>
> * In the related-work section, we now more explicitly position BTZSC as complementary to MTEB:
>
>   * **MTEB** classification tasks are primarily used to evaluate **embedding models** under **supervised** fine-tuning / linear probes.
>   * **BTZSC** is explicitly designed for **zero-shot document–label matching** across **four model families** (NLI cross-encoders, embeddings, rerankers, LLMs), under a unified verbalizer-based protocol.
> * In response to the reviewer’s follow-up request, we added an experiment on the **MTEB (English, v2) classification subset**:
>
>   * Appendix E (“Comparison with MTEB”) and Tables 9–12 report disaggregated results for all 38 models included in the main body on the MTEB-v2 classification tasks under our zero-shot protocol.
>   * We find that the main qualitative conclusions of the paper (relative ordering of model families, speed–accuracy trade-offs) hold on MTEB classification tasks as well as on BTZSC, and we provide a detailed discussion of similarities and differences in Appendix E.
>   * We also find that BTZSC’s curated task suite is more well-rounded to probe zero-shot classification performance than MTEB’s classification subset, which is more limited in size and diversity. This makes BTZSC more informative for practitioners interested specifically in zero-shot classification. Moreover, the current MTEB infrastructure does not natively support any zero-shot protocol or non-embedding model families.
>
> This both clarifies BTZSC’s contribution beyond MTEB and empirically checks how our conclusions hold on MTEB classification datasets.
>
> ---
>
> **5. Empirical analysis, scaling behaviour, and future directions**
> *Concern: Analysis seen as mostly descriptive; trade-off and scaling discussion could be more structured and informative (R9fKe, RMu).*
>
> We improved the structure and explicitness of the empirical narrative:
>
> * The scaling and trade-off analysis (§§4–5) has been reorganized around clearer questions (how each family scales with size, where embeddings saturate vs rerankers, differences across task families).
> * We more clearly link observed patterns to differences in supervision signal (NLI vs retrieval vs generic contrastive vs instruction-tuned LLMs) and data curation, while keeping the paper empirical rather than theoretical.
> * We more explicitly integrate three main lessons:
>
>   * Fine-grained and subjective tasks (emotion, fine-grained intent) remain substantially harder than sentiment/coarse topic.
>   * An efficiency–accuracy gap persists: scaling especially benefits rerankers but at high latency cost, while embeddings and some LLMs offer better trade-offs.
>   * There is substantial dataset-level heterogeneity and brittleness to domain shift, label cardinality, and domain (banking, political, social media).
>
> Together with the disaggregated tables in the appendix, this makes the empirical contribution more explicit and easier to interpret.
>
> ---
>
> **6. Verbalizer multiplicity and robustness**
> *Concern: Single verbalizer per label; no study of multi-verbalizer / CLIP-style averaging robustness (RyjwX).*
>
> The reviewer suggested probing robustness to label verbalization using multi-verbalizer / CLIP-style averaging schemes. We agree this is an important direction. During the revision:
>
> * We ran exploratory experiments with multiple paraphrased verbalizers per class and CLIP-style averaging for a representative embedding model and several datasets.
> * We found performance to be quite sensitive to the exact paraphrases, their number, and the specific model–dataset combination, making it difficult to present a concise and robust story within the current page and complexity budget.
> * To avoid introducing a new, highly design-dependent protocol variant into the main benchmark, we decided **not** to include these preliminary results in the paper and instead:
>
>   * Make our single-verbalizer choices fully explicit and reproducible (via the new template documentation).
>   * Treat multi-verbalizer schemes (CLIP-style averaging, k-NN over paraphrases) as an avenue for future work on verbalizer learning and robustness, rather than part of the core BTZSC protocol.
>
> We hope this trade-off is acceptable: we acknowledge the importance of the suggestion, experimented with it, but ultimately chose to keep BTZSC’s evaluation protocol simple and stable.

---

> > ### Author Response · Authors · 2025-12-02
> > **Final Revisions and Author Response Summary (III/III)**
> >
> > **7. Terminology and minor inconsistencies**
> > *Concern: Inconsistencies in dataset statistics and metric wording; potential confusion over “rerankers vs cross-encoders” (RyjwX, RtZ9R).*
> >
> > Finally, we fixed several smaller issues, including the Banking77 class count, the wording around the ‘0.64’ accuracy vs macro-F1, and clearer terminology to explain that rerankers are cross-encoder architectures trained with retrieval-style supervision.
> >
> > ---
> >
> > Overall, the revised manuscript (plus appendices) implements the main revisions promised in the rebuttal: clearer benchmark construction and protocol, added LLM baselines, richer and more reproducible reporting, an explicit MTEB-v2 comparison, and a sharper discussion of limitations and future directions. Where we chose not to fully pursue a suggested direction (multi-verbalizer ablations), we did so consciously to keep the benchmark protocol simple and stable, and we explicitly frame this as future work.

---

### Meta-Review · Area_Chair_wJBD · 2026-01-08

**Summary:**

The paper sets up a new benchmark for zero-shot text classification. The authors say that no such benchmark exists, and set up a benchmark to compare the performance of various text embedding models, generative LLMs, specialized reranker models, etc.

Most reviewers liked the paper and appreciated the need for such a benchmark. Some criticisms were: (1) not enough methodological details; (2) lack of generative LLMs in the comparison; (3) only single metric used for evaluation; (4) absence of MTEB classification datasets.

**Reviewer Concerns:**

The authors did a really thorough job to address the comments and uploaded a revised manuscript. They addressed all four concerns that I listed above: added details, ran experiments with generative LLMs, added further metrics, ran experiments on MTEB datasets. All reviewers participated in the discussion and welcomed the authors' plans to address the criticisms.

I read the revised version of the manuscript and liked it. My main concern is: line 182 says that the datasets overlap "to a large extent" with the ones used by Laurer et al. 2023. This sounds like the datasets here were basically taken from that paper, which raises the question to what extent this is a new benchmark. I feel the authors have to give more details about what "to a large extent" means, and clarify why this should be seen as a NEW benchmark.

Another concern: what are the standard deviations over in Table 2? This is not explained. I suspect they are over datasets included into each group. But if so, I am not sure they are meaningful to report. Datasets are very different and have very different F1 values. Averaging them makes sense, but reporting std is misleading, because it is not really the uncertainty of the final score. If you do think it is the uncertainty of the final score, then almost no comparison would be statistically significant as stds are HUGE!! This would destroy the paper. If you do NOT think that this reflects the uncertainty of the final score, then maybe rather remove these values. Or at least be very clear about what they mean. Also, I would suggest to make everything bold that is within some margin, e.g. 0.02, of the best value. E.g. I am sure 0.51 and 0.50 are not meaningfully different.

Minor comments: line 165 -- why does this incur higher computational costs? Line 205 -- what are standard NLI benchmarks used here? Figure 3b: what is "CLF score" on the y axis, should it be "BTZSC score"?

**Reviewer Scores:**

The initial scores were 4/4/4/6. One reviewer commented that they raised the score from 4 to 6. The other reviewers seemed positive about the planned changes, and the authors did address most of their specific issues by running further experiments, so I feel everybody else who gave 4 would in the end be inclined to change the score at least to 6. I think it is reasonable to assume the final scores of 6/6/6/6.

Therefore I recommend acceptance.

---

> ### Public Comment · ~Ilias_Aarab1 · 2026-03-12
> **Response to final AC remarks**
>
> Thank you for the careful reading of the revised manuscript and for the positive overall assessment.
>
> Regarding the overlap with Laurer et al. (2023), we agree that the wording “to a large extent” was too vague and could give the impression that the benchmark is merely reused from prior work. We revised this part in the manuscript to make the distinction clearer. More specifically, Laurer et al. (2023) use popular classification datasets primarily to train and study NLI-based encoders for zero-shot classification, whereas we repurpose a subset of datasets that pass our own selection criteria and quality checks, aggregate them into a unified benchmark, and build the surrounding benchmarking infrastructure needed for systematic comparison across different model families. In particular, BTZSC provides a standardized interface ([GitHub repository](https://github.com/IliasAarab/btzsc)), an aggregated public dataset ([Hugging Face dataset](https://huggingface.co/datasets/btzsc/btzsc)), and an easy-to-use public leaderboard ([Hugging Face Space](https://huggingface.co/spaces/btzsc/btzsc-leaderboard)). Our aim is thus not to train a particular model family, but to enable broad, reproducible, and accessible evaluation of zero-shot classification methods.
>
> On Table 2, your interpretation is correct: the reported standard deviations are computed across the datasets within each group, and they are not intended to reflect uncertainty around the final aggregate score. We agree that this distinction should be stated clearly. We have therefore revised the table caption to make the meaning of these values more transparent. We decided to retain them because, while they are not uncertainty measures, they still provide some indication of the heterogeneity of model performance across datasets within a group. At the same time, we agree that dataset-specific conclusions should rely on the full per-dataset results reported in the appendix.
>
> Finally, thank you for the minor comments. These have now been addressed in the text. We clarified why the relevant setting incurs higher computational cost, specified what we mean by standard NLI benchmarks (MNLI, ANLI, WANLI, FEVER-NLI, and LingNLI), and corrected the Figure 3b y-axis label.
>
> Thank you again for the constructive final feedback.

---

### Decision · Program_Chairs · 2026-01-26

Accept (Poster)